# A versatile toolbox for determining IRES activity in cells and embryonic tissues

Philipp Koch [ID][1,9], Zijian Zhang [ID][2,9], Naomi R Genuth [ID][2,5,9], Teodorus Theo Susanto [ID][2,6], Martin Haimann [ID][1], Alena Khmelinskaia [ID][3,4,7], Gun Woo Byeon [ID][2,8], Saurabh Dey [ID][1], Maria Barna [ID][2✉] & Kathrin Leppek [ID][1✉]

## Abstract

**Widespread control of gene expression through translation has emerged as a key level of spatiotemporal regulation of protein expression. A prominent mechanism by which ribosomes can confer gene regulation is via internal ribosomal entry sites (IRESes), whose functions have however, remained difficult to rigorously characterize. Here we present a set of technologies in embryos and cells, including IRES-mediated translation of circular RNA (circRNA) reporters, single-molecule messenger (m)RNA isoform imaging, PacBio long-read sequencing, and isoform-sensitive mRNA quantification along polysome profiles as a new toolbox for understanding IRES regulation. Using these techniques, we investigate a broad range of cellular IRES RNA elements including Hox IRESes. We show IRES-dependent translation in cir-cRNAs, as well as the relative expression, localization, and translation of an IRES-containing mRNA isoform in specific embryonic tissues. We thereby provide a new resource of technologies to elucidate the roles of versatile IRES elements in gene regulation and embryonic development.**

**Keywords** mRNA Translation Regulation; mRNA Isoforms; Internal Ribosome Entry Site; cellular IRES; Embryo Development
**Subject Categories** Methods & Resources; Translation & Protein Quality

## Introduction

Messenger (m)RNAs can contain conserved functional regions that regulate post-transcriptional gene expression. mRNA elements that reside in the untranslated regions (UTRs) can harbor the regulatory capacity to control mRNA translation and stability (Jia et al, 2020; Wu and Bazzini, 2023). One such category of 5' UTR regulatory elements are internal ribosome entry sites (IRESes) in certain eukaryotic mRNAs that facilitate alternate internal, cap-independent translation initiation, bypassing most required steps of conventional initiation (Leppek et al, 2018; Martinez-Salas et al, 2017; Brito Querido et al, 2023; Thompson, 2012b). A strong body of work over decades has unraveled clear IRES activity of diverse structured RNA elements in viruses for host-dependent translation of the viral genome (Johnson et al, 2017; Thompson, 2012b). Many viral IRESes, often assisted by protein cofactors, sustain the translation of viral mRNAs upon infection. These findings led to the fundamental question decades ago: Do such regulatory RNA sequences also exist in eukaryotes, and how can we find and characterize them? "Cellular IRES" RNA elements—for lack of a better term—are eukaryotic mRNA elements that may act as translational enhancers by non-canonical cap-independent recruitment of ribosomes for translation initiation. As all mRNAs are capped, clearly distinguishing IRES activity in a 5' UTR is difficult because IRES-containing mRNAs can be translated by both cap- and IRES-dependent mechanisms. The major challenge in the field is the lack of robust methods to confidently characterize cellular IRES elements. A deficiency of standardized tools has for decades discouraged the translation field to further explore the existence and diversity of cellular IRES activity.

The catalog of cellular IRES elements represents a list of often scarcely characterized examples. They were often found because they mediate continued translation upon cellular stimulation or stress (reviewed in (Leppek et al, 2018; Jackson, 2013; Holcik and Sonenberg, 2005; Komar and Hatzoglou, 2011; Spriggs et al, 2010)). The first IRES in a cellular mRNA was discovered in the stress-induced chaperone BiP mRNA by its activity in translation initiation in poliovirus-infected cells when host translation is inhibited (Sarnow, 1989; Macejak and Sarnow, 1991). Many IRES-containing viral transcripts solely rely on their IRES for translation (Thompson, 2012b). In contrast, mammalian mRNAs are capped and cap-dependent initiation is very strong compared to most known cases of IRES elements in steady-state conditions, which can be dynamic across cell states (Weingarten-Gabbay et al, 2016; Koch

[1]Institute of Clinical Chemistry and Clinical Pharmacology, Biomedical Center II (BMZ II), Venusberg-Campus 1, University Hospital Bonn, University of Bonn, Bonn 53127, Germany. [2]Department of Genetics, Stanford University, Stanford, CA 94305, USA. [3]Transdisciplinary Research Area "Building Blocks of Matter and Fundamental Interactions", University of Bonn, Bonn 53113, Germany. [4]Life and Medical Sciences Institute, University of Bonn, Bonn 53121, Germany. [5]Present address: Department of Molecular and Cell Biology, Howard Hughes Medical Institute, University of California, Berkeley, Berkeley, CA 94720, USA. [6]Present address: Epigenetic and Epitranscriptomic Systems, Genome Institute of Singapore, A*STAR, Singapore 138672, Singapore. [7]Present address: Department of Chemistry, Ludwig-Maximilians-Universität München, München 81377, Germany. [8]Present address: Department of Electrical and Computer Engineering, University of Washington, Seattle, WA 98195, USA. [9]These authors contributed equally: Philipp Koch, Zijian Zhang, Naomi R Genuth. ✉E-mail: mbarna@stanford.edu; kleppek@uni-bonn.de

et al, 2020). Mammalian IRES elements can have roles in a variety of biological functions even during ongoing cap-dependent translation. One example is the translation of two different proteins from a naturally occurring bicistronic transcript guiding organismal development (Cornelis et al, 2000). An unbiased screen of human genes has suggested that IRES RNA activity may be present in ~10% of human 5' UTRs (Weingarten-Gabbay et al, 2016). Therefore, it is reasonable that IRES regulation may contribute to widespread gene-specific translation regulation in many mammalian mRNAs. Of great interest is that IRES elements can make more direct contacts with the ribosome, by virtue of IRES *trans*-acting factors (ITAFs) (Komar and Hatzoglou, 2011), as well as direct base pairing or tertiary interactions with ribosomal RNA (Leppek et al, 2020, 2021; Sherlock et al, 2023). Given the complexity of translation regulation embedded in *cis*-acting elements in mammalian mRNAs, such increased possibilities for interactions between IRES elements and the ribosome renders important the discovery and determination of the regulatory contribution of mammalian IRES RNA activity to overall protein levels.

A key reason for our poor understanding of cellular IRES elements is the lack, limitations, and challenges of existing, often reporter-based tools to validate and functionally describe cellular IRES activity. These techniques are prone to experimental artifacts and noise (Jackson, 2013; Thompson, 2012a). Caveats that have led to false positive detection of IRES activity, particularly by the widely used, artifact-prone bicistronic reporters. Technically challenging is the distinction of the effect of an RNA element on both or either transcription and translation. There is no gold standard of required tests for genuine IRES activity of a cellular RNA element. We offer a spread of technologies as a guideline applied to a broad range of viral and cellular IRES elements.

We carefully selected cellular IRES elements as test cases for applying our toolset. We included five 5' UTR elements with previously assigned cellular IRES activity, several of which were also found in IRES activity screens: *Bcl2* (Sherrill et al, 2004; Marash et al, 2008; Kampen et al, 2019; C.-K. Chen et al, 2021), *c*-Myc (Weingarten-Gabbay et al, 2016; Stoneley et al, 1998; Nanbru et al, 1997), *CACNA1A* (Chen et al, 2021; Weingarten-Gabbay et al, 2016; Du et al, 2013), *Cofilin* (Choi et al, 2018), and *Fmr1* (Choi et al, 2019). We selected an additional set of five recently identified hyperconserved 5' UTRs with IRES activity (*Chrdl1*, *Dlx1*, *Gdf5*, *Sema3a*, *Zfx*), for which an impact on translation efficiency of their endogenous mRNAs has been shown (Byeon et al, 2021). These five 5' UTRs drive cell type-specific IRES-mediated initiation in bicistronic mRNA reporters. They are expressed in E11.5 mouse embryos and are important for embryonic development (Byeon et al, 2021). For example, similar to the Homeobox (Hox) gene cluster, *Dlx1* is a homeobox transcription factor that has critical roles in craniofacial patterning, and in the differentiation and survival of neurons in the brain (Panganiban and Rubenstein, 2002; Depew et al, 2005). We also include structured 5' UTR IRES mRNA elements from a subset of *Hoxa* mRNAs, in these cases distinguished as "IRES-like" elements, that the Barna lab has previously found to recruit ribosomes for ribosome-directed translation regulation in the embryo (Xue et al, 2015).

Only a strong combinatorial approach allows for a comprehensive interpretation of the contribution of IRES activity to multi-layered mammalian mRNA translation regulation. We have devised a diverse set of methodologies as a new standard to study IRES activity: (1) Circular RNA (circRNA) reporters: Plasmid-derived backspliced circRNA reporters encoding split EGFP that employ IRES elements for translation in cells which allows for IRES-screening and -mutagenesis. (2) Single-molecule fluorescent in situ hybridization (smFISH): Tissue-specific detection of IRES-containing mRNAs in mouse embryos using IRES- and coding region (CDS)-specific probes. This assay can precisely quantify physiological expression patterns across tissues to map transcript isoforms; (3) Promoterless reporters: Sensitive detection of luciferase protein and mRNA levels from Nanoluciferase (Nluc)-β-*globin* fusion mRNA reporters to detect artificial cryptic promoter activity from DNA sequences encoding IRES elements; (4) PacBio long-read sequencing: mRNA isoform detection of tissue-specifically expressed IRES-containing mRNA variants; (5) mRNA isoform detection: reverse transcription (RT)-quantitative PCR (qPCR)-dependent absolute quantification of isoform abundance and isoform-sensitive polysome-qPCR analysis of relative translation levels across isoforms. Together, this versatile guideline enables the field to confidently assess IRES activity from mRNA elements.

# Results

## circRNA: cellular IRES elements drive circRNA reporter translation

We first tested IRES activity of ten cellular 5' UTR elements in a circRNA reporter (Chen et al, 2021). circRNAs have diverse cellular roles in health and disease (Hansen et al, 2013; Jeck and Sharpless, 2014; Salzman et al, 2012, 2013; Panda et al, 2017; Zhang et al, 2016; Liu et al, 2024), but only a few hard-to-identify functional circRNA-encoded mammalian proteins are known (Legnini et al, 2017; Liang et al, 2019; Pamudurti et al, 2017; Yang et al, 2018; Zhang et al, 2018). There is an increasing interest in exploiting circRNAs for sustained translation of stable therapeutic biologics (Wesselhoeft et al, 2019; Wen et al, 2022), for which important optimization steps have recently been achieved (Chen et al, 2021, 2023; Nielsen et al, 2022; Pamudurti et al, 2017). circRNAs present a powerful tool to study cap-independent translation. We adapted a plasmid-based circRNA system that uses cellular backsplicing of a reporter pre-mRNA into circRNA for the study of cellular IRES elements (Chen et al, 2021) (Fig. 1A). In the split-EGFP reporter plasmid, a full-length (FL) EGFP is fused from an N- and C-terminal fragment by backsplicing mediated by ZKSCAN1 sites (Liang and Wilusz, 2014; Chen et al, 2021), with the IRES element upstream of the 5' EGFP fragment. Such split-EGFP approaches have recently been invaluable for the optimization of large gene delivery for muscular dystrophy therapy (Lindley et al, 2024). In our case, FL EGFP is exclusively translated through ribosome recruitment by the IRES. The linear reporter mRNA does not encode full EGFP, but two fragments encoded in opposite directions (Fig. 1A). Promoter activity from the insert would result in a linear capped mRNA that would only translate the 5' EGFP fragment. The linear mRNA also encodes cap-initiated mRuby as a transfection control. We apply the circRNA reporter to test five recently identified ultraconserved 5' UTR elements with IRES activity from a screen: *Chrdl1*, *Dlx1*, *Gdf5*, *Sema3a*, *Zfx* (Byeon et al,

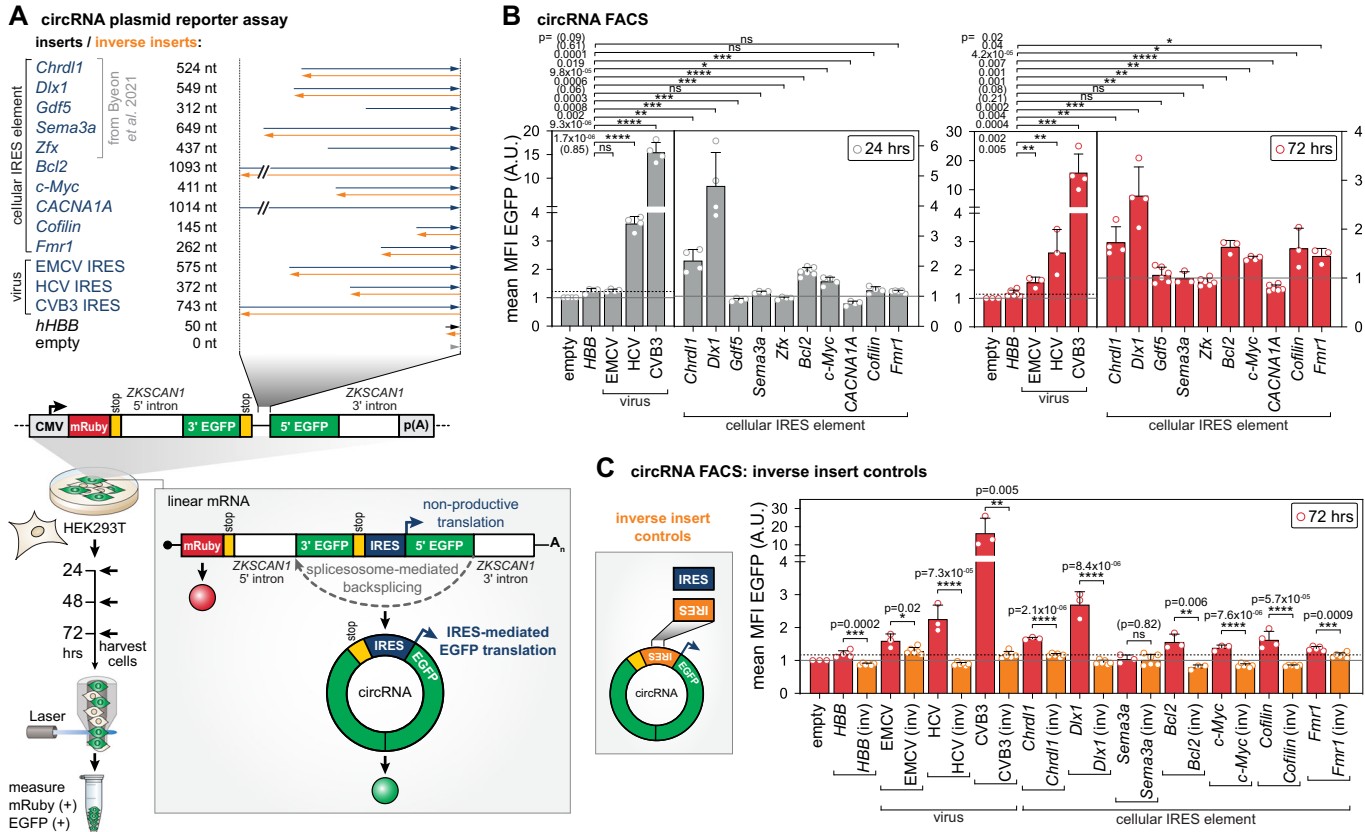

**Figure 1.  Specific IRES activity in an in-cell circRNA reporter system.**

(**A**) Experimental outline of the circRNA reporter assay based on the mRuby-ZKSCAN-split-EGFP plasmid for the screening of IRES activity of the different tested insert sequences, including inverse sequences as controls for circRNA translation activity dependent on insert length and GC-content. Following plasmid transfection, the expression of the reporter system under CMV promoter control leads to the linear pre-mRNA which is circularized through spliceosome-mediated backsplicing (gray box). Cells were harvested after 24 h and 72 h, and their mRuby signal (transfection control) and EGFP signal (readout for IRES activity) was detected. Schematic partially adapted from (Chen et al, 2021). (**B**) Calculated median fluorescence intensities (MFIs) of EGFP of the mRuby+ subfractions are shown at 24 h and 72 h. Bar graphs are indicating mean values ± SEM, $n = 3$–6. Empty vector control was normalized to 1; ns not significant. (**C**) We tested the inverse sequences of active cellular IRESes from (**B**) as controls for circRNA translation activity dependent on insert length and GC-content, which are identical in the forward and inverse sequences. We did a pairwise comparison of the respective inserts. MFIs of EGFP of the mRuby+ subfractions are shown at 72 h, as presented in (**B**). Bar graphs are indicating mean values ± SEM, $n = 3$–6. Empty vector control was normalized to 1. In all figures, data were presented as mean, SD or SEM as stated, and *$P \leq 0.05$ was considered significant (ns: $P > 0.05$; *$P \leq 0.05$; **$P \leq 0.01$; ***$P \leq 0.001$; ****$P \leq 0.0001$). Tests, two-tailed unpaired Student's $t$ test if not stated otherwise, and specific $P$ values used are indicated in the figure or legends. Please see the "Quantification and Statistical Analysis" section in "Methods" for details. We present all exact $P$ values for all bar graphs in the figure. Source data are available online for this figure.

2021). We also included five 5' UTR elements with previously assigned cellular IRES elements: *Bcl2*, *c-Myc*, *CACNA1A*, *Cofilin*, and *Fmr1*. We included three viral IRES elements from encephalomyocarditis virus (EMCV), hepatitis C virus (HCV) and coxsackievirus B3 (CVB3) as references, as well as *human hemoglobin subunit beta* (hHBB) 5' UTR as a negative control (Fig. 1A). EMCV, HCV, CVB3 and hHBB were not tested as controls in previous studies (Chen et al, 2021), and we therefore have included them. We transiently transfected human HEK293T cells and measured relative EGFP signal of the mRuby + cells from 24 and 72 h post transfection (Figs. 1B and EV1). At 24 h, compared to the empty vector, we observe low background IRES activity for hHBB, no activity for EMCV and strong HCV and CVB3 IRES activity (Fig. 1B). The EMCV, HCV, and CVB3 IRES mediate 0–1.4-fold, 4–5-fold, and 12.5–13.6-fold higher IRES activity compared to the hHBB control at 24 and 72 h, respectively (Fig. 1B). We find that in this assay, EMCV follows slower kinetics

than the other two viral IRESes and is only robustly active from 72 h on (see also Fig. 2B). We find that for the cellular IRES elements, at 24 and 72 h, *Chrdl1* and *Dlx1* are very active, and *c-Myc* and *Bcl2* are moderately active, with 1.7–1.5-fold, 3.7–2.3-fold, 1.2-fold, and 1.4–1.5-fold higher IRES activity compared to hHBB, respectively. *Cofilin* and *Fmr1* are active only at 72 h with a 1.4-fold and 1.2-fold increased IRES activity over hHBB; and *Gdf5*, *Sema3a*, *Zfx*, *CACNA1A* are inactive even at 72 h (Fig. 1B). Multiple upstream AUGs (uAUGs) in the insert sequence out-of-frame of EGFP, particularly for *CACNA1A*, may contribute to the observed absence of IRES activity (Fig. EV2A). From the five cellular IRES elements from (Byeon et al, 2021), we find that 2/5 are active in HEK cells. For all five, IRES activity has been found to vary across different cell types (Byeon et al, 2021). It remains to be determined whether required cofactor ITAFs absent in HEK are needed for IRES activity regulation of the inactive constructs. When applied to various cell types, we think the circRNA assay is a great tool to also

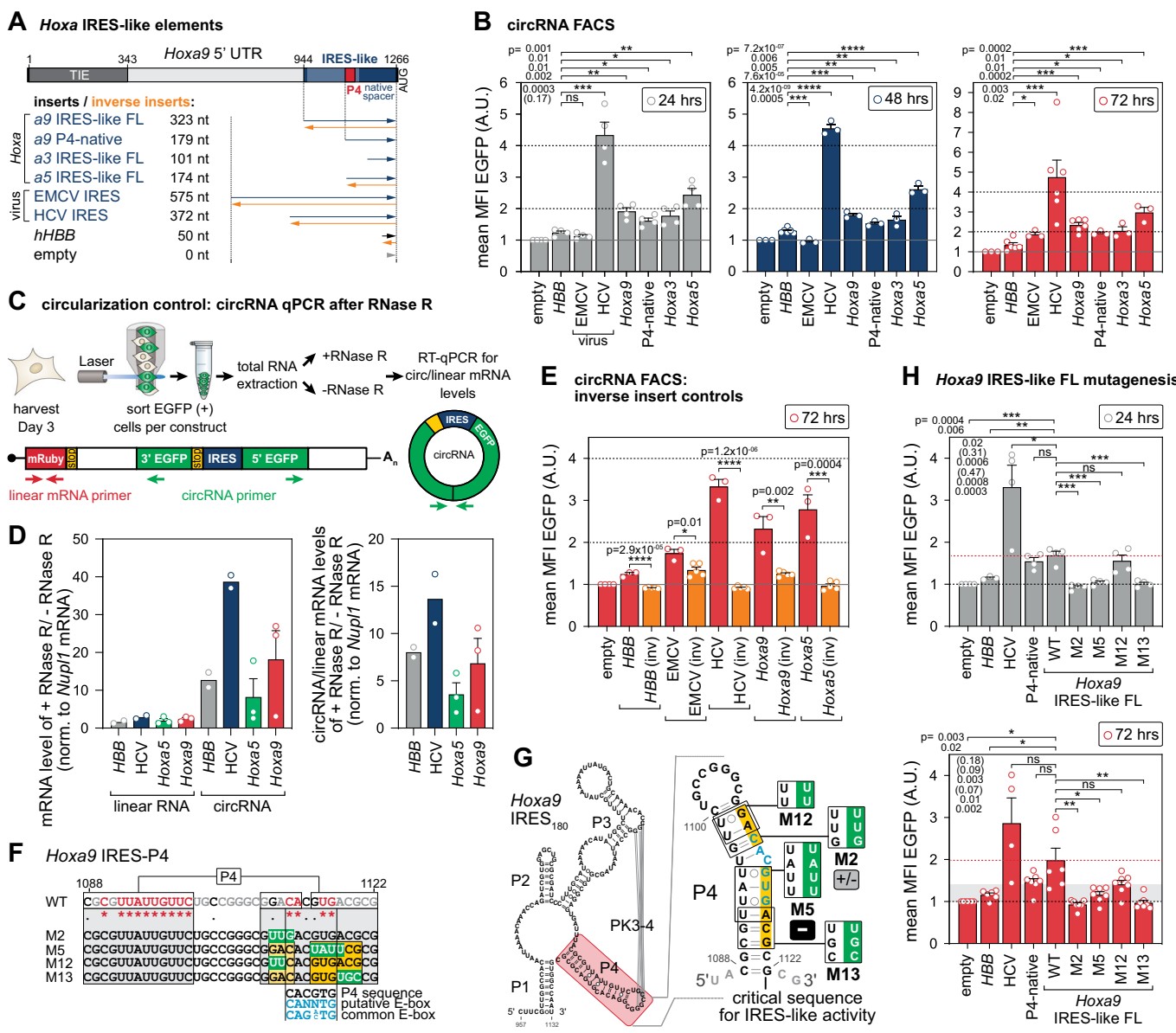

**Figure 2. *Hoxa* IRES-like activity in circRNA reporters.**

(A) Experimental outline of the circRNA reporter assay based on the mRuby-ZKSCAN-split-EGFP plasmid for the screening of IRES-like activity of different insert sequences of *Hoxa* IRES-like elements, as described in Fig. 1A. (B) Calculated MFIs of EGFP of the mRuby+ subfractions are shown at 24 h, 48 h, and 72 h. Bar graphs are indicating mean values ± SEM, n = 3–6. Empty vector control was normalized to 1; ns, not significant. (C) Schematic overview of the procedure to validate the linear and circRNA content of the HEK293T cells post transfection using RNase R digestion followed by RT-qPCR quantification (primers indicated as red and green arrows). Outward-directed circRNA-specific qPCR primers exclusively detect circRNAs. The EGFP primer set only leads to an amplification product after successful backsplicing. (D) The linear and circRNA content of the cells is shown after RNase R digestion relative to undigested samples after normalization to *Nupl1* mRNA for each construct. The enrichment of circRNA content over RNase R digestion indicates the successful backsplicing in cells (left). Relative circRNA/linear RNA levels increase upon RNase R digestion after normalization to *Nupl1* (right). Bar graphs are indicating mean values ± SEM, n = 2–3. In the case of n = 2, we do not show error bars. (E) We tested the inverse sequences of *hHBB*, EMCV, HCV, *Hoxa9*, and *Hoxa5* as controls and calculated MFIs of EGFP of the mRuby+ subfractions 72 h as described in Fig. 1C. Bar graphs indicate mean values ± SEM, n = 3–6. Empty vector control was normalized to 1. (F) Substitution mutations were mapped onto the linear P4 sequence (labeled according to conservation) that were tested in the context of the FL *Hoxa9* IRES-like element (323 nts). Nts critical for IRES activity are highlighted in yellow, also in (G). Numbers refer to nt positions within the *Hoxa9* 5′ UTR. The suspected 6-nt putative and common E-box is aligned accordingly (blue). (G) Schematic representation of the mouse *Hoxa9* RNA secondary structure model of the 180 nt-long *Hoxa9* IRES-like element RNA (Xue et al, 2015) with P4 highlighted (red). The structure model of the P4 stem-loop and disruptive substitution mutations mapped onto the P4 stem are shown. P4 mutants moderately active are labeled "+/−", and inactive mutants are labeled "−". M12 and M13 mutants are new. (F, G) Are adapted from (Leppek et al, 2020). (H) Calculated MFIs of EGFP of the mRuby+ subfractions are shown at 24 h and 72 h. Mutants (M2, M5, M12, M13) are tested against the FL IRES-like *Hoxa*9 WT sequence. Bar graphs are indicating mean values ± SEM, n = 4–7. Empty vector control was normalized to 1. Source data are available online for this figure.

filter out false positive IRES activity possibly detected by orthogonal methods. To control for the specificity of circRNA translation dependent on insert length, RNA structure and GC-content, we tested the inverse sequences of all active IRES sequences from Fig. 1B (*hHBB*, EMCV, HCV, CVB3, *Dlx1*, *Chrdl1*, *Bcl2*, *c-Myc*, *Cofilin*, *Fmr1*) compared to their forward sequence (Fig. 1C). We observe that all inverse inserts tested completely or nearly completely diminished IRES activity to the levels of the empty vector or *hHBB* (Fig. 1C). These data confirm that cellular or viral IRES elements can specifically recruit ribosomes to circRNAs for internal translation initiation. These findings reflect the usefulness of circRNAs to describe bona fide IRES elements from previous screen results and other studies, where this analysis has not yet been possible.

## circRNA: *Hoxa* IRES-like elements and mutagenesis in circRNA reporters

Previous work from the Barna lab and collaborators has shown that the ribosome is recruited to structured 5' UTR IRES-like mRNA elements in a subset of *Hoxa* mRNAs for translation (Xue et al, 2015). The transcription factors of the Hox cluster are master regulators of body plan formation and are among the most spatiotemporally controlled transcripts (Mallo and Alonso, 2013). IRES-like elements are critical for gene expression of several *Hoxa* mRNAs and anterior-posterior patterning of the mouse axial skeleton as foundational examples of IRES-dependent translational control in mammalian development (Kondrashov et al, 2011; Xue et al, 2015).

We tested a subset of *Hoxa* IRES-like elements (Xue et al, 2015), including the *Hoxa9* FL IRES-like element and its shorter derivative P4-native (Leppek et al, 2020) in circRNAs (Fig. 2A). Upon transient transfection we measured the relative EGFP signal of mRuby+ cells after 24, 48, and 72 h (Fig. 2B). Compared to the controls, we see activity for all *Hoxa*-derived IRES-like elements to about half the levels of HCV. *Hoxa9* IRES-like FL and P4-native display similar activity. *Hoxa5* IRES-like FL drives the strongest activity of the *Hoxa* elements tested. We emphasize that the *Hoxa9* IRES-like FL has half the activity of the strongest viral IRES tested (HCV), which reveals considerable activity, sustaining a comparable activity to HCV and EMCV at 120 h (Fig. EV3A). For *Hoxa* IRES-like elements, we also detect uAUGs in the insert sequences (Fig. EV2B), but they do not prevent IRES-like activity. As EGFP accumulates over time, the EMCV IRES only shows activity in HEK293T cells with increasing incubation times over 72 h, but the relative activities between all the *Hoxa* IRES-like elements are similar at 48 and 72 h. To qualitatively show that the detected IRES-like activity is due to circRNA translation, we confirmed the existence of cytoplasmic circRNAs by RNase R-treatment of total RNA to degrade all RNAs except circRNAs. We FACS-sorted mRuby + /EGFP+ cells at 5 days post transfection, and treated total RNA with RNase R for RT-qPCR analysis (Figs. 2C and EV3B). Normalized mRNA levels after +/-RNase R indicate that linear mRNA is depleted in the *hHBB*, HCV, *Hoxa9* and *Hoxa5* samples, and circRNA species accumulate 8-38-fold for all four constructs (Figs. 2D, left and EV3C). The circ/linear RNA ratio indicates that the circRNA is enriched 4–14-fold for the four constructs (Fig. 2D, right). This RNase R-mediated enrichment of circRNA by degradation of linear species, including to an extent of

the abundant reference mRNA *Nupl1*, shows the existence of circRNA irrespective of insert identity. Variability in the detected relative circRNA levels between inserts (Fig. 2D) may stem from the elaborate RNase R procedure (Fig. EV3B). circRNA detection confidently supports that the EGFP signal is derived from IRES-like activity. Similar to other cellular IRES-like elements (Fig. 1), the inverse insert controls tested for *Hoxa5* and *Hoxa9* specifically reduced IRES-like activity to that of the empty vector or *hHBB* (Figs. 2E and EV3D).

In the *Hoxa9* IRES-like structure, we previously identified a 35 nt-long stem-loop termed P4 (Xue et al, 2015; Cheng et al, 2015) that recruits the 40S ribosomal subunit through mRNA-rRNA interactions with rRNA expansion segment 9S as visualized by cryo-EM (Leppek et al, 2020). P4 acts as a "translation enhancer" that can increase m⁷G-capped mRNA reporter translation (Leppek et al, 2020). Through mutagenesis (Fig. 2F,G), we isolated a 4-nucleotide (nt) P4 mutation, termed M5, that selectively decreased ribosome binding and endogenous *Hoxa9* mRNA translation (Leppek et al, 2020). This allowed us to conclude that the M5 mutation decreases translation of *Hoxa9* mRNA in the cytoplasm independent from transcription. Nevertheless, it appears that there is the possibility of a 6-nt-long putative E-box motif (consensus motif: 5'-CANNTG-3'). It is a rather degenerate sequence in the genome to which transcription factors may bind (Pan et al, 2023), that can be mapped to the P4 in the DNA (Fig. 2F). It would overlap with mutants M5 (inactive) and M2 (moderately active) (Leppek et al, 2020) (Fig. 2G). We designed two new mutants M12 (2 nt) and M13 (3 nt) flanking each side of the putative E-box. Particularly, we tested these four 2–4 nt P4 mutants in the context of the 323 nt-long *Hoxa9* IRES-like FL in circRNAs (Fig. 2F,G). At 24 and 72 h, we observe that M2, M5, and M13 are clearly inactive and display EGFP expression at the level of the empty vector and *hHBB*, while M12 is dampened, to half the activity of the wild-type (WT) at 72 h (which is not significant) (Fig. 2H). As previous mutations changed 3–4 nt, mutating only 2 nt in M12 in the distal P4 may not be sufficient to affect overall *Hoxa9* IRES-like activity. Indeed, a previous 3-nt mutation (M2) overlapping M12 also only showed moderate activity (Leppek et al, 2020) (Fig. 2F,G).

Overall, our studies show that plasmid-derived split-reporter circRNAs are an elegant and sensitive system to investigate a range of IRES-like element activities and their mutagenesis, in any transfectable cell type. A major argument for circRNAs, and for their superiority over artifact-prone, bicistronic reporter mRNAs, is that promoter activity of IRES inserts as a source of false positive IRES activity can be excluded. We also provide appropriate controls and clearly confirm specific IRES activity from several cellular IRES-like elements, including *Hoxa3*, *a5* and *a9*. Strong IRES activity found in circRNAs is also directly relevant for emerging synthetic mRNA medicine.

## Single-molecule mRNA imaging: local *Hoxa9* mRNA isoform expression in the embryo

mRNA isoforms with distinct 5' UTRs can be expressed across tissues that include or exclude IRES elements. What has been lacking in the field is a reliable method to characterize these alternative isoforms. We aimed to visualize in a cell- and tissue-specific manner the quantitative tissue-specific expression pattern

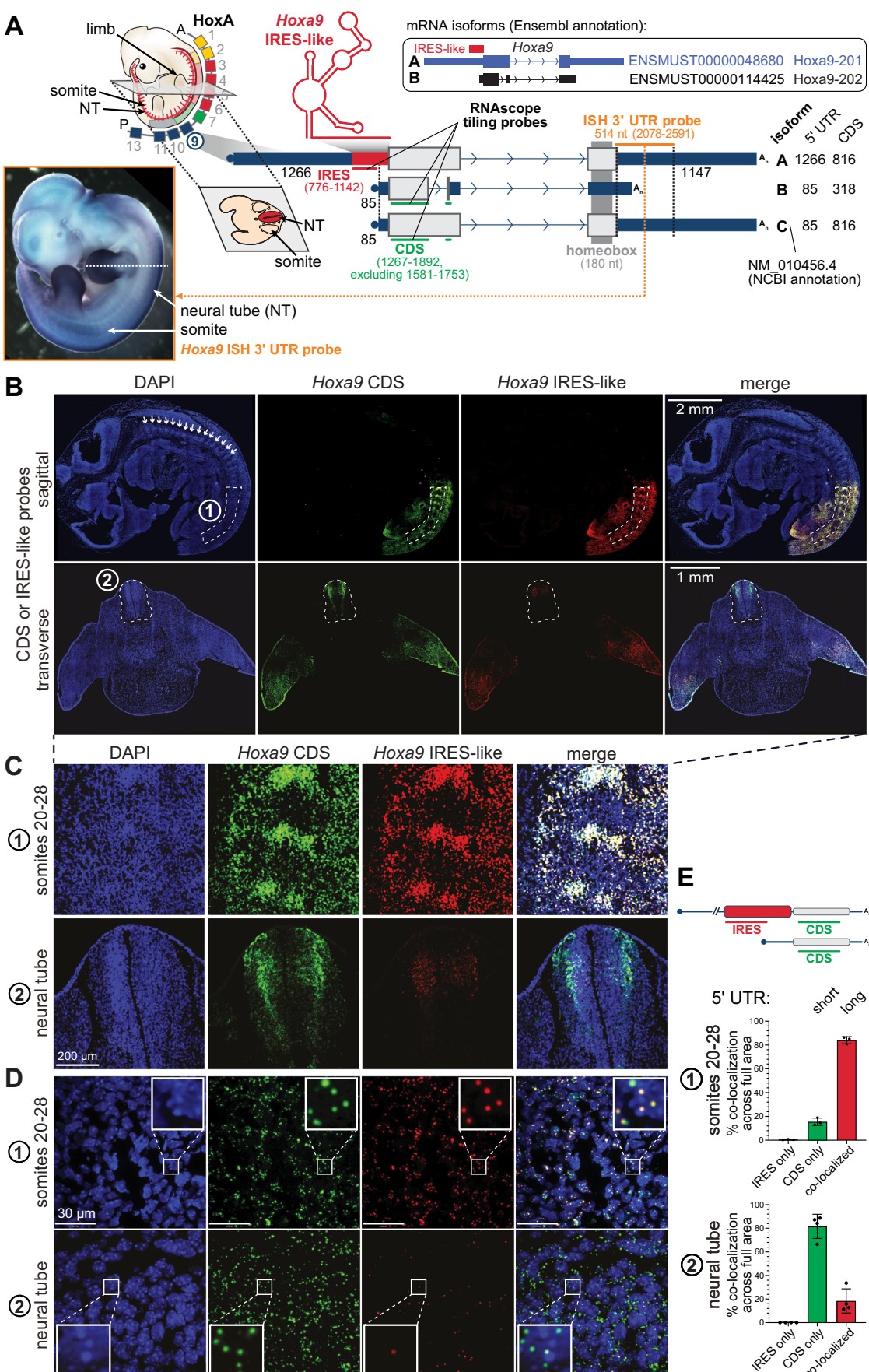

**Figure 3.  Tissue-specific *Hoxa9* mRNA isoform expression in the mouse embryo by smFISH.**

(**A**) Model depicting the expression pattern of the HoxA cluster genes in mouse embryos. The Hoxa9 gene is expressed in posterior somites and the neural tube (NT), and can be expressed in mRNA isoforms with either a long 5' UTR and full CDS (isoform A; ENSMUST00000048680.8) or short 5' UTR and truncated CDS (isoform B; ENSMUST00000114425.3), as annotated in mouse GENCODE release M32 (GRCm39). A third mRNA isoform C is annotated in NCBI to contain the short 5' UTR and the full CDS (NM_010456.4). Functional RNA elements in the mRNA 5' UTR include an IRES-like element (Xue et al, 2015). IRES, internal ribosome entry site. Regions targeted by smFISH probes are indicated as "IRES" and "CDS" with the respective coordinates of the ENSEMBL-annotated isoforms. Insert: Whole-mount in situ hybridization (ISH) of a representative WT stage E11.5 embryo using a *Hoxa9* probe targeting the 3' UTR region as annotated in the gene model (orange). The dashed white line indicates the position of the transverse section used for smFISH in (**B**). (**B**) Representative images of E11 mouse embryo sections immunostained with DAPI (blue), the *Hoxa9* CDS probe (green), the *Hoxa9* IRES-like probe (red) and the 3-plex Negative Control probes (white). Boxed regions #1 and #2 indicate the regions for which quantifications were performed shown in (**D**, **E**). White arrows point to individual somites. (**C**, **D**) Representative zoomed-in views of the numbered boxes in (**B**). #1 represents somites 20–28 and #2 represents the anterior NT. The scale bars indicate the different magnifications in (**C**, **D**). (**E**) Quantification of the proportion of the long and short 5' UTR isoforms in the respective regions from (**D**) as indicated by dotted lines in (**B**). Colocalized CDS and IRES-like probe signals represent the long 5' UTR isoform, whereas the non-colocalized CDS-only probe signals represent the short 5' UTR isoform. IRES-only signal is not detected at a biologically relevant level. Bar graphs indicate mean values ± SD, *n* = 3–4 embryos (somites *n* = 3; neural tube *n* = 4). Source data are available online for this figure.

of an IRES-like element-containing 5' UTR compared to other isoforms, exemplified for mouse *Hoxa9* mRNA (Xue et al, 2015). For reference, we performed whole-mount in situ hybridization (ISH) of *Hoxa9* mRNA in stage E11.5 embryos (lateral view) using a 3' UTR-specific probe (Fig. 3A, insert) to indicate overall *Hoxa9* mRNA expression that captures different 5' UTR variants (isoform A and C) explained later.

We use the RNAscope technique (Wang et al, 2012) for advanced smFISH of *Hoxa9* mRNA isoforms in sectioned embryos (section sites indicated in the ISH in Fig. 3A). We use mRNA small tiling probes that distinguish mRNA isoforms with different 5' UTRs. smFISH allows for precise quantification of the expression and abundance of the different isoforms. The target-specific signal is amplified in intact cells and tissues of sectioned embryos. Single mRNA molecules can be imaged and precisely quantified as individual dots in the tissue (Wang et al, 2012; Gaspar and Ephrussi, 2015; Gross-Thebing et al, 2014). There are two FL protein-coding mRNA isoforms, and a third isoform, annotated for mouse *Hoxa9* mRNA with different protein-coding regions and 5' UTRs. We here name them mRNA isoform A–C for convenience (A: long 5' UTR, 1266 nt, complete CDS, ENSMUST00000048680 that contains the IRES-like element; B: short 5' UTR, 85 nt, short CDS, ENSMUST00000114425; C: short 5' UTR, 85 nt, complete CDS, NM_010456.4) (Fig. 3A). Both A and C isoforms encode the FL functional HOXA9 protein, while isoform B generates an early stop codon through alternative splicing which results in the lack of the C-terminal homeobox domain (Fujimoto et al, 1998). The CDS probes (green) tile a region of the CDS shared between all isoforms. The IRES probes (red) tile the 323 nt-long IRES-like region in the long 5' UTR isoform A.

We use co-localization of the probes to distinguish A and C isoforms at the single-molecule level: colocalized signal indicates the long 5' UTR isoform and the CDS-only signal indicates the other two short 5' UTR isoforms. In the sectioned E11 embryo, we focused on somites 20–28, as HOXA9 is strongly expressed caudally to somite level 21–22, which corresponds to the tenth thoracic vertebrae (Sekimoto et al, 1998; Burke et al, 1995). There is an overall clear tissue-specific distribution of the signal and mRNA staining with low background (Fig. EV4) in the longitudinal and sagittal section of WT E11 embryos for the long 5' UTR mRNA isoform (Fig. 3B,C). For *Hoxa9* mRNA in sagittal embryo sections, both the short and long 5' UTR *Hoxa9* mRNA isoforms are expressed. In the posterior somites (somites 20–28), the long 5'

UTR isoform A is predominantly expressed in nearly 80% of total *Hoxa9* mRNA (Fig. 3D,E). No IRES-only signal is expected to exist. Indeed, we do not observe IRES-only probes that do not co-localize with the CDS probes across the embryo (Fig. 3E). Before, with CDS-targeting probes used in whole-mount ISH, we were blind to these distinct isoform expression patterns and their overall abundances (Xue et al, 2015). Our new data indicate that already at the transcription start site (TSS) level, a tissue-specific decision is made to predominantly employ the long mRNA isoform containing the IRES-like element in *Hoxa9* mRNAs at posterior regions of the embryo (somites 20–28) that leads to translation of the homeobox-containing HOXA9 transcription factor. smFISH with IRES-like-specific probes allows for distinct detection of tissue-localized IRES-like-containing mRNA isoform expression. Strikingly, in more anterior regions of the embryo where there is strong *Hoxa9* mRNA expression selectively in the neural tube, we see a complete reversal of the transcript isoforms that are abundantly expressed. We see that more than 80% of short 5' UTR transcript is predominantly expressed at these anterior positions in the transverse embryo sections, where the complete neural tube could be captured (Fig. 3C). Given the complexities of *Hox* gene expression, we observe transcript isoform-specific regulation and differential expression of *Hox* transcripts with unique 5' UTRs along the A–P body axis that could only be assessed by examining endogenous embryonic tissue sections. We envision that smFISH will be applicable to confirm and quantify expression of other mRNAs with IRES-like elements in various tissues. We also anticipate that transcript-specific isoform retention or exclusion of selective 5' UTR isoforms could be an important mechanism for the regulation of translation and protein expression in vivo.

## Promoterless reporters: cryptic promoter activity from IRES element DNA is not relevant for IRES mRNA element function

A 5' UTR IRES element may simultaneously function in both transcription and translation—regulating mRNA translation in the cytoplasm and the corresponding DNA sequence potentially acting as a cryptic or real promoter in the nucleus. To explore this, some studies have used "promoterless" luciferase reporter plasmids that contain IRES elements upstream of a reporter CDS but lack a promoter (Jackson, 2013; Thompson, 2012a; Akirtava et al, 2022). However, these reporters do not assess IRES-driven translation but

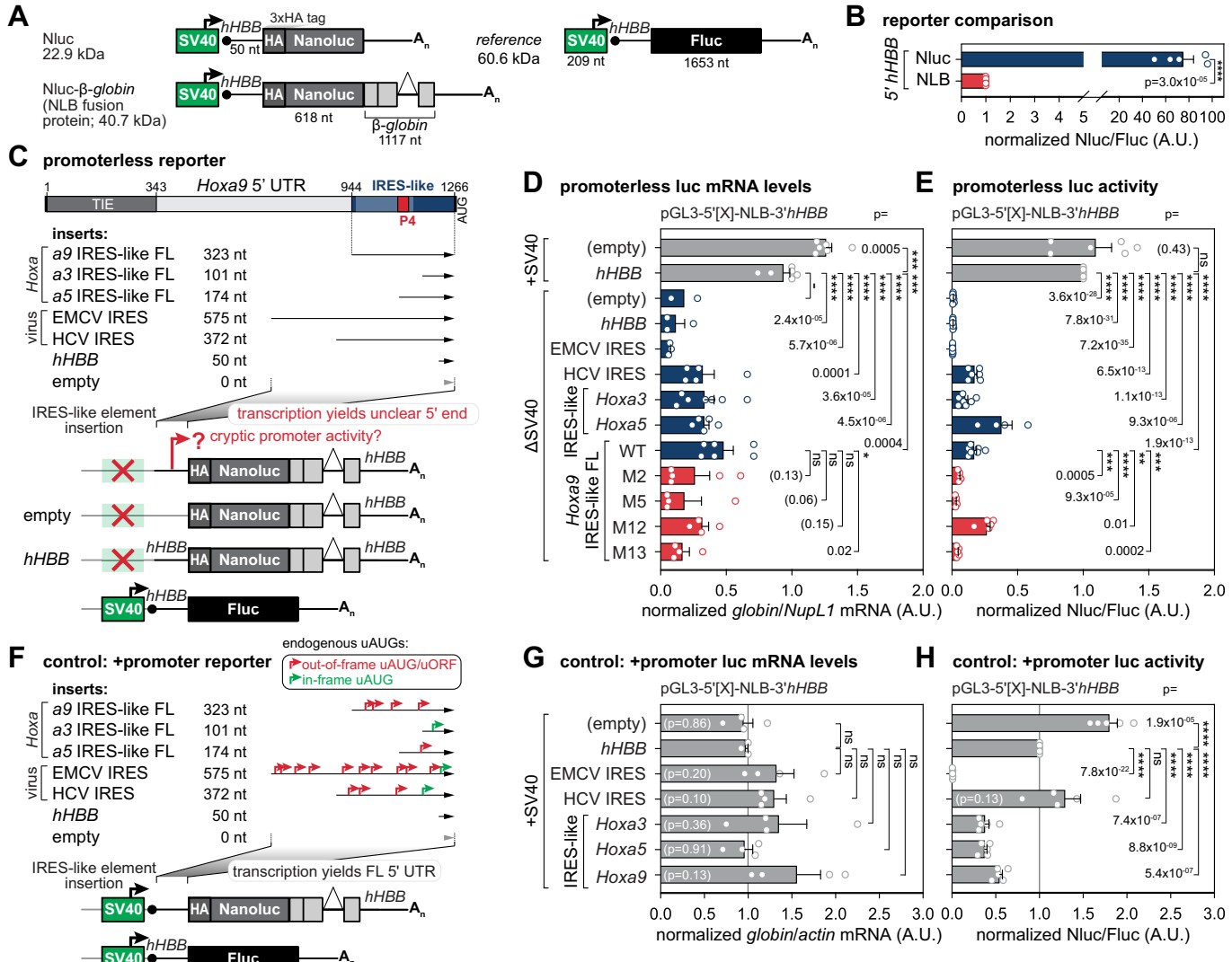

rather detect unintended cryptic transcription, as they cannot differentiate between IRES-mediated and cap-dependent translation. A key issue with monocistronic promoterless reporters is that they allow any upstream DNA element to function as an artificial promoter, producing capped mRNA isoforms with truncated 5' UTRs. This makes it impossible to determine if an inserted sequence functions as an IRES at the mRNA level. On the other hand, the commonly used bicistronic luciferase assay, designed to test IRES activity, can also be problematic because the intergenic IRES sequence can drive artificial transcription to generate mRNAs that only contain the second reporter CDS. Despite this limitation, we have previously validated the IRES activity of *Hoxa* IRES-like elements using such bicistronic reporter assays with mRNA-normalization to control for this effect (Leppek et al, 2020, 2021).

To improve the sensitivity and specificity of reporter assays, we designed a novel Nanoluciferase (Nluc)-β-*globin* (NLB) fusion reporter, which differs from previous systems by including an intron-containing rabbit β-*globin* reporter gene (Leppek et al, 2020; Ozgur et al, 2010; Simsek et al, 2017) in the same plasmid, as well as a more sensitive Nluc reporter (Osuna et al, 2017; England et al,

2016). This ensures that the derived mRNA produces an Nluc-β-*globin* (NLB) fusion protein (Fig. 4A), enabling reliable mRNA detection through RT-qPCR and sensitive translation measurements via luminescence. The *hHBB* 5' and 3' UTRs attached to the NLB sequence mediate efficient translation (Kozak, 1994; Babendure et al, 2006). The *hHBB*-Fluc plasmid serves as a transfection control (Leppek et al, 2020, 2022). Thereby, while in previous reporters there could be DNA contamination from plasmids in RT-qPCR analysis, unable to distinguish the plasmid or cDNA origin of the amplicon, the NLB fusion reporter overcomes this important disadvantage by using intron-spanning primers specific for the β-*globin* exon sequence.

We first compared the relative activities of Nluc and Nluc-β-*globin* (NLB) expressed from the simian virus 40 (SV40) promoter in transiently transfected mouse C3H/10T1/2 cells and detected a 70-fold decrease in the activity of NLB (Fig. 4B). This effect likely stems from a less compact fold of Nluc in the fusion protein (Fig. EV5) (Lin et al, 2023; Tomabechi et al, 2016). Still, the advantage of confident mRNA detection of the mature spliced reporter mRNA with NLB outweighs the reduced luciferase activity

**Figure 4. Cryptic promoter activity from IRES element DNA in promoterless reporters is not relevant for IRES mRNA element function.**

(A) Schematic of the SV40 promoter-driven Nanoluciferase (Nluc) reporter ORF (621 nt) attached to an N-terminal 3xHA tag, the Nluc-β-*globin* (NLB) reporter of Nluc fused to an intron-containing rabbit β-*globin* gene that leads to the NLB fusion protein, and the control Firefly (Fluc) reporter with the calculated molecular weight of their encoded protein in kilo Dalton (kDa). mRNA reporters contain *hHBB* as a control 5′ UTR. IRES-like elements were introduced into the Nluc 5′ UTR instead of *hHBB*. The Fluc reporter mRNA with the *hHBB* 5′ UTR serves as an internal transfection control from a separate plasmid. See also Fig. EV5. (B) Reporter plasmids were transiently transfected into mouse C3H/10T1/2 cells. Relative luciferase activity is expressed as a Nluc/Fluc ratio. Average luciferase activity ± standard error of the mean (SEM), n = 5; NLB was normalized to 1; A.U. arbitrary units. (C) Schematic of the topology of regulatory elements in the mouse *Hoxa9* 5′ UTR and promoterless reporter assay design. The 323 nt-long *Hoxa9* FL IRES-like RNA element (*a9* IRES-like FL) harbors the P4 stem-loop. Cryptic promoter activity from *Hoxa* IRES-like elements (*a3, a5, a9*) is tested by inserting FL IRES-like elements upstream NLB in a plasmid lacking the SV40 promoter (ΔSV40). Viral IRES controls (EMCV, HCV), an empty vector control (empty; no insert in the 5′ UTR region), and control 5′ UTR (*hHBB*) were included. A co-transfected control reporter (*hHBB*-Fluc) under an active SV40 promoter served as reference. Cryptic transcription from the promoterless plasmids will lead to unclear 5′ ends and 5′-shortened 5′ UTR fragments. (D) Normalized Nluc mRNA levels from promoterless NLB constructs were measured in transiently plasmid-transfected C3H/10T1/2 cells as in (B). Cells from the same transfection were split in half for mRNA (D) and protein (E) analysis. Average Nluc mRNA levels are expressed as respective *globin/NupL1* mRNA levels ± SEM, n = 2–7. Promoter-containing plasmids ( + SV40) of empty and *hHBB* inserts served as expression controls. Empty and *hHBB* 5′ UTR serve as negative controls for ΔSV40 constructs; +SV40-*hHBB* was normalized to 1. In the case of n = 2 we do not show an error bar nor P value. (E) Normalized Nluc/Fluc luciferase activity from promoterless constructs was measured from samples according to (D) and as described in (B). Average luciferase activity ± SEM, n = 4–8; +SV40-*hHBB* was normalized to 1; ns not significant. (F) Control assay with SV40+ promoter reporters. In analogy to (C), regular SV40 promoter activity with *Hoxa* IRES-like elements (*a3, a5, a9*) by inserting FL IRES-like elements upstream NLB in a plasmid with the SV40 promoter ( + SV40). Transcription from the SV40 promoter will yield FL 5′ UTRs as annotated. Capped mRNAs containing IRES-like elements as sole 5′ UTRs will contain upstream AUGs (uAUGs), uORFs, and out-of-frame uAUGs (red) that will reduce translation from the main ORF. In-frame uAUGs (green) will contribute to translation from the main AUG. (G) Normalized Nluc mRNA levels from +SV40 promoter NLB constructs were measured in transiently plasmid-transfected C3H/10T1/2 cells as in (B). Cells from the same transfection were split in half for mRNA (G) and protein (H) analysis. Average Nluc mRNA levels are expressed as respective *globin/actin* mRNA levels ± SEM, n = 3–4. Empty and *hHBB* 5′ UTR serve as controls; +SV40-*hHBB* was normalized to 1. For all pairwise comparisons, the P values are not significant (ns). (H) Normalized Nluc/Fluc luciferase activity from promoterless constructs was measured from samples in (G) and as described in (B). Average luciferase activity ± SEM, n = 3–4; +SV40-*hHBB* was normalized to 1. Source data are available online for this figure.

of still sufficient sensitivity. In promoterless plasmids, we encoded the NLB fusion with the *hHBB* 3′ UTR, but deleted the upstream SV40 promoter (ΔSV40). The empty or the *hHBB* 5′ UTR served as 5′ UTR controls. Their promoter-containing versions (SV40 + ) served as positive controls for NLB expression. We included *Hoxa3, a5, a9* IRES-like elements of various lengths (101–323 nt), viral EMCV and HCV IRES controls, and P4 mutants in the FL *Hoxa9* IRES-like element (Fig. 4C). We measured relative β-*globin* reporter mRNA levels (Fig. 4D) and Nluc/Fluc activity (Fig. 4E) from the same transfections. Compared to the strongly transcribed empty and *hHBB* SV40+ controls, we see diminished transcript production from ΔSV40 constructs for the empty, *hHBB* 5′ UTR, and EMCV IRES (Fig. 4D). However, we detect promoter-like activity and corresponding mRNA abundance compared to the *hHBB* SV40+ control for the HCV IRES, *Hoxa3, a5* and *a9* IRES-like elements. This indicates that without a promoter, random longer stretches of DNA can be biased to be used as promoters that would not be active in native chromatin, which highlights the artificial nature of the assay. In fact, these DNA elements in the presence of the SV40 promoter do not add any transcriptional activity (see below). Moreover, previous data showed that the HCV IRES DNA sequence can have promoter activity in reporters (Dumas et al, 2003). Together, these data highlight the artificial nature of the assay. We next tested the same 2–4 nt mutants in the FL *Hoxa9* IRES-like element as in circRNAs (Fig. 2F,G). Compared to the WT insert, M2, M5, and M13 mutations show decreased promoter activity that is only significant for M13 at the mRNA level, the other three mutations do not affect *Hoxa9* IRES-like-derived promoter activity (Fig. 4D). Thus, while the M2, M5, and M13 affect IRES activity (Fig. 2), only the mutant M13 which is not in the putative E-box but rather a 3 nt mutation downstream mildly affects transcription. Next, we asked if the relative luciferase activity corresponds to its transcript abundance (Fig. 4E). Importantly, we measured less luciferase expression compared to the mRNA levels in the case of *Hoxa3* and *Hoxa9* IRES-like elements. For the *Hoxa9*

mutants we only see luciferase activity for M12 in the range of the WT *Hoxa9* IRES-like element. For the other three mutants we see reduced luciferase activity (Fig. 4E). These findings highlight that potentially random alternative TSSs in the 5′ UTR can result in mRNAs with truncated 5′ ends that may be initiated early or out-of-frame of the NLB CDS, leading to different translation rates. We conclude that it is more likely that any 5′ UTR region is favored to be used as a promoter in promoterless plasmids. As a control, we re-introduced the SV40 promoter into all reporter plasmids which should result in capped mRNAs with FL 5′ UTRs (Fig. 4F). Here, transcription is standardized for all constructs but the cap-dependent translation rate differs between constructs. As expected, this yielded no significant differences in relative mRNA levels across the tested 5′ UTRs (Fig. 4G), but different relative luciferase activities (Fig. 4H). 5′-capped mRNAs with IRES elements are likely predominantly initiated cap-dependently. Multiple uAUGs in these IRES sequences likely lead to early initiation out-of-frame of the NLB CDS, most apparent in EMCV (Fig. 4H).

These results do not allow any interpretation of DNA inserts in their role as IRES elements in these promoterless assays. Furthermore, regulatory elements in highly conserved genes can co-exist and be repurposed in DNA and mRNA in the nucleus and cytoplasm, respectively, and may indicate the need for intricate gene-specific expression control. Unfortunately, these monocistronic constructs are used with the aim of showing that IRES elements can artifactually generate cryptic transcription in a promoterless context (Akirtava et al, 2022). We find several important conclusions about promoterless reporters: (1) Any 5′ UTR sequence placed in a promoter region has potential to have promoter activity and to generate mRNA; (2) Existing plasmid systems could not differentially quantify by RT-qPCR if an amplicon originates from plasmid DNA contamination or corresponding cDNA; (3) In monocistronic promoterless reporters, IRES sequences are artificially placed in a different location from their endogenous one in chromatin which can contribute to artificial results in absence of a

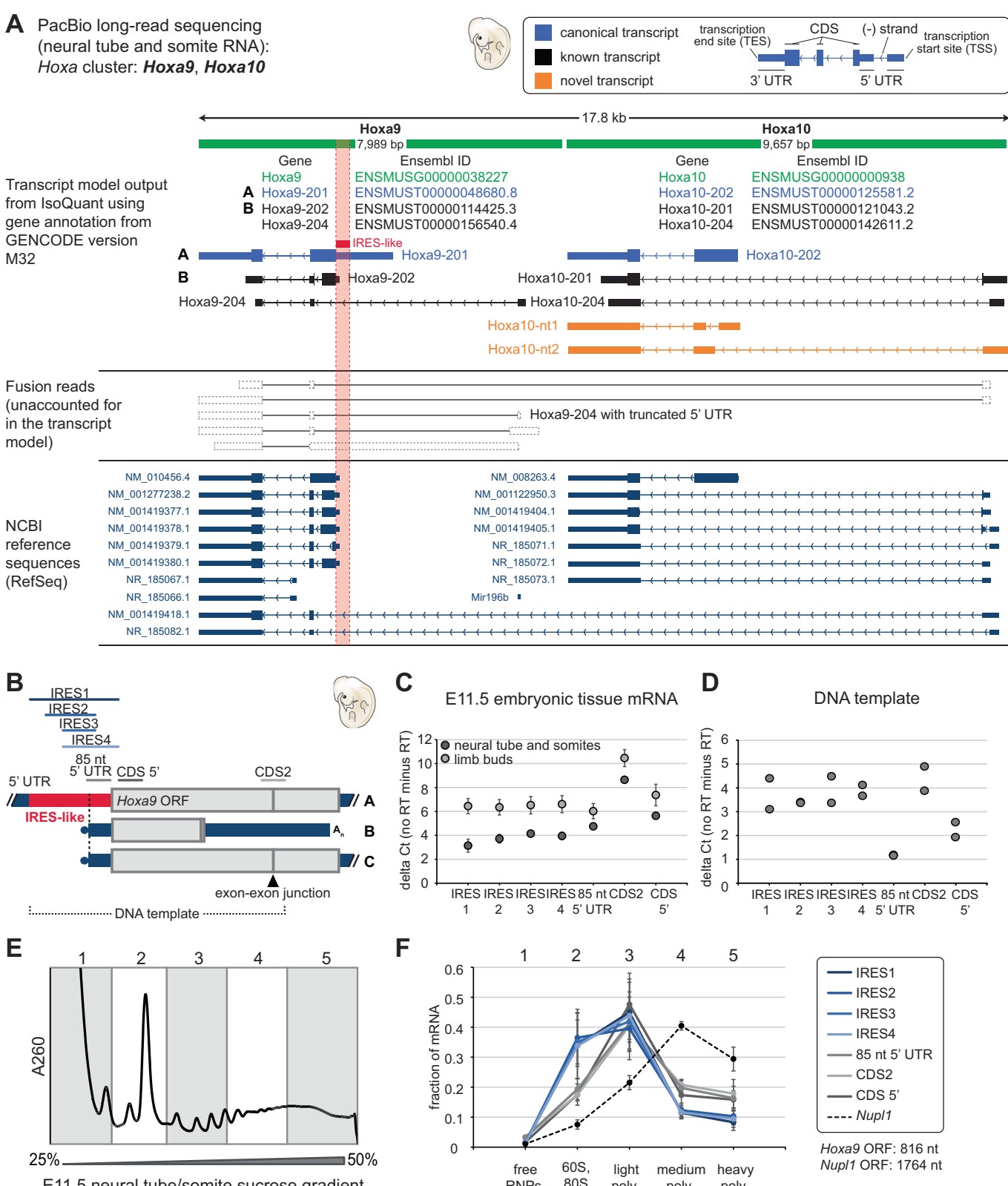

A PacBio long-read sequencing (neural tube and somite RNA): *Hoxa* cluster: **Hoxa9**, **Hoxa10**

physiological promoter; (4) With previous reporting of only luciferase activity from promoterless reporters (Akirtava et al, 2022), but not the respective mRNA levels, it is important to detect both relative mRNA levels and luciferase levels because of the discrepancy between them and therefore confounding the interpretations that can be made. We conclude that promoterless reporters do not add any functional value to understand IRES RNA elements, but induce random, artificial promoter activity of 5' UTR

**Figure 5.** *Hoxa9 and Hoxa10* mRNA expression in mouse embryo tissues by PacBio HiFi sequencing and *Hoxa9* mRNA abundance and translation in embryonic tissues.

(A) Illustration of a genome browser snapshot showing the transcript models predicted by IsoQuant, NCBI RefSeq, and corrected HiFi reads from the IsoQuant output around Hoxa9 and Hoxa10 loci (chr6:52,200,050-52,217,850). The 5' UTR IRES-like element is indicated by the red box. Despite 5' UTR truncations, reads of identical intron chains as in isoform A were assigned to the Hoxa9-201 IRES-containing reference transcript. Similarly, reads of identical intron chains as in isoform B but with an extended 3' UTR were assigned to Hoxa9-202. Only two novel transcripts were detected for the Hoxa10 gene. The previously reported Hoxa9/Mir196b fusion transcript is similar to Hoxa9-204 with a truncated 5' UTR. Otherwise, there was no novel isoform identified that indicates any fusion transcripts. The full list of the corrected reads is shown in Appendix Figs. S1 and 2. (B) Schematic of *Hoxa9* mRNA amplicons used for RT-qPCR analysis. The IRES1, IRES2, IRES3, IRES4, and CDS2 amplicons are the same as in (Ivanov et al, 2022) and (Leppek et al, 2020). In addition, we created new primer sets to amplify an upstream coding sequence region within exon 1 (CDS 5') and the 85 nt directly 5' to the start codon (85 nt 5' UTR). The latter is contained within the IRES-like region (Akirtava et al, 2022) which was putatively defined as the entire *Hoxa9* 5' UTR sequence. The CDS2 amplicon within the coding region spans an exon–exon junction. (C) RT-qPCR results from micro-dissected E11.5 mouse embryonic tissue mRNA. Values shown are the difference in Ct values between tissue samples (three biological replicates each) and a respective no-RT control. Error bars are SD, $n = 3$. (D) RT-qPCR results using 0.1 fg of a plasmid containing the *Hoxa9* IRES-like element and CDS as a template. Values shown are the difference in Ct values between DNA samples ($n = 2$) and a no-RT control as in (C). (E) Representative sucrose gradient fractionation trace from a E11.5 neural tube and somite sample (C) and quantification of the fraction of total mRNA found in each of the five portions of the gradient, as demarcated in the schematic. (F) Three biological replicates were performed, and the values for each *Hoxa9* mRNA (ENSMUST00000048680) amplicon are shown as mean $+/-$ SEM, $n = 3$. qPCR amplicons correspond to as illustrated in (B). For comparison, a highly translated housekeeping mRNA, *Nupl1* (ENSMUST00000225805), is also shown (dotted black line), as well as the respective mRNA ORF lengths. Source data are available online for this figure.

## PacBio long-read sequencing: no evidence for *Hoxa9-Hoxa10* mRNA fusion isoforms in embryonic tissues and analysis for additional IRESes

Long-read sequencing can be used to confirm the expression of different mRNA isoforms that may contain a known IRES element sequence in a specific tissue. So far, it can only be used to detect and distinguish differentially expressed mRNA isoforms for known IRES elements, but it does not aid the discovery of new IRES elements nor add to their functional characterization. Instead, it only supports the presence or absence of these elements. We next sought to test the existence of the previously described *Hoxa9* mRNA isoforms and the putative *Hoxa9/a10* fusion transcript (Akirtava et al, 2022). The latter was purported to explain the existence of the IRES-containing 5' UTR (Akirtava et al, 2022). We performed PacBio HiFi long-read sequencing (Hon et al, 2020) from micro-dissected WT E11.5 mouse embryo somite and neural tube tissues (Figs. 3A and 5A; Appendix Figs. S1 and 2; Dataset EV1). It can discover which mRNA isoform variants are expressed, but is limited in terms of accurate comparison of mRNA abundances across molecules of different lengths. Technical challenges with PacBio long-read sequencing can lead to false positive transcript identification (Soneson et al, 2016), which is particularly challenging for transcripts with identical intron chains or splice junctions, but different TSS or transcription end sites (TES). This is the case for the different TSS in the *Hoxa9* mRNA isoforms A and C that have very different 5' UTR lengths (Fig. 3A). We performed transcript discovery and quantification using the IsoQuant algorithm, a new gold standard for transcript annotation (Prjibelski et al, 2023) (Fig. 5A). We used the same PacBio kit as was previously used for the generation of the public ENCODE long-read dataset (Reese et al, 2023). It was previously independently used to claim no detection of the IRES-like-containing long *Hoxa9* mRNA isoform and detection of a *Hoxa9/a10* fusion transcript (Akirtava et al, 2022). Importantly, in our data, a *Hoxa9/a10* fusion transcript (Akirtava et al, 2022; Reese et al, 2023) was

not detected by IsoQuant transcript discovery (Fig. 5A). Instead, two previously unannotated transcript isoforms for *Hoxa10* mRNA were discovered (Fig. 5A, orange).

With PacBio diffusion-based sequencing runs (Rhoads and Au, 2015), shorter fragments will be sequenced much more than longer ones introducing strong technical bias toward shorter mRNA isoforms. Particularly, RT drop-off during reverse transcription that starts at the 3' end can make data interpretation and isoform calling difficult (Soneson et al, 2016). This effect is best visualized by the example of *Dlx1* described later (Appendix Fig. S3). Given our parameters using the GENCODE release M32 for gene annotation, we detect the reference 1266 nt-long *Hoxa9* 5' UTR isoform A (ENSMUST00000048680) that encodes the IRES-like RNA element and the HOXA9 transcription factor. But we could not detect the short 5' UTR *Hoxa9* mRNA isoform C (NM_010456.4) as described in (Akirtava et al, 2022). This may be due to the algorithm prioritizing isoforms A and B present in the gene annotation, while being unable to distinguish between isoforms A and C because they have the same intron chain and only differ in their TSS. Nevertheless, we detected an annotated isoform B (ENSMUST00000114425) with a short 5' UTR of 85 nts and a truncated CDS producing a protein lacking the homeodomain (Fujimoto et al, 1998) (Figs. 3A and 5; Appendix Figs. S1 and 2; Dataset EV1). The reads assigned to this isoform have a longer 3' sequence, but with an identical intron structure. Additionally, we investigated *Dlx1*, *Chrdl1*, *Cofilin*, and *Fmr1* mRNA expression, and found that for all of these genes, their annotated IRES-containing mRNA isoforms are also expressed in E11.5 mouse embryo somite and neural tissues (Appendix Figs. S3–6). As we have already seen for the *Hoxa9* 5' UTR region (Appendix Figs. S1 and 2), a large percentage of the corrected reads, especially prominent for the reads aligned to *Chrdl1* and *Dlx1*, were shown to have truncated 5' ends (Appendix Figs. S3–6). Importantly, it is not likely that the mapped reads indicating variability in 5' end-truncated 5' UTRs represent biological diversity and variability of TSS and 5' UTRs. Rather, these truncations more likely correspond to technical variability and RT drop-off effects which create a bias towards shorter 5' UTRs. We can observe this truncated 5' ends not only in *Hoxa9* (Appendix Figs. S1 and 2) but also in *Chrdl1* (Appendix Figs. S3 and 4), *Dlx1* (Appendix Fig. S3),

and *Fmr1* (Appendix Figs. S5 and 6), which are all expressed in the embryo. As such, despite advances in long-read sequencing to identify novel transcripts, we need to account for PacBio sequencing limitations especially in definitively capturing the complete 5' UTRs without the support of other orthogonal methods. Given this technical bias, we do not recommend at this point to include PacBio long-read sequencing, but hopefully upon technical improvements in the future, in the toolbox for cellular IRES element mapping.

## Isoform quantification and polysome-qPCR: *Hoxa9* mRNA isoforms are translated in embryonic tissues

Given a 5' UTR and mRNA isoform repertoire in which IRES elements may be included or not, it is important to determine if an IRES-containing mRNA is translated or not. This can be achieved by polysome fractionation of tissue material and isoform-specific RT-qPCR analysis to determine to which extent an IRES-containing isoform is associated with translating ribosomes. Importantly, for mRNAs with an IRES in the 5' UTR that do not have a built-in upstream cap-repressive element as in *Hoxa9* mRNA (Xue et al, 2015), they can simultaneously be translated by an IRES or cap-dependently. We aimed at absolute quantification of mRNA isoforms of *Hoxa9* in embryo tissues with primer sets that span different 5' UTR and CDS regions as a test case (Fig. 5B). With this, we directly quantified the abundance of the *Hoxa9* IRES-like-containing mRNA by RT-qPCR on cDNA from micro-dissected E11.5 mouse embryo limb buds or neural tube and somites. Contrary to the previous E11.5 somite data (Ivanov et al, 2022) relied upon in a previous study (Akirtava et al, 2022) that reported an ~8 Ct cycle difference between IRES and CDS amplicons, we found that the CDS2 and IRES-like amplicons only differed by ~4 Ct cycles (Fig. 5C). This discrepancy could be due to differences in tissue harvesting techniques or RT, 3' end-biases, or ability to process highly structured RNA elements. Furthermore, the CDS 5' and 85 nt 5' UTR amplicons, the latter representing the short 5' UTR previously proposed to be the primary isoform (Akirtava et al, 2022), had similar Ct cycle counts to the IRES-like amplicons, suggesting that the CDS2 amplicon may overestimate *Hoxa9* mRNA abundance or that the *Hoxa9* mRNA isoform landscape may be more complex as can be resolved by tissue dissection and RT-qPCR. To absolutely quantify the abundance and to account for primer efficiency differences, we cloned a *Hoxa9* DNA template from E11.5 neural tube and somite cDNA using a primer set spanning all qPCR amplicons. Sequencing of this product showed correct splicing, further confirming that the IRES-like element is present in fully processed mRNAs. We used serial dilutions of this DNA template to create a standard curve in the qPCR (one dilution is shown in Fig. 5D). Accordingly, we determined that CDS2 is more than 2-fold as abundant as the CDS 5' or 85 nt 5' UTR amplicon, which are found to be at similar levels. From this, we can conclude that the IRES-like element is at least 8–15% of the abundance of the CDS 5' or 85 nt 5' UTR amplicon. Up to 15% abundance of the IRES-like element mRNA in tissues harboring both the neural tube and somites analyzed by RT-qPCR is vastly underestimated as shown by the 80% abundance of the long isoform in somites 20–28 compared to the short isoform in RNAscope (Fig. 3). As these results clearly demonstrate, smFISH is more tissue-specific than embryo micro-dissection for tissue isolation and RT-qPCR analysis which can never be complete and thus cannot clearly distinguish distinct isoform

expression between tissues. In addition to tissue specificity, there may also exist chemical modifications or structures in the *Hoxa9* mRNA that prevent efficient cDNA synthesis but not hybridization of the RNAscope probes.

Next, we performed sucrose gradient analysis of polysomes to examine the association of specific mRNA isoforms with ribosomes by RT-qPCR analysis on micro-dissected E11.5 neural tube and somite samples (Fig. 5E), using the same primer sets as for isoform quantification (Fig. 5B,D). We find two clusters of mRNA isoforms in terms of their distribution across the sucrose gradient from E11.5 neural tube and somites (Fig. 5F). We find that IRES-like amplicons were present in polysomes, peaking in the light polysome (2–5 ribosomes) fraction, importantly indicating that IRES-like-containing mRNAs are not only expressed but also actively translated. The CDS and 85 nt 5' UTR amplicons also peaked in the light polysomes, showing a more similar pattern to the IRES-like amplicons than to the highly translated housekeeping mRNA *Nupl1* with a longer CDS (1764 nt, NM_170591.1). The differences in 60S subunit and monosome abundance between the IRES-like amplicons and the CDS and 85 nt 5' UTR amplicons may be indicative of different kinetics and modes of ribosome recruitment between *Hoxa9* mRNA isoforms which will require further mechanistic experimentation to be unraveled. Overall, we find all detected *Hoxa9* mRNA isoforms to be efficiently translated in embryonic tissues. Therefore, absolute isoform quantification and polysome-qPCR is useful to determine how abundant and how well translated a specific IRES-like containing mRNA is.

# Discussion

We present an extensive spread of different techniques as a toolbox to critically investigate the function of cellular IRES elements (Fig. 6). For all viral IRESes, for 6/10 cellular IRES elements we tested, and for three representative *Hoxa* IRES-like elements, we find support for them to drive bona fide IRES initiation in circRNA reporters. Therefore, employing circRNA reporters represents a strong technical advance in the clear delineation and identification of IRES elements. Of all the tools we describe, circRNA reporters represent the greatest advance in our understanding of IRES elements. When applied to different cell types for cofactor requirement, ideally the physiological cell type the mRNA containing a putative IRES is expressed in, we believe that the circRNA assay can robustly filter out false positive IRES activity found through previous orthogonal methods. We suggest performing the circRNA assay as an initial step in functional IRES characterization.

We find the *Hoxa9* IRES-like element to be expressed as part of a long 5' UTR isoform A containing the IRES-like element that is visualized by smFISH imaging, and that is translated in mouse embryo tissues in a tissue-specific manner (Fig. 3). In the translation profile of *Hoxa9* mRNA isoforms from the combined neural tube and somite tissues by RT-qPCR, all isoforms with the long and the short *Hoxa9* 5' UTR are detected and translated (Fig. 5). This is in accordance with our smFISH imaging data of ~80% of the long 5' UTR isoform being expressed in posterior somites and over 80% of the short 5' UTR isoform being expressed in the anterior neural tube (Fig. 3). As these results clearly demonstrate, smFISH can clearly distinguish distinct isoform expression between tissues. This is an advantage compared to

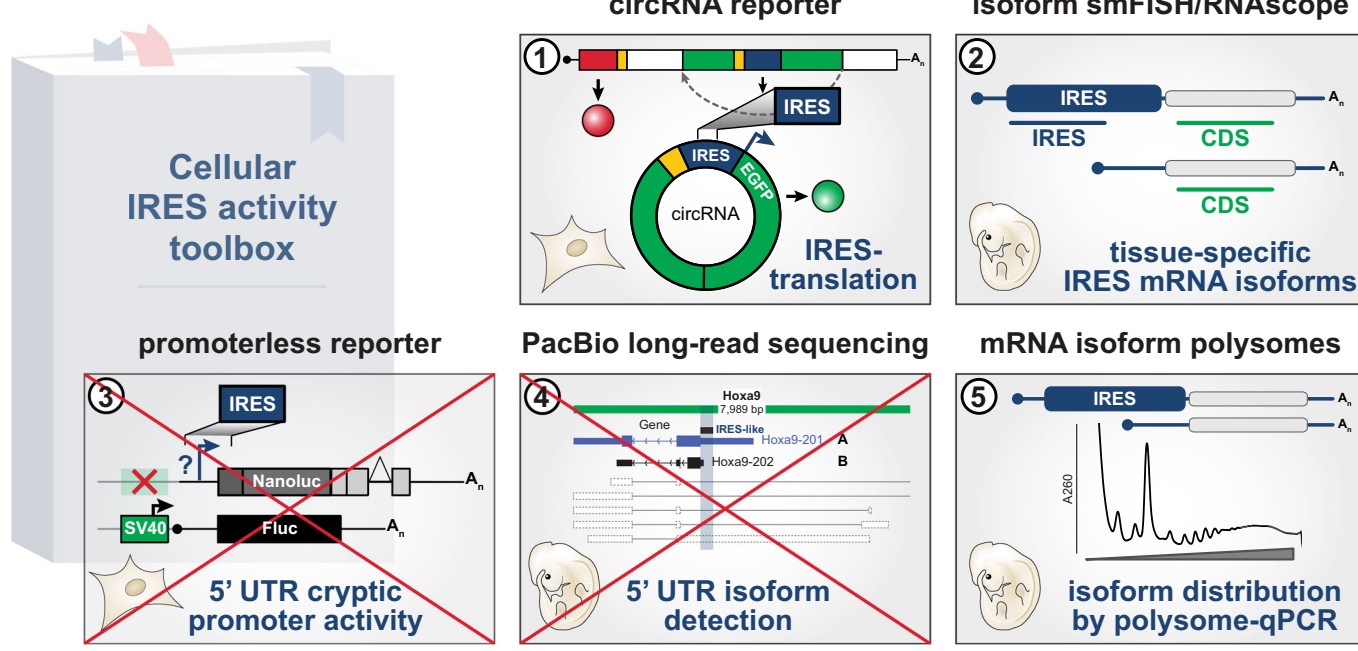

**Figure 6. A versatile toolbox to determine IRES activity in cells and embryonic tissues.**

Schematic summary of the presented technologies as a toolbox for investigation and functional characterization of cellular IRES activity.

embryo tissue micro-dissection where this isoform specificity is lost. Moreover, chemical modifications or RNA structures in the *Hoxa9* mRNA may prevent efficient cDNA synthesis that are overcome by hybridization of the RNAscope probes. We envision that this toolset is widely transferable to various IRES elements from any species for confirmation in diverse tissues. We hypothesize that this tissue-specific expression of transcript isoforms that do or do not contain IRES elements may be more widely employed to control the specificity of translation in a cell type-specific manner. In vivo, genomic deletion of the *Hoxa9* IRES-like element, which is upstream of the short 5' UTR *Hoxa9* isoform, does not appear to affect transcriptional output at the level of mRNA detected by RT-qPCR, while the protein is diminished (Xue et al, 2015). Moreover the *Hoxa9* IRES-deleted 5' UTR mice show a canonical Hox homeotic transformation in vertebral elements derived from somites, where the long *Hoxa9* IRES-containing transcript is predominantly expressed.

Defining the role of different mRNA isoforms, that may have different 5' UTRs, in how they affect transcription and/or translation is a difficult problem. This challenge has only very recently been explored with isoform-sensitive long-read sequencing and RNA base editing-mediated technologies to discover transcriptional and/or translational differences at isoform level and the responsible mRNA features (Jagannatha et al, 2024). With regard to our tested examples, the IRES-like regions of the *Hoxa9* gene are higher conserved than the rest of the 5' UTR in the long isoform (Xue et al, 2015), and *Dlx1* and *Chrdl1* were part of a screen for IRES activity in overall hyperconserved 5' UTRs (Byeon et al, 2021). Genome-wide mouse RefSeq transcripts annotated with long, hyperconserved 5' UTRs were experimentally confirmed to be expressed and IRES activity was detected for many of them in bicistronic reporters (Byeon et al, 2021).

In that previous study, cryptic promoter activity was indicated for a very small number of IRES candidates, and not for *Hoxa*, *Dlx1*, and *Chrdl1* IRES elements. Our circRNA reporter system provides an invaluable resource to definitely test for bona fide IRES activity. While it has been impossible to definitively verify IRES activity among hundreds of transcripts typically employed for large IRES screens (Weingarten-Gabbay et al, 2016; Chen et al, 2021; Byeon et al, 2021), we encourage testing of subsets of these using orthogonal circRNA reporters in the future.

Importantly, our findings illustrate that promoterless reporters are not a useful technology to measure IRES activity, as we find induced promoter activity from IRES DNA inserts in the absence of a promoter, that we do not see in the presence of one (Fig. 4). Moreover, even if respective DNA sequences corresponding to 5' UTR regions have functionally significant promoter activity in the nucleus, it does not preclude them from co-existing functionally at the DNA and RNA level, also serving as mRNA regulatory elements in translation. In particular, one of the earliest hypotheses about the function of hyperconserved sequences, such as a subset of IRES elements tested in our study, may result from multiple, overlapping constraints e.g., overlapping transcription factor binding sites, binding sites overlapping with RNA structural or protein-coding constraints, or overlapping protein-coding and RNA structural constraints (Siepel et al, 2005). The question remains whether these sequences are so highly conserved due to the selective pressure of maintaining these putative functions or due to hyperefficient repair of any mutations in these regions (Snetkova et al, 2021). Additionally, it is critical to point out that the ability to predict reporter assay activities from genomic features, while an important result, is not actually inconsistent with their function as translational regulatory elements. The presence of transcriptional regulatory elements in 5'

UTR regions and their activity at the DNA level are not uncommon (Burke and Kadonaga, 1996; ENCODE Project Consortium et al, 2007), and a transcription-translation balance is relevant in cancer (Jana et al, 2023). Common are also alternative TSS or 5' UTRs that result from the presence of such regulation (Chia et al, 2017; Tresenrider et al, 2021). It would be extraordinary to expect DNA- and RNA-level regulatory sequences to evolve in mutual exclusivity. On the contrary, transcriptional and translational effects may co-vary, potentially with observing greater levels of conservation. Thus, it is interesting to discover that the conserved 5' UTR regions are enriched in other regulatory elements such as transcriptional regulatory elements, especially in the most conserved ones.

IRES elements in the genome may provide new layers of regulatory specificity to gene expression. Similar to transcription where *cis*-acting promoters or enhancers are guided in *trans* by DNA binding factors, IRES elements may provide the basis for more directed and regulatory translational control by the ribosome or additional RNA-binding factors. This may diversify gene expression. As the field is turning to more genome-wide methods to identify IRES elements (Weingarten-Gabbay et al, 2016; Chen et al, 2021; Byeon et al, 2021), this can provide an entry point for in-depth characterization of individual 5' UTR examples, which will profit from the guideline of technologies presented (Fig. 6). Such work will broaden our understanding of the function of IRES activity in mRNAs. For example, circRNA reporter assays have the advantage that only strong IRES activity will be detected while cell type specificity can be tested. Strong IRES elements found through this assay are directly relevant for synthetic biology and recently emerging mRNA therapeutics. Meanwhile, at the organismal level, smFISH is capable of detecting distinct mRNA isoforms with varying 5' UTRs in embryo sections. Together, multiple experiments that converge on functional characterization of IRES elements and 5' UTR mRNA isoforms provide a culmination of technologies in a toolbox that can confirm IRES-dependent translation and the ability of IRES elements to be entry points for ribosome-mediated control of gene expression.

# Methods

### Reagents and tools table

| Reagent/resource | Reference or source | Identifier or catalog number |
|---|---|---|
| **Experimental models** | | |
| C3H/10T1/2 embryonic mesenchymal mouse cells | ATCC | Cat# CCL-226 |
| HEK293T human cells | ATCC | Cat# CRL-3216 |
| FVB/NJ mouse (*M. musculus*) | The Jackson Laboratory | Cat# 001800 |
| **Recombinant DNA** | | |
| Plasmids used and generated | This study | Table S1 |
| **Oligonucleotides and other sequence-based reagents** | | |
| DNA oligonucleotides for cloning, RT-qPCR analysis | This study | Table S2 |

| Reagent/resource | Reference or source | Identifier or catalog number |
|---|---|---|
| Raw PacBio long-read sequencing E11.5 data | This study | GEO: GSE250214 |
| GENCODE release M32 (GRCm39) | GENCODE 2021 | https://doi.org/10.1093/nar/gkaa1087 |
| **Chemicals, enzymes, and other reagents** | | |
| Cycloheximide | Sigma-Aldrich | Cat# C7698-1G |
| RNA PureLink columns | Ambion | Cat# 12183018 |
| RNA Clean and Concentrator-5 columns | Zymo Research | Cat# R1016 |
| TURBO DNase | Ambion | Cat# AM2238 |
| Ribonuclease R (RNase R) | Biozym Scientific | Cat# 172010 |
| SUPERase In RNase Inhibitor | Ambion | Cat# AM2696 |
| TRIzol | Invitrogen | Cat# 15596026 |
| AccuPrime Pfx DNA Polymerase | Invitrogen | Cat# 12344024 |
| iScript Reverse Transcription Supermix | Bio-Rad | Cat# 1708840 |
| SsoAdvanced SYBR Green supermix | Bio-Rad | Cat# 1725270 |
| my-Budget 5x EvaGreen QPCR-Mix | Bio-Budget Technologies GmbH | Cat# 80-5820000 |
| cOmplete Mini Protease Inhibitor Cocktail, EDTA-free | Roche | Cat# 11836170001 |
| Dulbecco's Modified Eagle's Medium | Gibco | Cat# 11965–118 |
| Fetal calf serum | Sigma, EMD Millipore | Cat# TMS-013-B |
| Fetal calf serum | Gibco | Cat# ES009-B |
| Bovine Serum Albumin | Sigma | Cat# A9647-100G |
| Trypsin-EDTA (0.5%) | Life Technologies | Cat# 15400-054 |
| Penicillin–Streptomycin | Life Technologies | Cat# 15140163 |
| Glutamin 200 nM | Life Technologies | Cat# 25030081 |
| DPBS, no calcium, no magnesium | Life Technologies | Cat# 14190169 |
| DMEM/F12, no phenol red | Life Technologies | Cat# 21041025 |
| DMEM/F12, with HEPES without phenol red | Life Technologies | Cat# 11039021 |
| Hanks' Balanced Salt Solution (HBSS) | Thermo Fisher | Cat# 14025-076 |
| Opti-MEM | Gibco | Cat# 11058-021 |
| Trypsin, powder | Gibco | Cat# 27250018 |
| Lipofectamine 2000 | Invitrogen | Cat# 11668-019 |
| Lipofectamine 3000 | Invitrogen | Cat# L3000001 |
| Lipofectamine MessengerMAX | Invitrogen | Cat# LMRNA001 |

| Reagent/resource | Reference or source | Identifier or catalog number |
|---|---|---|
| 1× PBS | Gibco | Cat# 14190-250 |
| SYBR Gold | Invitrogen | Cat# S11494 |
| GlycoBlue | Ambion | Cat# LSAM9516 |
| Tris-HCl pH 7.5 | Ambion | Cat# AM9850G, AM9855G |
| NaCl | Ambion | Cat# AM9759 |
| MgCl₂ | Ambion | Cat# AM9530G |
| DTT | Ambion | Cat# 10197777001 |
| glycerol | Sigma-Aldrich | Cat# G5516 |
| Triton X-100 | Sigma-Aldrich | Cat# T8787 |
| Sucrose | Fisher Scientific | Cat# S5-12 |
| Acid-phenol:chloroform, pH 4.5 (with IAA, 125:24:1) | Ambion | Cat# AM9722 |
| Passive Lysis Buffer, 5x | Promega | Cat# E1941 |
| AgeI Fast Digest | Thermo Scientific | Cat# FD1464 |
| BglII Fast Digest | Thermo Scientific | Cat# FD0083 |
| HindIII Fast Digest | Thermo Scientific | Cat# FD0505 |
| EcoRV Fast Digest | Thermo Scientific | Cat# FD0303 |
| Van91I Fast Digest | Thermo Scientific | Cat# FD0714 |
| XbaI Fast Digest | Thermo Scientific | Cat# FD0684 |
| Sodium azide | Carl Roth | Cat# 4221.1 |
| Acid-Phenol Chloroform | Life Technologies | Cat# AM9722 |
| Agarose, standard | Carl Roth | Cat# 3810.3 |
| DNA Gel Loading Dye (6x) | Thermo Scientific | Cat# R0611 |
| UltraPure DNase/ RNase-Free distilled water | Life Technologies | Cat# 10977049, 10977015 |
| NEB 5-alpha competent E. coli (High Efficiency) | NEB | Cat# C2987I |
| RNAscope probes | ACDBio | Custom |
| Nano-Glo Dual-Luciferase Reporter Assay System | Promega | Cat# N1610 |
| Monarch Gel Extraction Kit | NEB | Cat# T1020S |
| Monarch PCR Purification Kit | NEB | Cat# T1030S |
| NEBuilder HiFi DNA Assembly Master Mix | NEB | Cat# E2621S |
| QIAGEN Plasmid Plus Midi Kit (100) | Qiagen | Cat# 12945 |
| SMRTbell prep kit 3.0 | PacBio | Cat# 102-182-700 |
| Zero Blunt TOPO Cloning Kit | Invitrogen | Cat# 450245 |
| **Software** | | |
| minimap2 | Li, 2018 | https://doi.org/10.1093/bioinformatics/bty191 |

| Reagent/resource | Reference or source | Identifier or catalog number |
|---|---|---|
| IsoQuant | Prjibelski et al, 2023 | https://doi.org/10.1038/s41587-022-01565-y |
| IGV | Robinson et al, 2011; Thorvaldsdóttir et al, 2013 | https://igv.org/ |
| FastQC | Babraham Bioinformatics | http://www.bioinformatics.babraham.ac.uk/projects/fastqc/ |
| ESMFold | Lin et al, 2023 | https://esmatlas.com/resources?action=fold |
| ChimeraX | Meng et al, 2023 | https://www.cgl.ucsf.edu/chimerax/ |
| R | R Foundation | https://www.r-project.org/ |
| ImageJ | NIH | https://imagej.nih.gov/ij/ |
| PeakChart software | Brandel | Version v1.02 |
| Volocity Acquisition suite 6.3 | Perkin Elmer | Version 6.3 |
| FlowJo | BD | Version 10.10 |
| Prism | GraphPad Software Inc. | Version 10.0 |
| Adobe Creative Suite 2024 | Adobe | Version 25.2.0; 27.9 |
| **Other** | | |
| DNA LowBind Tubes | Eppendorf | Cat# 10051232 |
| Cell culture plate, 24-well | Sarstedt | Cat# 83.3922.005 |
| Cell culture dish 150, standard | Sarstedt | Cat# 83.3903 |
| 384-well plate, laser marked | Bioplastics BV | Cat# B70515L |
| Opti-Seal Optical Sealing Sheet | BIOplastics | Cat# 157300 |
| 96-well plate black with TC cover | VWR | Cat# 732-2194 |
| RNase-free microcentrifuge pestle | Fisher Scientific | Cat# 12-141-364 |
| QuantStudio5 qPCR machine | Thermo, Applied Biosystems | Cat# 15721248 |
| CFX384 Touch qPCR machine | Bio-Rad | Cat# 1855485 |
| ABI Attune NxT Acoustic Focusing Cytometer | Thermo Scientific | Cat# A24860 |
| FACSAria III Cell Sorter | BD | Cat# N/A |
| UVP GelStudio PLUS AnalyticJena Imager | analytik jena | Cat# 849-97-0943-04 |
| Density Gradient Fraction System | Brandel | Cat# BR-188 |
| GloMax Explorer Multiplate Reader | Promega | Cat# GM3500 |

## Cell culture and transfection

Mouse embryonic mesenchymal C3H/10T1/2 (ATCC: CCL-226) cells or human HEK393T (ATCC: CRL-3216) were cultured in Dulbecco's Modified Eagle's Medium (DMEM, Gibco, 11965–118) containing 2 mM L-glutamine, supplemented with 10% fetal calf serum (Gibco, ES009-B), 100 U/ml penicillin and 0.1 mg/ml streptomycin (EMD Millipore, TMS-AB2-C or Gibco, 15140–122) at 37 °C in 5% $CO_2$-buffered incubators.

For luciferase assays, ~0.6 × 10$^6$ C3H/10T1/2 cells were seeded per well in 12-well dishes and transfected the following day with total 2 µg of plasmid (1 µg Nluc and 1 µg Fluc control plasmid) using 4 µL Lipofectamine 2000 (Invitrogen, 11668-019) and Opti-MEM (Gibco, 11058-021) according to the manufacturer's instructions in serum-free and antibiotic-free DMEM. For transfection with monocistronic Nluc constructs (pGL3-Nluc or pGL3-NLB), 1 µg of a HBB-Fluc control plasmid (pGL3-HBB-Fluc; pKL039) was co-transfected per well. The medium was changed to regular DMEM 4–6 h after transfection and cells were collected 24 h post transfection and samples were split in half for protein and mRNA analysis.

For the mRuby-ZKSCAN-spEGFP plasmid reporter assay and transfection of circRNA plasmid into HEK393T cells using Lipofectamine 3000 (Figs. 2, EV1, and EV3A,D), ~0.08 × 10$^6$ cells were seeded per well in 24-well dishes 16 h before transfection (total volume 500 µl/well). After overnight incubation at 37 °C in 5% $CO_2$ the medium was replaced by 450 µL antibiotic-free DMEM and cells were transfected with 1 µg of plasmid DNA using 1 µL Lipofectamine 3000 combined with 1 µL P3000 (Invitrogen, L3000001) and Opti-MEM (Gibco, 11058-021) according to the manufacturer's instructions in serum-free and antibiotic-free DMEM. The medium was changed to 1 mL regular DMEM 12 h after transfection and cells were collected 24, 48, or 72 h post transfection. In order to obtain further information about the reaction kinetics over time and to detect even weak IRES-like activity, the incubation times were varied between 24, 48, and 72 h. For testing these later time points, the number of seeded cells on day 0 were adapted to 16,000 cells and an additional media change was performed 48 h post transfection for the 72 h test series.

For the mRuby-ZKSCAN-spEGFP plasmid reporter assay and transfection of circRNA plasmid into into HEK393T cells using Lipofectamine 2000 (Fig. 1), ~0.08 × 10$^6$ cells were seeded per well in 24-well dishes 16 h before transfection (total volume 500 µL/well). After overnight incubation at 37 °C in 5% $CO_2$ the medium was replaced by 450 µL antibiotic-free DMEM and cells were transfected with 1 µg of plasmid DNA using 2 µL Lipofectamine 2000 (Invitrogen, 11668-019) and Opti-MEM (Gibco, 11058-021) according to the manufacturer's instructions in serum-free and antibiotic-free DMEM. The medium was changed to 1 mL regular DMEM 12 h after transfection and cells were collected 24, 48, or 72 h post transfection. In order to obtain further information about the reaction kinetics over time and to detect even weak IRES-like activity, the incubation times were varied between 24, 48, and 72 h. For testing these later time points, the number of seeded cells on day 0 were adapted to 40,000 cells, and an additional media change was performed 48 h post transfection for the 72 h test series. For verification of circRNA content of the transfected HEK293T cells, 1.6 × 10$^6$ cells HEK293T cells were seeded 16 h before transfection in 15-cm cell culture dishes. Cells (covered with 8 mL antibiotic-free DMEM) were transfected with

45 µg of plasmid DNA using 180 µL Lipofectamine 2000 (Invitrogen, 11668-019) according to the manufacturer's instructions. Twenty-four hours post transfection, the media was replaced with 25 mL of full cell culture media (DMEM supplemented with 10% FBS, 1% penicillin–streptomycin and 1% glutamine). After 5 days of incubation, cells were harvested.

## Mice

All animal work was performed in accordance with protocols reviewed and approved by Stanford Administrative Panel on Laboratory Animal Care (APLAC). Mice were housed under a 12 h light/dark cycle with free access to food and water. FVB/NJ mice (JAX, 001800) were purchased from the Jackson Laboratory (Bar Harbor, ME, USA) and used as WT. Pregnant females were euthanized at E11.5, the uterus was dissected, and embryos were taken out and placed into 10% fetal bovine serum (FBS) in DMEM/F12 without phenol red (Gibco, 21041025). Embryos were individually micro-dissected to isolate somites and neural tube or limb buds, respectively, as described in (Kondrashov et al, 2011). For RNA extraction and sucrose gradient fractionation, the tissues were dissociated with 1% trypsin (Gibco 27250018) in Hanks' Balanced Salt Solution (HBSS, Thermo Fisher, 14025-076) at 37 °C prior to lysis. For RNAscope, E11.5 WT embryos were sectioned longitudinally as described separately. For PacBio long-read sequencing, E11.5 WT embryos were micro-dissected, and total RNA was extracted from tissues. The biological sex of the embryos was not determined.

## Plasmid construction

The following plasmids have been described previously: mRuby3-ZK-spEGFP and mRuby3-ZK-spEGFP-IRES2(EMCV) (Chen et al, 2021) were kindly provided by Chun-Kan Chen (Stanford University and University of Washington, St. Louis); pGL3-FLB (Fluc, Firefly luciferase reporter genes, driven by the SV40 promoter) and all *Hox* IRES-like element encoding plasmids used as PCR templates (Leppek et al, 2020), pRF-HCV (Yoon et al, 2006) were kindly provided by Davide Ruggero (UCSF, San Francisco, CA, USA) and pcDNA3.1–5'UTR-3xHA-NLuc (pKL401) (Osuna et al, 2017) encoding for a monocistronic HA-tagged nanoluciferase (3xHA-Nluc) followed by a 50 nt-poly(A) tail, fused to a 46 nt scrambled 5' UTR, which allows insertion of 5' UTR sequences between a T7 promoter and 3xHA-Nluc, was kindly provided by Conor J. Howard (UCSF, San Francisco, CA, USA).

For Nluc and NLB luciferase reporter constructs, the following plasmids were used: pGL3-FLB-fusion-HBB (pKL082, this study), encoding for a monocistronic firefly luciferase (Fluc) fused to *human hemoglobin* (*hHBB*) 5' UTR and *β-globin* (exons 1–3, including exon 2, 3 intron) reporter gene fused at its 3' end. To generate the pGL3-NLB-fusion-HBB5' (pMH001) plasmid that encodes a Nluc reporter gene fused to a rabbit *β-globin* reporter gene (intron included) resulting in a Nluc/*β-globin* fusion protein termed NLB, by deleting the stop codon of the Nluc ORF. For this, first pGL3-FLB-fusion-HBB (pKL082) was digested with *EcoRV/Van91I* (Thermo Scientific, FD0303, FD0714), creating a linearized *β-globin* encoding plasmid, with non-overlapping sticky ends excising the Fluc reporter gene. After digestion, the digested backbone plasmid was purified by agarose gel extraction according

to the manufacturer's instructions (NEB, T1020S). Using primers encoding for a β-*globin* 5'-overlap, two Nluc gene fragments were amplified from pcDNA3.1–5'UTR-3xHA-NLuc (pKL401) template and fused by overlap PCR. The *EcoRV*/*Van91I*-digested pKL082 was then ligated with the Nluc insert downstream of the 5' *hHBB* and upstream of the β-*globin* reporter gene via Gibson assembly (NEB, E2621S), generating pGL3-NLB-fusion-HBB5' (pMH001). The second construct is the pGL3-Nluc-HBB5' (pMH002) plasmid for which pKL082 was digested with *EcoRV*/*XbaI* (Thermo Scientific, FD0303, FD0684). The linearized plasmid (excising the Fluc reporter gene) was purified by agarose gel extraction. A Nluc insert was amplified from pKL401 using oligonucleotides with corresponding Gibson-overlap sequences for the *EcoRV*/*XbaI*-digested pKL082 backbone. The Nluc insert was purified via agarose gel extraction. Gibson assembly of the insert with pKL082 generated plasmid pMH002, which contains no β-*globin* reporter gene after Nluc. For generating pGL3-NLB-hHBB3'-HBB5' (pMH023), the hHBB3' UTR was amplified with primers MH008/MH009 and Gibson-inserted into *AgeI*/*XbaI* (Thermo Scientific, FD1464, FD0684) digested pGL3-NLB-fusion-HBB5' (pMH001).

In order to generate the series of promoterless pGL3-NLB plasmids without the SV40 promoter-containing candidate 5' UTRs which were analyzed for their potential promoter-like activity, we generate a SV40Δ version by cutting out the SV40 promoter from pMH023 that contains the *hHBB* 5' UTR by *BglII*/*HindIII*-digest and Gibson assembly using the NEBuilder HiFi DNA Assembly Master Mix (NEB, E2621S) of the annealed oligo 1 LL042/LL043 to seal the ends and generate pLL015. Second, to generate an SV40Δ empty vector, pMH023 is *BglII*/*EcoRV*-digested and followed by Gibson insertion of the annealed oligo 2 LL044/LL045 to seal the ends and generate pLL016. The IRES-like 5' UTR sequences to be tested (including *hHBB*, HCV, EMCV, *Hoxa9*, *Hoxa3* and *Hoxa5*) were PCR-amplified using the AccuPrime Pfx DNA Polymerase (Thermo, Invitrogen, 1876525) and the primers LL046-LL057 and LL68/LL69 cloned into the *BglII*-site of pLL016 to generate SV40Δ-versions of pGL3-NLB-hHBB3' in the series of plasmids pLL015 and pLL017-pLL022. For SV40+ positive control empty vector, pMH023 is digested with *HindIII*/*EcoRV* to cut out the *hHBB* 5' UTR and followed by Gibson insertion of the annealed oligo 3 LL064/LL065 seal the ends and generate pLL026. The IRES-like 5' UTR sequences to be tested in these SV40+ plasmid controls (including HCV (LL58/LL47), EMCV (LL59/LL49), *Hoxa3* (LL61/LL53), *Hoxa5* (LL62/LL55) and *Hoxa9* (LL63/LL57)) were PCR-amplified using the AccuPrime Pfx DNA Polymerase (Thermo, Invitrogen, 1876525) and the indicated primers, and cloned into the *HindIII*/*EcoRV*-site of pLL026 to generate +SV40-versions of pGL3-NLB-hHBB3' in the series of plasmids pLL027, pLL028, pLL030, pLL031, and pLL032, respectively. To generate mutants in the context of the full-length *Hoxa9* IRES-like element, the mutated versions of the sequences (*Hoxa9*-M2/ M5/ M12/ M13) were generated by overlap stitch PCR using the primers LL056/LL057 and LL070-077 and pLL022 as template and inserted using Gibson assembly into the *BglII*/*EcoRV*-digested pMH023 plasmid to generate plasmids pLL036-pLL039.

To generate ZKSCAN-split-EGFP plasmid constructs for circRNA reporter assays, the sequences which were analyzed for their potential IRES-like activity (including *Chrdl1*, *Dlx1*, *Gdf5*, *Sema3a*, *Zfx*, *Bcl2, c-Myc*, *CACNA1A*, *Cofilin*, *Fmr1*, *hHBB*, HCV,

CVB3, *Hoxa9*, P4-native, *Hoxa3* and *Hoxa5*) were amplified by PCR using the AccuPrime Pfx DNA Polymerase (Thermo, Invitrogen, 1876525) with primers LL017-LL028 and PK100-PK119; PK130-PK143; PK175- PK194; PK215-PK224; PK268-PK273 and cloned into the EcoRV-cloning site of the mRuby-ZKSCAN-spEGFP-EcoRV (pKL477) plasmid (see also (Chen et al, 2021)) by Gibson Assembly using the Gibson HiFi DNA Assembly Kit (NEB, E2621S) according to manufacturer's instructions. The complete IRES sequences were tested, only *Bcl2* was truncated by deleting nts 13-365, equivalent to those previously tested (Sherrill et al, 2004). Accordingly, inverse sequence controls of the IRES-like sequence as reverse complement inserts in the mRuby-ZKSCAN-spEGFP-EcoRV (pKL477) plasmid were generated accordingly. The respective plasmids (Appendix Table S1) and primers (Appendix Table S2) are listed in the Supplemental Material. The mutated versions of the tested sample sequences (*Hoxa9*-M2/M5/M12/M13) were generated by PCR amplification using primer LL027/LL028 and LL070-LL077 and cloned into the *EcoRV*-cloning side of the mRuby-ZKSCAN-spEGFP-EcoRV (pKL477) plasmid as described before to generate plasmids pLL040-pLL043.

After transformation of NEB 5-alpha competent *E. coli* (High Efficiency) bacteria (NEB, C2987I) and following Midiprep (QIAGEN Plasmid Plus Midi Kit, 12945), mutations, cloning boundaries and coding sequences in all plasmids were verified by DNA sequencing (Eurofins Genomics or Microsynth). A list of all plasmids and primer sequences used are provided in Appendix Table S1 and Appendix Table S2, respectively. All oligonucleotides were purchased from IDT or Microsynth.

## Mouse embryo whole-mount in situ hybridization

Digoxigenin (DIG)-labeled probes were synthesized, and E11.5 embryos were used for whole-mount in situ hybridization following standard methods (Lufkin, 2007a, 2007b). In brief, DIG-labeled RNA probes were generated by *BamHI*-template digestion of a *Hoxa9* 3' UTR-encoding plasmid (pCR-Blunt-II-TOPO-HoxA9-3'UTR (pKL090); target sequence of 514 nt (position 2078-2591 in *Hoxa9* mRNA isoform A): AGTGAGCCTTTTAGGGGCTCATT-TAAAAAGAGAGCAAGCTAGAAAGAAAAAGAAAGGACTGT CCGTCTCCCTCTGTCTCCTCTCCCCCAAACCCAGCCTCCAC CCGCACAAAGGGGCTCTAAATCCCAGGCCTCATCTCCCCA CTGGCAGTCCGTGCTCAGGCTGGCTCTTAGGCCTGCGGCT TTGATGGAGGAGGTATTGTAAGCTTTCCATTTTATAGAAG GCACACACACACACAAGGGAGGGCATTAGCGCTATTGGCT GTATGTGCTAGCTTGTATATATATATATATATTTAAAAAAA ATCTACCTGCTTCTGACTTTAAGCAAAAGGAAAGAAAACT ACCTTTTTATATAATGCACAACTGTTGATGACTGGCTGTA-TAGTTTTTAGTCTCTGTAGCTAATTTAATTTGCTCTTCGTG TGGCAGATCATTCTGCCAAAATACTTGAACACTGTGTTT-TATTGTGGTAATTATGTTTTGTGACTCAAACTTCTGTGCTG GGTGAAGTA), and run-off T7 in vitro transcription using the DIG RNA labeling mix (Roche, 11277073910). Embryos were fixed in 4% paraformaldehyde, washed twice with PBST, and dehydrated through a methanol series. Embryos were rehydrated through 75%, 50%, 25% methanol in PBST and then washed twice in PBST before treating with 5 μg/mL proteinase K in PBST at room temperature for 20 min for E11.5 stage embryos. Embryos were then washed with PBST, post-fixed in 4% paraformaldehyde and 0.1% glutaraldehyde in PBST at room temperature for 20 min, washed

with PBST, and equilibrated to hybridization mix (50% formamide, 1.3× SSC pH 5.0, 5 mM EDTA, 50 µg/mL yeast RNA (Sigma, R-6625), 0.2% Tween 20, 0.5% CHAPS, 100 µg/mL heparin (Sigma, H-7005)). Embryos in the hybridization mix were incubated at 65 °C in a hybridization oven with rocking for at least 1 h before the addition of 1 µg/mL DIG-labeled RNA probe (Hoxa9 3' UTR probe) and incubated with rocking at 65 °C overnight. The following day, the embryos were washed multiple times (rinse twice, wash 4 × 30 min) at 65 °C with hybridization mix, then a 1:1 mixture of hybridization and MABT (100 mM maleic acid, 150 mM NaCl, 0.1% Tween 20, pH 7.5) for 10 min at 65 °C, and then rinsed twice and washed for 15 min with MABT at room temperature. Blocking was performed by rolling at room temperature with 2% Boehringer Blocking Reagent (Roche, 1096176) in MABT for 1 h followed by rolling with 2% Boehringer Blocking Reagent and 20% heat-inactivated sheep serum in MABT for 1 h. AP-anti-DIG antibody (Roche, 11093274910) at a 1:10,000 dilution in fresh 2% Boehringer Blocking Reagent and 20% heat-inactivated sheep serum in MABT was added to the embryos and incubated by rolling at 4 °C overnight. Embryos were rinsed 3× and washed in MABT 7× for 1 h at room temperature, then washed in MABT overnight at 4 °C. Then, 3 × 20 min washes at room temperature in NTMT (100 mM NaCl, 100 mM Tris-HCl pH 9.5, 50 mM MgCl$_2$, 0.1% Tween 20). Embryos were developed by washing at room temperature with BM Purple (Roche, 11442074001) AP substrate at room temperature in the dark until adequate signal was achieved and then rinsed and washed with PBST, re-fixed in 4% paraformaldehyde and 0.1% glutaraldehyde in PBST for 2 h (room temperature) or overnight (4 °C). Embryos are rinsed once and washed 2 × 10 min with PBST, and stored in PBST with 0.1% azide.

## Mouse embryo smFISH and image analysis

The C57 mouse E11 sagittal frozen sections were directly purchased from Zyagen (Zyagen, MF-104-11-C57). The mouse E11.5 embryos were collected from WT FVB/NJ mice, and processed by Histo-Tec Laboratory Inc. to obtain the transverse frozen sections. The RNAscope probes were custom-designed and purchased from ACDBio. The probe targeting the Hoxa9 mRNA CDS region is termed Mm-Hoxa9-O6 and targets the 1267–1892 region of NM_010456.3 excluding the short isoform-specific intron region 1581–1753. The probe targeting the IRES-like region is termed Mm-Hoxa9-O5 and targets the 776–1142 region of NM_010456.3. Mm-Hoxa9-O5 and Mm-Hoxa9-O6 probes were directly ordered from ACDBio. The exact probe sequences are proprietary and not available to us. smFISH was conducted according to the manufacturer's instructions. Confocal imaging was conducted as previously described (Zhang et al, 2021). Briefly, images were acquired with a custom-built inverted spinning disk microscope. A Zeiss Axio Observer Z1 microscope was coupled to the Perkin Elmer UltraVIEW Vox spinning disk confocal microscopy system with an encoded ASI MS2000 motorized piezo stage equipped with Plan-Apochromat 10x/0.45 and 63x/1.20 objectives (Carl Zeiss microscopy). The 405-nm, 488-nm and 561-nm solid-state laser lines were paired with the 452W25, 510W20, and 588W21 Semrock emission filters to minimize crosstalk between different fluorophores. Images were acquired with the Volocity Acquisition suite 6.3 (Perkin Elmer). All acquired high-resolution images of the posterior somites (somite number 20–28) and neural tube (the boxed region as depicted in the main figure) were

analyzed using custom Python scripts. In brief, local fluorescence intensity thresholding was applied to call individual fluorescence puncta. Then for each punctum from the CDS-probe channel, if it partially overlaps with a punctum from the IRES-probe channel, it is classified as a long 5' UTR Hoxa9 mRNA isoform; otherwise it is classified as a short 5' UTR isoform. The puncta counts from multiple images in the same embryo were then pooled together and used to quantify the respective isoform ratio. All acquired high-resolution images of the posterior somites (somite number 20–28) and neural tube (the boxed region as depicted in Fig. 3) were analyzed using custom Python scripts.

## PacBio long-read sequencing, library generation, and analysis

Neural tubes/somites were micro-dissected from E11.5 FVB/NJ mouse embryos. The uterus was washed in 1×PBS twice to remove blood. Each embryo was transferred into a plate containing filming media one by one, where microdissection was performed to take out the tissues. Filming media (10% FBS (Sigma-Aldrich, TMS-013-B) in DMEM/F12 with HEPES without phenol red (Gibco, 11039021)) was used during microdissection. Excess media was removed with a pipette without centrifugation, and the sample was then flash-frozen in liquid nitrogen. Tissue was then rapidly ground by hand for 30 s on ice with a disposable RNase-free microcentrifuge pestle (Fisher Scientific, 12-141-364). Immediately after, 1 mL of TRIzol reagent was added (Thermo Fisher, 15596026). The sample was lysed and mixed by vigorously inverting the tube several times, and then had the total RNA extracted following the manufacturer's protocol, using a cleanup column (Zymo Research, R1013) instead of ethanol precipitation. Subsequently, the RNA samples were treated with TurboDNase (Thermo Fisher, AM2238) at 37 °C for 30 min according to the manufacturer protocol and cleaned up using a column.

The Iso-seq library was prepared using SMRTbell prep kit 3.0 (PacBio, 102-182-700) following the kit's instructions with some added details below. For the section "Purification of amplified cDNA with SMRTbell cleanup beads", 84 µL of SMRTbell cleanup beads were added to the sample. Prepared library was then sequenced with PacBio Sequel II by Genome Sequencing Service Center at Stanford University, and also processed to obtain HiFi reads. Reference genome and comprehensive gene annotation on the reference chromosomes, scaffolds, assembly patches, and haplotypes were obtained from GENCODE release M32 (GRCm39) (Frankish et al, 2021). HiFi reads were aligned to the genome using minimap2 (Li, 2018) with default settings. Isoform discovery was performed using IsoQuant (Prjibelski et al, 2023) with default settings as follows: isoquant.py --reference /path/to/GRCm39.genome.fa --genedb /path/to/gencode.vM32.chr_patch_hapl_scaff.annotation.gtf --complete_genedb --bam /path/to/hifi_reads.bam --data_type pacbio_ccs --output /path/to/output. Transcript model covering Hoxa9 and Hoxa10 (chr6:52,200,050-52,217,850) was visualized using IGV (Robinson et al, 2011; Thorvaldsdóttir et al, 2013) (https://igv.org/).

## Mouse embryo RNA isolation and RT-qPCR

All mouse work was reviewed and approved by the Stanford Administrative Panel on Laboratory Animal Care (APLAC). FVB/

NJ mice (JAX, 001800) were housed in standard conditions with 12 h light/dark cycles, ambient temperatures between 20 and 26 °C, and humidity between 30 and 70%. To generate embryos, a male was housed with one or two females and the females monitored for vaginal plug formation. On the day a plug was observed, the female was considered to be pregnant at E0.5. Embryos were harvested from the pregnant female at E11.5, and the limb buds and neural tube/somites were dissected in filming media (10% fetal bovine serum (FBS) in DMEM/F12, no phenol red, 21041025). Each tissue was dissociated using 1% trypsin (Gibco 27250018) in Hanks' Balanced Salt Solution (HBSS, Thermo Fisher, 14025-076) at 37 °C and the trypsin was neutralized with filming media. Cells were washed with 1×PBS and then RNA was extracted with TRIzol (Thermo Fisher, 15596026) following the manufacturer's instructions. The RNA was treated with TURBO DNase (Ambion, AM2696) at 37 °C for 30 min to remove genomic DNA and purified using the Purelink RNA Mini Kit (Ambion, 12183018) following the manufacturer's instructions. cDNA synthesis was performed using 1 μg RNA with the Script Reverse Transcription Supermix kit (Bio-Rad, 1708841) according to the manufacturer's instructions. The cDNA was diluted tenfold and 4 μL used as a template for qPCR with SsoAdvanced SYBR Green supermix (Bio-Rad, 1725270) on a CFX384 machine (Bio-Rad).

In order to perform absolute quantification, a DNA standard template was made by cloning the *Hoxa9* IRES-CDS from E11.5 neural tube and somite cDNA using primers KL596/KL110. The PCR product was cloned into the pCR4 TOPO vector using the Zero Blunt TOPO Cloning Kit (Invitrogen, 450245) following kit instructions and sequenced to confirm that the plasmid insert was from the mature spliced Hoxa9 transcript sequence. A standard curve was made for each amplicon by performing qPCR on tenfold serial dilutions (1 pg to 0.1 fg) and used to determine the starting quantity of each amplicon in the embryonic cDNA samples. All primer sequences are listed in Appendix Table S2.

### Mouse embryo sucrose gradient fractionation analysis and RT-qPCR

For sucrose gradient fractionation of lysates of micro-dissected mouse embryo tissues from mouse tissues, E11.5 mouse embryo neural tube and somites were dissected and dissociated with 1% trypsin (Gibco 27250018) and 100 μg/mL cycloheximide (CHX) (Sigma-Aldrich, C7698-1G) in HBSS at 37 °C. After neutralization with cold filming media with 100 μg/mL cycloheximide, the cells were washed with cold 1xPBS with 100 μg/mL cycloheximide and lysed in lysis buffer (20 mM Tris-HCl pH 7.5 (Ambion, AM9850G, and Ambion, AM9855G), 150 mM NaCl (Ambion, AM9759), 15 mM MgCl$_2$ (Ambion, AM9530G), 1 mM DTT (Ambion, 10197777001), 8% glycerol (Sigma-Aldrich, G5516), 1% Triton X-100 (Sigma-Aldrich, T8787), 100 μg/ml Cycloheximide (CHX) (Sigma-Aldrich, C7698-1G), 20 U/ml TURBO DNase (Ambion, AM2238), and Complete Protease Inhibitor EDTA-free (Sigma-Aldrich, 11836170001) in nuclease-free water (Thermo Fisher Scientific, 10977015)) at 4 °C for 30 min on a rotator at 4 °C with occasional vortexing, followed by sequential centrifugation at 1800 × $g$ for 5 min at 4 °C and then 10,000×$g$ for 5 min at 4 °C. The clarified cytoplasmic extract was loaded onto 25–50% sucrose (Fisher Scientific, S5-12) (w/v) gradients (20 mM Tris pH 7.5, 100 mM NaCl, 15 mM MgCl$_2$, 100 μg/ml cycloheximide), made by

sequentially freezing 50%, 43.75%, 37.5%, 31.25%, and 25% sucrose. Gradients were centrifuged in a Beckman SW60 rotor at 35,000 rpm for 2.5 h at 4 °C in a Beckman L8-80M ultracentrifuge and then fractionated on a Density Gradient Fraction System (Brandel, BR-188) using PeakChart software (v1.02) with continuous A$_{260}$ measurements. To each fraction, we added 100 pg of in vitro transcribed Renilla luciferase RNA as a spike-in control and detected it using Rluc-specific primers. RNA was extracted using Acid-phenol:chloroform, pH 4.5 (with IAA, 125:24:1) (Ambion, AM9722), incubating for 5 min at 65 °C followed by centrifugation at 21,000×$g$ for 10 min. The aqueous phase was mixed 1:1 with ethanol and RNA isolated using the RNA Clean and Concentrator-5 Kit (Zymo Research, R1016). The RNA was treated with TURBO DNase (Ambion, AM2238) for 30 min at 37 °C and purified with the RNA Clean and Concentrator-5 Kit (Zymo Research, R1016). cDNA synthesis was performed using the iScript Reverse Transcription Supermix kit (Bio-Rad, 1708841) with 200 ng of RNA, diluted 20-fold, and 4 μl used as a template for qPCR with SsoAdvanced SYBR Green supermix (Bio-Rad, 1725270) on a CFX384 machine (Bio-rad). All primer sequences are listed in Appendix Table S2. The Ct value of each amplicon from each fraction was first normalized to the Ct value of the luciferase RNA spike-in, converted from log2 to linear values, and normalized to the total abundance of the mRNA across all fractions.

### ESMFold and fusion protein structure prediction

ESMFold, as implemented in the ESM Metagenomic Atlas (https://esmatlas.com/resources?action=fold) was used to predict the structure of the NLB reporter, given the low depth of co-evolutionary information available for the Nanoluc domain (Lin et al, 2023). Molecular graphics created and analysis performed with UCSF ChimeraX (Meng et al, 2023) (https://www.cgl.ucsf.edu/chimerax/).

### Luciferase activity assay after plasmid transfection

Mouse C3H/10T1/2 cells in 12-well plates were transiently transfected (Lipofectamine 2000, Invitrogen, 11668-019) with 1 μg NLB and 1 μg *hHBB*-Fluc control plasmid. Cells were washed with 1× PBS (Gibco, 14190-250) and collected by trypsinization 24 h post transfection for Nluc/Fluc luciferase activity assays. 60% the cells were used for assaying luciferase activity using the Nano-Glo Dual-Luciferase Reporter Assay System (Promega, N1610) to measure Nanoluc (Nluc) and Firefly (Fluc) luciferase activities, the rest 40% was collected in TRIzol (Invitrogen, 15596) for total RNA purification with the RNA Clean and Concentrator-5 Kit (Zymo Research, R1016), TURBO DNase (Ambion, AM2238) digest, repeat column purification, and detection of relative mRNA levels by RT-qPCR (see RT-qPCR section). For luciferase assays, cells were lysed in 50 μl of 1× passive lysis buffer (Promega, E1941) and directly assayed or frozen at −20 °C. After thawing, cell debris and nuclei were removed by centrifugation for 1 min at 13,000 rpm. In total, 25 μl of supernatant was assayed in a 96-well black-non-transparent plate (Falcon, 732-2194) for luciferase activity by mixing with 35 μl of Nano-Glo Dual-Luciferase Reporter Assay System substrates (Promega, N1610, concentrated substrates for promoterless assays and 1:10 dilutions in water for all other assays) using manual multi-channel pipetting. Fluc and Nluc activities were measured on a GloMax Explorer Microplate Reader

(Promega). Luciferase reporter activity is expressed as a ratio between Nluc and Fluc (Nluc/Fluc) which was compared to respective normalized mRNA levels. For monocistronic Nluc-b-*globin*-fusion constructs, Nluc/Fluc luciferase activity was normalized to β-*globin*/*Nupl1* or β-*globin*/*actin* mRNA levels to quantify variation in mRNA expression. Each experiment was performed a minimum of three independent times.

## Quantitative RT-PCR (RT-qPCR) analysis

Cells transfected with pGL3-NLB-fusion constructs were collected in 500 μL TRIzol (Invitrogen, 15596). Total RNA was isolated from the aqueous phase using Zymo Clean & Concentrator-5 column (Zymo Research, R1016) and treated with TURBO DNase (Ambion, AM2238) followed by a second Zymo column purification step. For reverse transcription-quantitative PCR (RT-qPCR) analysis, cDNA was synthesized from 100 to 200 ng of total RNA using iScript Supermix (Bio-Rad, 1708840) containing random hexamer primers, according to the manufacturer's instructions. PCR reactions were assembled in 384-well plates using 2.5 μL of a 1:4–1:5 dilution of a cDNA reaction, 300 nM of target-specific primer mix and the SsoAdvanced SYBR Green supermix (Bio-Rad, 1725270) or 5× EvaGreen qPCR-Mix (Bio-Budget, 80-5820000) in a final volume of 10 μl per well. SYBR green detection qPCR was performed on a CFX384 machine (Bio-Rad) or QuantStudio qPCR machine (Thermo, applied biosystems, 15721248). Data were analyzed and converted to relative RNA quantity manually or using CFX manager (Bio-Rad). Gene-specific qPCR primer sequences used for the detection of mRNAs are given in Appendix Table S2.

## circRNA plasmid-based EGFP reporter assay

Cells transiently transfected with mRuby-ZKSCAN-spEGFP plasmids were harvested 24, 48, or 72 h post transfection. After resuspension, cells were transferred into 1.5 ml Eppendorf tubes. Cells were washed with 500 μL 1xDPBS and finally resuspended in 180 μL FACS buffer (1×PBS, 1% BSA, 0.1% NaN₃) for the 24 h and 48 h test series and in 230 μL for the 72 h test series. The mRuby and EGFP signal intensity was then measured by using the ABI Attune Acoustic Focusing Cytometer (Thermo Scientific). FACS data were exported as FCS files and further analyzed by FlowJo v10.8 (BD). A visualization of the gating scheme is shown in Fig. EV1A. Low-quality data derived from cells measured after a clogging event of the FACS machine were initially excluded during quality control. Afterward, viable cells and singlets were identified based on the FSC and SSC channel. Finally, transfected cells were identified based on their mRuby signal, and the EGFP signal of this positive subfraction was measured and used for the calculation of the median fluorescence intensities (MFIs).

## Quantification and analysis of circRNA content by RT-qPCR

After 5 days of incubation, cells were harvested and transferred into 15-mL falcon tubes. After a quick centrifugation (500× g, 5 min at RT), media was replaced with 6 mL of full cell culture media. In order to reduce the noise level of untransfected cells during the following assay, cells were FACS-sorted by the Flow Cytometry

Core Facility of the Medical Faculty at the University of Bonn based on their mRuby and EGFP signal intensity. TRIzol-based total RNA extraction (Invitrogen, 15596026) was then performed on the double positive sorted cell fraction and purified using the RNA Clean & Concentrator −5 kit (Zymo, R1016). The remaining DNA contamination was removed by TURBO DNase treatment for 30 min at 37 °C (Thermo, AM2238) and second purification using the same Clean & Concentrator −5 column as beforehand. The obtained RNA was split into two and incubated with (positive sample) or without (negative control) RNase R (Biozym Scientific, B3539-172010) at a working concentration 1 U/μg RNA in presence of RNase R reaction buffer for 30 min at 37 °C and 500 rpm shaking. After subsequent RNA purification using the PureLink RNA Mini Kit (Thermo, 12183018 A), equal amounts of 200–500 ng RNA per construct were used for initial cDNA synthesis using the iScript RT supermix (Bio-rad, 1708841), according to manufacturer's instructions, and diluted 1:5 before final circRNA quantification by qPCR (my-Budget 5x EvaGreen qPCR-Mix II) with 300 nM of target-specific primer mix in a final volume of 10 μl. A schematic overview of the experimental procedure is shown in Fig. EV3B.

Raw qPCR data were transformed $(2^{-1*(dCt)})$, normalized to the housekeeping mRNA *Nupl1*, and linear and circular RNA content was visualized after RNase R digestion relative to the undigested control samples. In addition, the ratio of circRNA enrichment over RNase R digestion relative to the linear RNA enrichment over RNase R digestion was calculated for comparison between the different tested constructs.

## Data sources

For the PacBio long-read sequencing data analysis, the following transcript IDs were retrieved for the mouse transcriptome (*Mus musculus*) from the ENSEMBL database as data sources and aligned as described in the PacBio "Methods" with the following accession numbers: mouse *Hoxa9* 85 nt short 5' UTR mRNA isoform (ENSMUST00000114425), mouse *Hoxa9* 1266 nt-long 5' UTR mRNA isoform (ENSMUST00000048680), mouse *Hoxa10* mRNA isoforms (ENSMUST00000125581.2 hoxa10-202, ENSMUST00000121043.2 hoxa10-201, ENSMUST00000142611.2 hoxa10-204). For NLB fusion protein prediction we used the following X-ray crystal structures from PDB: 1.95 Å resolution structure of shrimp NanoLuc luciferase (*Oplophorus gracilirostris*, 5IBO), 2.0 Å resolution structure of rabbit *hemoglobin* (*Oryctolagus cuniculus*, 2RAO).

## Quantification and statistical analysis

In all figures, data were presented as mean, SD or SEM as stated in the figure legends, and $*P \leq 0.05$ was considered significant (ns: $P > 0.05$; $*P \leq 0.05$; $**P \leq 0.01$; $***P \leq 0.001$; $****P \leq 0.0001$). Blinding and randomization were not used in any of the experiments. The number of independent biological replicates used for the experiments are listed in the figure legends. Tests, two-tailed unpaired Student's *t* test if not stated otherwise, and specific *P* values used are indicated in the figure legends or figures themselves. In all cases, multiple independent experiments were performed on different days to verify the reproducibility of experimental findings. For mouse experiments, embryos from multiple litters were used to avoid litter-specific bias.

## Materials availability

All plasmids, resources, and reagents generated in this study are available upon request and will be fulfilled by the lead contacts, Maria Barna (mbarna@stanford.edu) and Kathrin Leppek (kleppek@uni-bonn.de).

# Data availability

RNA sequencing data from PacBio RNA-seq experiments are available in Dataset EV1 and were deposited as long-read RNA-seq data in the Gene Expression Omnibus (GEO) under accession number GSE250214. Source images from smFISH experiments were deposited at BioImage Archive under accession number S-BIAD1665. Custom Python scripts for smFISH image analysis can be found on the Barna lab github (https://github.com/barnalab).

The source data of this paper are collected in the following database record: biostudies:S-SCDT-10_1038-S44318-025-00404-5.

# Peer review information

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

## Acknowledgements

The authors would like to thank the Barna and Leppek lab members for support and constructive criticism of the work. The authors thank Katrin Reiners (IKCKP, University Hospital Bonn, University of Bonn) for technical advice with the Attune FACS, and Alina Niedzwetzki and Lisa Nauroth for technical assistance in the Leppek lab. The authors would like to thank Chun-Kan Chen (University of Washington, St. Louis, USA) for kindly sharing circRNA plasmids and for technical advice, Kotaro Fujii (Stanford University and University of Florida, Gainesville, USA) for kindly providing plasmid pCR-Blunt-II-TOPO-HoxA9-3′UTR; and Andrey Prjibelski (University of Helsinki, Helsinki, Finland) and Angela Brooks (UC Santa Cruz, USA) for bioinformatic advice. We would like to thank Howard Y. Chang (Stanford University), Anne Brunet (Stanford University), and Judith Frydmann (Stanford University) for helpful advice and support. We would like to thank the Flow Cytometry Core Facility of the Medical Faculty at the University of Bonn, Germany, for providing help, services, and devices funded by the Deutsche Forschungsgemeinschaft (DFG, German Research Foundation)—Project Number 216372545, and the Genome Sequencing Service Center at Stanford University for PacBio sequencing services. Molecular graphics and analyses were performed with UCSF ChimeraX, developed by the Resource for Biocomputing, Visualization, and Informatics at the University of California, San Francisco, with support from National Institutes of Health (NIH) R01-GM129325 and the Office of Cyber Infrastructure and Computational Biology, National Institute of Allergy and Infectious Diseases (NIAID). This work was supported by NIH grant 5R01HD086634 (MB), Rosetta Commons Mini Grant RC22025 (AK), NIH P50 training grant (ZZ), National Science Scholarship (Ph.D.) from the Agency for Science, Technology and Research (A*STAR) (TTS), National Science Foundation (NSF) Graduate Research Fellowship DGE-114747 (NRG), Ph.D. position funded by the Cluster of Excellence ImmunoSensation2 (MH); and by the TRA Life and Health Research Prize 2024 of the University of Bonn (KL). KL is supported by the Cluster of Excellence ImmunoSensation2 funded by the DFG under Germany's Excellence Strategy—EXC2151—390873048, the Strengthening the Equal Opportunity Process (STEP) Program of the University of Bonn, and start-up funds of the Medical Faculty at the University Hospital Bonn and University of Bonn, Germany.

## Author contributions

**Philipp Koch**: Data curation; Formal analysis; Validation; Visualization; Writing—original draft; Writing—review and editing; Designed the experiments, performed experiments, optimized, performed and analyzed circRNA and FACS-based experiments. **Zijian Zhang**: Data curation; Software; Formal analysis; Validation; Visualization; Writing—original draft; Writing—review and editing; Designed the experiments, performed experiments, performed and analyzed smFISH experiments. **Naomi R Genuth**: Data curation; Formal analysis; Validation; Visualization; Writing—original draft; Writing—review and editing; Designed the experiments, performed experiments, performed and analyzed absolute qPCR quantification and polysome-qPCR experiments. **Teodorus Theo Susanto**: Data curation; Software; Formal analysis; Visualization; Writing—original draft; Writing

—review and editing; Designed the experiments, performed experiments, performed and analyzed PacBio long-read sequencing experiments, analyzed sequencing and qPCR data. **Martin Haimann**: Data curation; Formal analysis; Designed the experiments, performed experiments, generated constructs, performed and analyzed NLB reporter and promoterless reporter experiments. **Alena Khmelinskaia**: Resources; Data curation; Formal analysis; Funding acquisition; Visualization; Writing—original draft; Designed the experiments, performed experiments, performed and analyzed fusion protein structure predictions. **Gun Woo Byeon**: Formal analysis; Writing—original draft; Designed the experiments, analyzed sequencing and qPCR data. **Saurabh Dey**: Data curation; Formal analysis; Performed experiments. **Maria Barna**: Conceptualization; Resources; Supervision; Funding acquisition; Writing—original draft; Project administration; Writing—review and editing; Designed the experiments, performed experiments, performed embryo micro-dissections and sectioning. **Kathrin Leppek**: Conceptualization; Resources; Data curation; Formal analysis; Supervision; Funding acquisition; Visualization; Writing—original draft; Project administration; Writing—review and editing; Designed the experiments, performed experiments, performed embryo whole-mount ISH, generated constructs, performed and analyzed NLB reporter and promoterless reporter experiments.

Source data underlying figure panels in this paper may have individual authorship assigned. Where available, figure panel/source data authorship is listed in the following database record: biostudies:S-SCDT-10_1038-S44318-025-00404-5.

## Funding

## Disclosure and competing interests statement

KL and MB are inventors of patents related to the *Hoxa9* P4 stem-loop and RNA therapeutics and their various uses. PK, MB, and KL are inventors on a patent related to IRES-like elements in circRNA reporters and RNA therapeutics and their various uses. The remaining authors declare no competing interests.

# Expanded View Figures

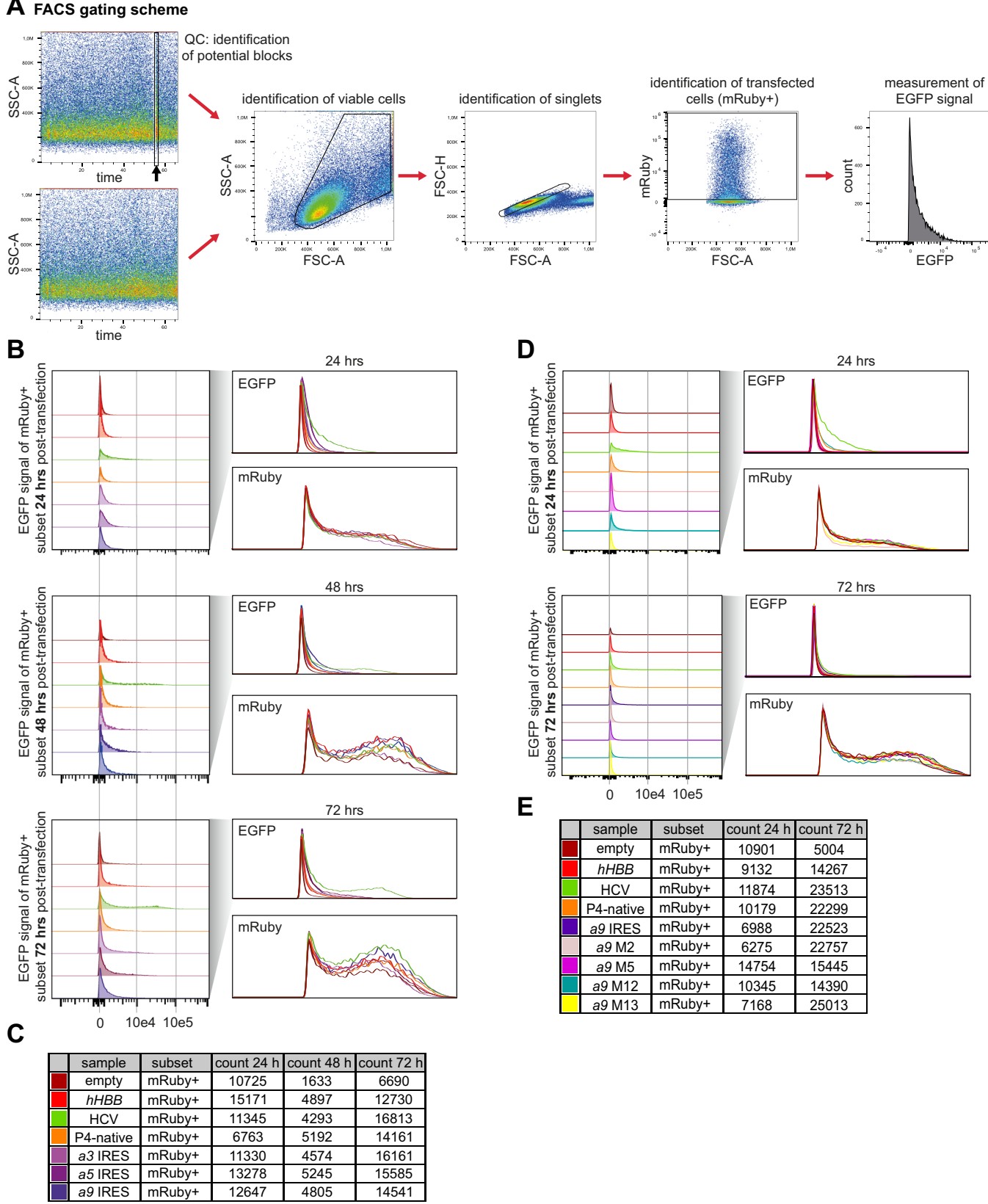

◀ **Figure EV1. FACS Gating Scheme for circRNA plasmid-derived EGFP assays.**

(A) Schematic overview of the raw FACS data processing pipeline including: initial quality control (to exclude low-quality data derived from cells measured after a clogging event of the FACS machine), identification of viable cells and singlets, identification of transfected cells (based on the positive mRuby signal), and final measurement of the EGFP signal of the mRuby+ subfraction. (B) EGFP signals of the mRuby+ subfraction of cells representatively shown as histograms for one experiment for each observed time point (left panels) and overlayed histograms of the EGFP and mRuby signal of the mRuby+ subfraction of cells (right panels), colorized according to the tested sample sequences according to Fig. EV1C. (C) Cell numbers of the mRuby+ cell fractions of the indicated samples shown for one representative experiment per time point. (D) EGFP signals of the mRuby+ subfraction of cells representatively shown as histograms for one experiment for each observed time point (left panels) and overlayed histograms of the EGFP and mRuby signal of the mRuby+ subfraction of cells (right panels), colorized according to the tested sample sequences according to Fig. EV1E. (E) Cell numbers of the mRuby+ cell fractions of the indicated samples shown for one representative experiment per time point.

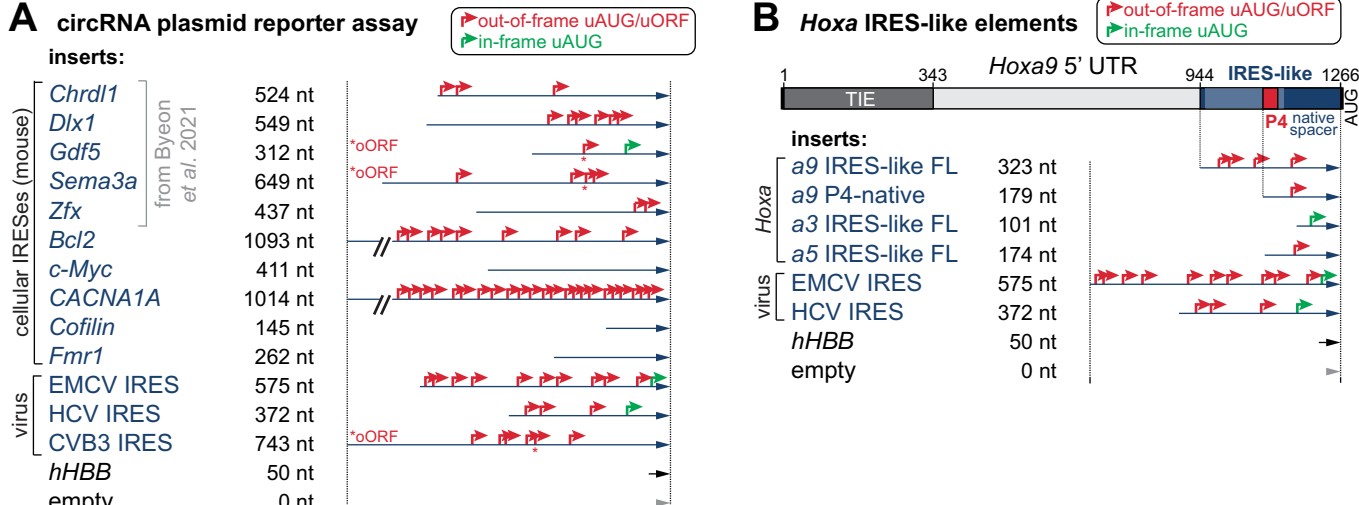

**Figure EV2. Schematic of IRES-like elements for circRNA plasmid-derived EGFP assays and annotated uAUGs.**

(A) Schematic of the circRNA reporter assay based on the mRuby-ZKSCAN-split-EGFP plasmid for the screening of IRES-like activity of the different tested insert sequences, with annotated out-of-frame upstream (u)AUGs/uORFs (red) and overlapping ORFs (oORF, asterisks), and in-frame uAUGs (green) mapped onto the IRES-like sequences (adapted from Fig. 1A). (B) Schematic of different tested insert sequences as in (A) for the *Hoxa* cluster IRES-like elements, with annotated out-of-frame uAUGs/uORFs (red), and in-frame uAUGs (green) mapped onto the IRES-like sequences (adapted from Fig. 2A).

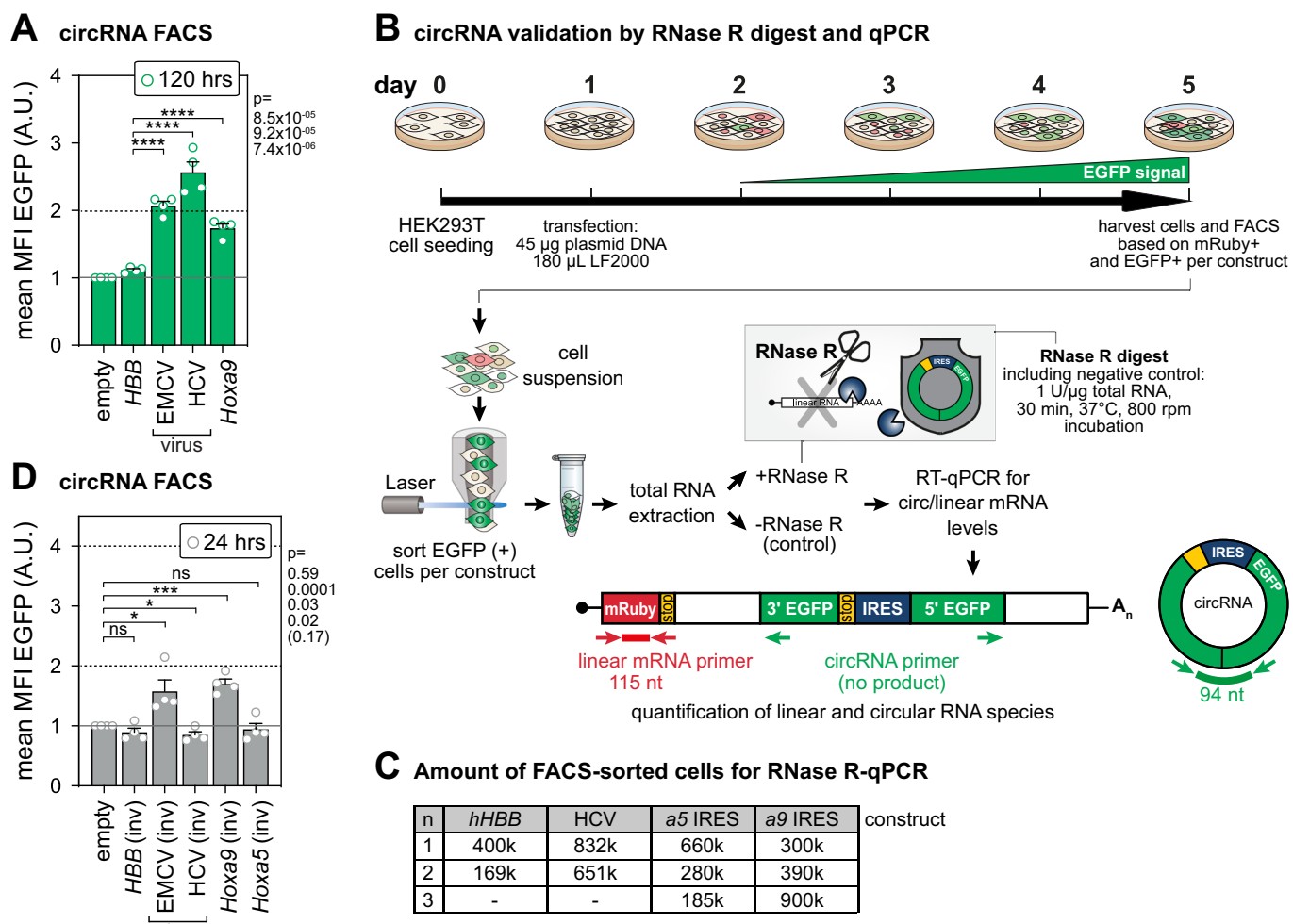

**Figure EV3.  circRNA detection after plasmid transfection, RNase R cleanup and RT-qPCR.**

(**A**) Calculated median fluorescence intensities (MFIs) of the mRuby+ subfractions are shown after normalization to the empty vector control in dependency of the tested insert sequences 120 h post transfection. Bar graphs are indicating mean values ± SEM, $n = 4$. See also Fig. 2B. (**B**) Experimental outline to proof the circRNA content of the HEK293T cells, generated by spliceosome-mediated backsplicing, after plasmid DNA transfection in order to validate the origin of the observed EGFP signal. Transfected cells were harvested 5 days post transfection and FACS-sorted according to their mRuby/EGFP signal. Afterwards, total RNA extraction was performed on the double positive cell fraction and subsequently digested with RNase R (1 U/µg RNA) for 30 min at 37 °C. The negative control was incubated with RNase R reaction buffer. qPCR was used for final circRNA quantification. EGFP primer will only lead to a product of 94 nt length after successful backsplicing. The 115 nt-long mRuby product was used for linear pre-mRNA quantification. The enrichment of circRNA over the RNase R digestion is shown in Fig. 2D. (**C**) Cell numbers of the EGFP+ cell fractions of the indicated samples shown for the $n = 2$–3 experiments used for RNase R-qPCR. (**D**) Calculated median fluorescence intensities (MFIs) of the mRuby+ subfractions are shown after normalization to the empty vector control in dependency of the tested insert sequences 24 h post transfection. Bar graphs are indicating mean values ± SEM, $n = 4$. See also Fig. 2E. Source data are available online for this figure.

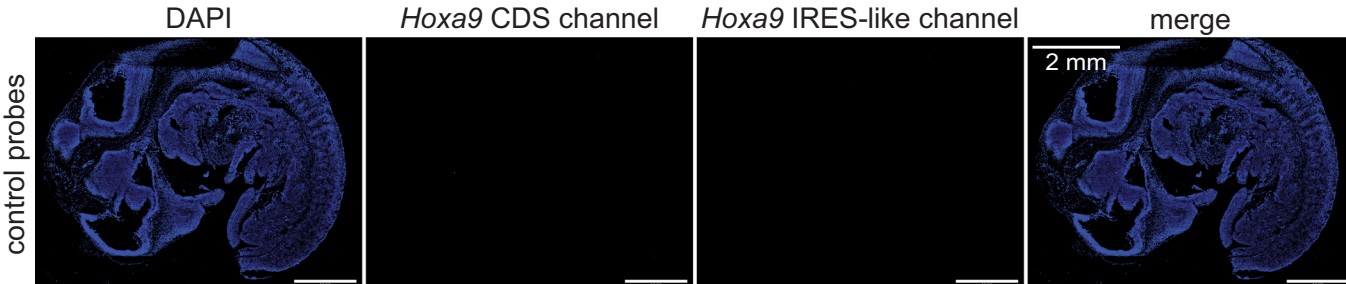

**Figure EV4. Low background is observed for smFISH with control probes.**

Representative images of the E11 mouse embryo sagittal sections stained with commercially available RNAscope Negative Control Probes targeting the *DapB* gene. Low background is observed in the channels used for visualizing *Hoxa9* CDS and IRES-like regions. This control probe staining was performed in parallel to the specific probe staining Fig. 3B in the same experiment but Fig. 3B and the panel shown here represent different embryo sections.

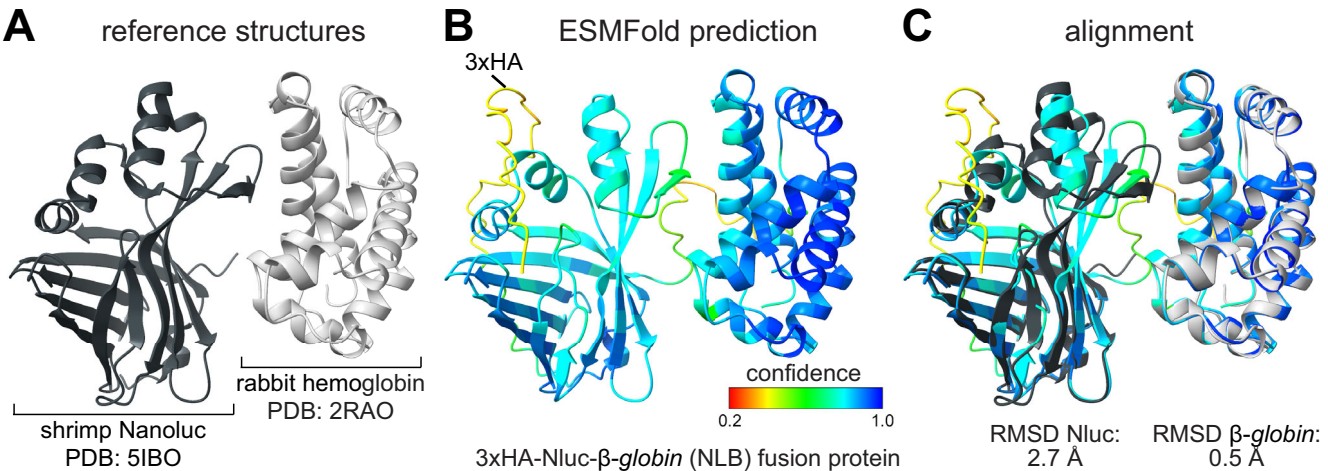

**Figure EV5. ESMFold structure prediction of the NLB reporter fusion protein.**

(A) For comparison, the crystal structure of shrimp Nanoluc (Nluc, PDB: 5IBO) and chain B of rabbit *hemoglobin* (PDB: 2RAO) are shown, in dark and light gray, respectively. Nluc and rabbit *hemoglobin* fold into a β-barrel and a globular fold, respectively, of similar size. (B) ESMFold, a large language model for protein structure prediction with evolutionary information (Lin et al, 2023), yields a high confidence prediction of the structure of the designed NLB reporter (average per-residue model confidence score plddt of 0.78). Structure prediction of the fusion protein NLB with the flexible 3xHA tag and interdomain linker reveals that both Nluc and β-*globin* can adopt their native folds, with a less compact thus less stable fold of the Nluc β-barrel, while the flexible linker and N-terminal 3xHA tag are unstructured. Predicted structure is colored by the model confidence. (C) Crystal structures aligned to the respective domains in the NLB reporter and the root-mean-square deviation (RMSD) for each of them are provided. The alignment of the predicted NLB fusion protein with the individual crystal structures reveals that in the fusion protein, the β-barrel of the Nluc is less compact in the fusion protein, suggesting that the fold is less stable within the proposed construct. Given that the luciferase activity of Nluc is tightly linked to substrate oxidation in its central cavity for luminescence (Tomabechi et al, 2016), this effect on the native Nluc folding state may explain the reduced activity of NLB.

