## [Peer Review File · The EMBO Journal]

A versatile toolbox for determining IRES activity in cells and embryonic tissues

Philipp Koch, Zijian Zhang, Naomi Genuth, Teodorus Susanto, Martin Haimann, Alena Khmelinskaia, Gun Byeon, Saurabh Dey, Maria Barna, and Kathrin Leppek

Corresponding author(s): Kathrin Leppek (kleppek@uni-bonn.de) , Maria Barna (mbarna@stanford.edu)

Review Timeline:

Submission Date:	26th May 24
Editorial Decision:	15th Jul 24
Appeal Received:	20th Nov 24
Editorial Decision:	17th Jan 25
Revision Received:	26th Jan 25
Accepted:	18th Feb 25

Editors: Kelly M Anderson and Cornelius Schneider

Transaction Report:

Dear Prof. Leppek,

Thank you for submitting your manuscript for consideration by the EMBO Journal. It has now been seen by three referees whose comments are shown below. Given these opinions and the fact that the EMBO Journal can only afford to accept papers which receive enthusiastic support from a majority of referees, I am afraid we cannot offer to publish it here.

Thank you in any case for the opportunity to consider this manuscript. I am sorry we cannot be more positive on this occasion, but we hope nevertheless that you will find our referees' comments helpful.

Yours sincerely,

Kelly M Anderson, PhD
Editor, The EMBO Journal
k.anderson@embojournal.org

Referee #1:

In the manuscript Koch et al describe a holistic approach, using both standard and novel techniques, to robustly interrogate IRES-like function in mammalian cells. The Hoxa9 5'UTR is used as an example. IRES-like activity in 5' UTRs of cellular mRNAs was described over 25 years ago with work in this area carried out in the Sarnow, Prats and Willis laboratories. However there has been controversy in the translation field in that there has been some dispute about existence of cellular IRESs, albeit from a fairly small number of individuals. In particular, the over-reliance of discistrionic reporter vector systems and the use of transient transfections has resulted in a number of sequences being attributed with the ability to directly recruit the ribosome in a cap-independent manner, similar to viral IRESs, when in fact this has actually resulted from splicing or cryptic promoter activity. The ability of certain 5'UTRs within cellular RNAs to recruit ribosomes using a mechanism that is distinct from cap-dependent scanning represents an important research area within the translational control field and there are many independent methods, including the work from the Thompson laboratory and others, which show that IRES-like activity exists in mammalian 5' UTRs, including an unbiased screen by Weingarten-Gabbay which suggests ~10% of mRNAs contain IRES. The authors make the very important and well considered point that there are complex networks within mammalian cells for gene regulation and it is unsurprising that a mRNA can be regulated by many parallel mechanisms, e.g. promoters may be present in the DNA sequences, but the corresponding mRNA when transcribed by an upstream promoter, can also harbour RNA elements which have IRES-like function. Overall, the work shown in the manuscript represents an important addition to the field.

The data shown in the first three figures are robust and support the claims made.

However, in my opinion some more controls are required for Figure 4 to strengthen the data.

1. Ideally in a circular RNA IRES assay there should be no contaminating linear RNA (e.g. if RNA transfections are used) but it is not possible to generate solely circular RNAs using the system described. Therefore in this case, translation could also occur from the linear RNA and is not possible to be entirely certain that the activity measured comes from only circular RNA. The controls shown in 4D are useful and do take this into account to a certain extent, but these need to be carried out for all the circular RNAs derived from the plasmids (ie those used in Figure B and E).

2. In addition, if I have understood this correctly, the data suggest that there is greater activity where there was most circular RNAs generated. For example, HCV transfections have the greatest IRES activity, so is the higher level of activity a reflection of the fact that there is more circular RNA generated?

3. In the paper it is suggested that EMCV is a "weak" IRES, which from other cellular assays does not appear to be the case. Moreover IRESs show cell tropism and HEK cells are not the cell type that would be infected by EMCV. I am therefore not convinced that "weak" in terms of IRES function is an appropriate description. The original paper from the Sarnow lab shows that EMCV works very well on circular RNAs <https://www.science.org/doi/epdf/10.1126/science.7536344> that are directly transfected in to the cytoplasm. Is it simply that relatively little circular RNA is made from the EMCV plasmid?

4. The data could also be related to factor requirement, with HCV requiring very little in terms of canonical initiation factors so perhaps it is unsurprising it works well in this assay, whereas EMCV requires part of eIF4G and has some dependence on eIF4A, perhaps (if the points above are addressed 2 and 3) this could be discussed.

Referee #2:

While the paper's title indicates an interest in a general technique for examining IRES activity, most of the techniques in the paper are not particularly novel, although their combination may be. In the opinion of this reviewer, the goal of the paper is to address the specific controversy of whether Hoxa9 translation is driven by an IRES as illustrated by previous papers by the authors, or whether this is mainly an artifact as argued by two recent papers by other investigators. I am not sure about the significance of this particular pipeline for general IRES examination since (i) usually the 5'UTR is not quite as controversial, (ii) the PAC-BIO assemblies do not appear to be a general solution to this issue and (iii) the smFISH data does not correspond to qRT-PCR and this study does not determine which should be relied on. Without resolving this, it is hard to support using smFISH as a technique in this field.

There are two major independent issues that are addressed concerning Hoxa9: (i) whether and to what proportion the 5'UTR of Hoxa9 contains the putative IRES sequence; (ii) whether or not this sequence has IRES activity, or whether its major effect is due to its effects on transcription.

(i) Does the endogenous 5'UTR contain the IRES sequence

The paper uses three techniques to determine whether the putative IRES sequence is present in the 5'UTR of the Hoxa9 mRNA: qRT-PCR, smFISH, and PAC-BIO long read sequencing.

The simplest to understand and most comparable is qRT-PCR. This paper sees a smaller number of mRNAs that contain the IRES sequence than those that contain the CDS, approximately 20% for IRES compared to CDS, suggesting that 20% of the mRNAs contain the 5'UTR containing the putative IRES sequence. This is a larger percentage than seen in a similar study (Ivanov et al, 2022), but both studies agree that a) there are mRNAs containing this sequence and b) they are not a large fraction of the total mRNA.

The smFISH technique, used for the first time to examine 5'UTR usage of Hoxa9 arrives at a distinct conclusion since the finding is that most transcripts contain both the IRES and CDS (probes overlap in approximately 80% of the transcripts). The conclusion is that "Our new data indicate that already at the transcription start site level, a tissue-specific decision is made to predominantly employ the long mRNA isoform containing the IRES-like element in Hoxa9 mRNAs in somites and the neural tube that leads to translation of the homeobox-containing HOXA9 transcription factor. "

This discrepancy between the studies smFish result (80% of transcripts contain IRES), and their qRT-PCR result (less than 20% of transcripts contain IRES) is explained in the discussion by the ability of smFISH to examine tissue specificity. I agree that the advantage of the smFISH technique to examine tissue specificity, but if this were the cause of the discrepancy, there should be tissues where the results are opposite than what was observed to explain the discrepancy with the qRT-PCR. These are not shown. In the absence of this evidence, the difference is likely to be technical (i.e. differential availability of distinct complexes; other explanations for colocalization, etc) and it is hard to argue against the statement that these technical issues may make this not a reliable technique for this type of analysis.

The final technique used to address this issue is the assembly of PAC-BIO long sequences. One assembly found contains the long 5'UTR, while the shorter 5'UTR is only found on an assembly with a later splice in the CDS. This differs from the previous paper that found an assembly fusing the Hoxa10 transcript with Hoxa9 containing the IRES part of the 5'UTR, but no assemblies with the long 5'UTR connected directly to the CDS. In my opinion, this is not a particularly appropriate technique for this question, since as stated in the paper, the algorithm uses previous annotation in its decision making process, and even knowing the algorithm, the assembly process is difficult to assess without independent evidence. Thus, it is extremely difficult to assess which groups results are more 'correct'. I did not find this comparison useful.

In conclusion, I am not sure much advance has been made on this question without more experiments that successfully address the discrepancy with the smFISH result. There is clearly some mRNA that contains the 5'UTR IRES sequence, but the plurality of evidence from all the studies suggest that it is a small proportion of the total mRNA.

(ii) Is there IRES activity in the long 5'UTR

Figure 3 examines whether the putative IRES region of the UTR has some transcriptional activity in a promoterless context. There is a large section on explanation of why an altered luciferase is less active, although this is irrelevant to the paper. I would suggest removing most, if not all of this discussion. I also do not understand the title "Hoxa IRES-like elements are independent of transcription" as the result of this experiment is that all of the IRES elements show some transcriptional activity in this assay. As pointed out by the reviewer, elements can have both activities. Moreover, transcriptional elements do not act in isolation, so whether or not a sequence has activity in the promoterless assay, does not strongly determine whether this sequence plays an important role in controlling transcription from this genetic locus.

Figure 4 tests the IRES in a circular mRNA context that removes many of the caveats of other techniques set up to examine

cap-independent translation. The circular mRNA appears to be an excellent test for the ability of the sequences to mediate cap-independent translation and these results do show that the complete 5'UTR as well as the P4 element alone does show cap-independent activity. Mutations in the P4 region can abrogate this activity. This is conclusively demonstrated by this assay. This agrees with the previous results of this group demonstrating that this sequence has the ability to promote cap-independent translation.

(iii) Other issues

The discussion has a large section arguing that even if this sequence acts as a transcriptional element, it cannot explain results in the 2015 paper because this paper shows that the mRNA levels do not change. This is not appropriate for this paper, as it does not reflect on any results in this paper and the actual arguments used in the other papers to question this result (whether or not it was conclusively demonstrated that mRNA levels were not changed in the transgenic animal) are not directly addressed and in this reviewer's opinion addressing issues raised in the conflicting papers would require additional experiments using these mice.

Referee #3:

The article by Foch et al aims to demonstrate that HOXA IRESs, that have been questioned by other authors as containing cryptic promoters, function in a circular RNA system. The final data of the paper are convincing regarding internal ribosome entry mediated by several HOXA elements, however it is a pity that the reader must wait Figure 4 to obtain this data. The reviewer appreciates that there is a lot of work, but the article is difficult to read and the figures are very complex. This prevents to focus on the main conclusion of the paper. More data should be provided with the circular RNA to provide a general interest to this paper, in addition to answer to the question about the existence of HOXA. In particular, the absence of EMCV IRES activity in the circular RNA cannot be concluded by "the EMCV IRES is a weak IRES". The title of the paper is not attractive.

Major concerns:

1/ During all the text : Why IRES-like? Many cellular IRESs have been identified and the reviewer does not understand why in the present paper the authors raise a doubt about all these papers by qualifying these IRESs as only "IRES-like". Such a doubt is not acceptable. IRES means internal ribosome entry site: thus if there is an internal entry the is an IRES.

2/ Page 4, Figure 1A: The schema is difficult to understand, too complex. If an mRNA is shown, why indicate TIE and homeobox?

3/ Page 5, Figure 2A: Very difficult to understand again: what is the transcript represented with the legends (top right)? TSS and TES are mentioned in the text but not in the figure. There is often confusion between DNA and RNA in the drawings.

4/ Page 7 and Figure 3B: Why such a decrease? It is surprising to have used such an inefficient reporter in conditions where sensitivity is necessary (and we see later that it cannot detect the activity of EMCV IRES).

5/ Page 7, Figure 3C: This structure prediction seems a very long explanation out of the main purpose of the paper. Maybe this should be in supplementary data?

6/ Page 9, Figure 3F: Again difficult to understand. Regarding the artefactual activity, it is wellknown that there is residual promoter activity in the promoterless constructs (if not, nobody could have measured LucF/LucR activity; this is indeed non sense with a promoterless vector but is published in many papers). In addition, this paragraph does not provide any new information as regards the methodology to analyse IRES activity. At least it should lead to the conclusion that the promoter-less construct is not a useful technology to measure IRES activity, as there is an residual promoter activity that can bias the results.

7/ Page 10, Figure 4: EMCV is not a weak IRES. Although circular RNA is in principle a nice system to detect ribosome internal entry, the absence of EMCV IRES activity, known to work very well in HEK293, means that the test cannot detect all IRES activities.

offer to publish to another EMBO publication or the open access journal Life Science Alliance launched in partnership between EMBO Press, Rockefeller University Press and Cold Spring Harbor Laboratory Press. The full manuscript and if applicable, reviewers' reports, are automatically sent to the receiving journal to allow for fast handling and a prompt decision on your manuscript. For more details of this service, and to transfer your manuscript please click on Link Not Available. **

Editor: Kelly Anderson, EMBO Journal (16.7.2024)

Dear Prof. Leppek,

Thank you for submitting your manuscript for consideration by the EMBO Journal. It has now been seen by three referees whose comments are shown below. Given these opinions and the fact that the EMBO Journal can only afford to accept papers which receive enthusiastic support from a majority of referees, I am afraid we cannot offer to publish it here.

Thank you in any case for the opportunity to consider this manuscript. I am sorry we cannot be more positive on this occasion, but we hope nevertheless that you will find our referees' comments helpful.

Yours sincerely,

Kelly M Anderson, PhD
Editor, The EMBO Journal
k.anderson@embojournal.org

Referee #1:

In the manuscript Koch et al describe a holistic approach, using both standard and novel techniques, to robustly interrogate IRES-like function in mammalian cells. The Hoxa9 5'UTR is used as an example. IRES-like activity in 5' UTRs of cellular mRNAs was described over 25 years ago with work in this area carried out in the Sarnow, Prats and Willis laboratories. However there has been controversy in the translation field in that there has been some dispute about existence of cellular IRESs, albeit from a fairly small number of individuals. In particular, the over-reliance of discistronic reporter vector systems and the use of transient transfections has resulted in a number of sequences being attributed with the ability to directly recruit the ribosome in a cap-independent manner, similar to viral IRESs, when in fact this has actually resulted from splicing or cryptic promoter activity. The ability of certain 5'UTRs within cellular RNAs to recruit ribosomes using a mechanism that is distinct from cap-dependent scanning represents an important research area within the translational control field and there are many independent methods, including the work from the Thompson laboratory and others, which show that IRES-like activity exists in mammalian 5' UTRs, including an unbiased screen by Weingarten-Gabbay which suggests ~10% of mRNAs contain IRES. The authors make the very important and well considered point that there are complex networks within mammalian cells for gene regulation and it is unsurprising that a mRNA can be regulated by many parallel mechanisms, e.g. promoters may be present in the DNA sequences, but the corresponding mRNA when transcribed by an upstream promoter, can also harbour RNA elements which have IRES-like function. Overall, the work shown in the manuscript represents an important addition to the field.

We thank the reviewer for their enthusiasm for our work as well as the importance of our findings. We also thank them for their thoughtful suggestions for improving the manuscript, which we have carried out and described below.

The data shown in the first three figures are robust and support the claims made.

However, in my opinion some more controls are required for Figure 4 to strengthen the data. 1. Ideally in a circular RNA IRES assay there should be no contaminating linear RNA (e.g. if RNA transfections are used) but it is not possible to generate solely circular RNAs using the system described. Therefore in this case, translation could also occur from the linear RNA and is not possible to be entirely certain that the activity measured comes from only circular RNA. The controls shown in 4D are useful and do take this into account to a certain extent, but these need to be carried out for all the circular RNAs derived from the plasmids (ie those used in Figure B and E).

We need to first clarify a misunderstanding that from the linear mRNA only the cap-initiated mRuby is translated into functional protein. However, downstream in the same linear mRNA, an IRES is placed upstream of only a 5' fragment of EGFP. Thus, IRES-initiation will lead to non-productive translation of only the N-terminal part of EGFP (now indicated clearer in Fig. 1A). Secondly, stop codon readthrough

of ribosomes that continue to translate downstream of the mRuby CDS cannot lead to functional EGFP either. Both EGFP fragments are present here, but in opposing orientation (3' part of EGFP-IRES-5' part of EGFP). That means, only upon backsplicing and circularization of the full-length EGFP CDS, when EGFP N- and C-terminus are fused in the circRNA, functional EGFP can be detected translated through the IRES in the circRNA.

The control experiment in Fig. 2C, D (also Fig. S3) shows that upon RNase R digest of all non-circular RNAs the majority of RNA species left is circRNA. This finding indicates successful backsplicing and circularization to yield the circRNA. The FACS-based control experiment in Fig. 2D is very complicated, time-/work-intensive, and expensive. Thus, we performed it for a limited, select handful of constructs that we deemed most important. The observed variability across experiments of the circularization rate across different IRES inserts is more likely due to the several days-long protocol including multiple purification and FACS sorting steps. It is less likely that there is a direct link between insert identity and the rate of circularization. In fact, we now included a set of new experiments with a better control for insert-circularization dependency by using inverse sequence controls for all IRES elements tested in Fig. 1 and 2, now in the new Fig. 1C and 2E. These experiments indicated that differences in length and GC-content (which are identical in the forward and inverse inserts) is not the cause for IRES activity and the EGFP signal we see, but rather the IRES RNA sequence and structure. All inverse controls diminish EGFP signal to that of the empty or HBB negative controls. Finally, we included a whole new set of recently identified cellular IRES-like elements and older examples from the literature and see the same specific diminished activity of the IRES-dependent signal in the inverse controls (Fig. 1C, 2E). We can therefore confidently conclude that circRNA-derived EGFP expression is due to *bona fide* IRES-like activity of the tested inserts.

2. In addition, if I have understood this correctly, the data suggest that there is greater activity where there was most circular RNAs generated. For example, HCV transfections have the greatest IRES activity, so is the higher level of activity a reflection of the fact that there is more circular RNA generated? We indeed cannot exclude that different inserts affect the efficiency of backsplicing and circularization at different rates. This has actually been speculated previously (Meganck *et al.* Mol Ther Nucleic Acids, 2021). However, as just explained, we don't just see more EGFP signal with more circRNAs, but rather that the overall variability in circRNA abundance is due to the nature of the elaborate control experiment of multiple steps of RNase R treatment, RNA purifications and FACS sorting over several days (Fig. S3). This experiment has also been performed by Chen *et al.* (Mol Cell 2021, Figure S1) to show circRNA resistance to RNase R. There only for a single construct, so we cannot estimate the variability in circRNA levels among their circRNA screen hits. It is less likely that there is a direct link between insert identity and the rate of circularization.

To control for the effect of different backsplicing efficiencies across inserts on EGFP expression, we now included inverse sequence controls for all IRES elements tested in Fig. 1 and 2, now in the new Fig. 1C and 2E (as explained above). These inverse controls have the same length and GC-content as the forward IRES-like sequences (both features may influence backsplicing efficiency). We compared the activity of all inverse controls to the respective IRES-like constructs and we see a clear reduction of the EGFP signal to that of the empty and *hHBB* control with the inverse inserts. Additionally, we cannot find support for a direct link between an increased backsplicing efficiency and a stronger EGFP signal in our data. For example, the RNase R data (Fig. 2D) would suggest an increased backsplicing efficiency for *Hoxa9* compared to *Hoxa5* and correspondingly an increased EGFP signal of *Hoxa9* compared to *Hoxa5*. But this is not the case (Fig. 2B), strengthening our initial data interpretation of insert-specific IRES-like activity and technical difficulty to quantify absolute circularization and circRNA abundance in the cytoplasm, as well as highlighting the importance of our new series of inverse sequence controls.

3. In the paper it is suggested that EMCV is a "weak" IRES, which from other cellular assays does not appear to be the case. Moreover IRESs show cell tropism and HEK cells are not the cell type that would be infected by EMCV. I am therefore not convinced that "weak" in terms of IRES function is an appropriate description. The original paper from the Sarnow lab shows that EMCV works very well on circular RNAs <https://www.science.org/doi/epdf/10.1126/science.7536344> that are directly transfected in to the cytoplasm. Is it simply that relatively little circular RNA is made from the EMCV plasmid? We are aware that HEK cells are not the ideal infection model for EMCV but we know from previous studies including our own (Byeon *et al.*, 2022; Leppek, Byeon *et al.*, 2021; Leppek *et al.*, 2022) that the EMCV IRES is functional in HEK and fibroblast cells. We were also puzzled by the low activity of EMCV in the circRNA plasmid we initially detected. Thus, we generated a new plasmid Midiprep and re-sequenced the construct. We observed that the previous plasmid stock was not performing well also because the mRuby signal as an internal control was generally lower from the old EMCV plasmid stock than for all other constructs tested. Data generated with the new plasmid prep are now included in Fig. 1B, 2B. They show activity of EMCV of slower kinetics than HCV. However, EMCV activity is sustained

to similar levels of HCV after 120 hours (Fig. S3A). In addition, the new data for the inverse insert control showed for EMCV the specific reduction of its IRES activity at 72 hours (Fig. 1C, 2E). Thus, EMCV is indeed displaying IRES activity in the circRNA assay and thus finds all required factors for ribosome recruitment expressed in HEK cells.

4. The data could also be related to factor requirement, with HCV requiring very little in terms of canonical initiation factors so perhaps it is unsurprising it works well in this assay, whereas EMCV requires part of eIF4G and has some dependence on eIF4A, perhaps (if the points above are addressed 2 and 3) this could be discussed.

This is a fair point. We so far know from the *Hoxa9* IRES-like element that it does not depend on additional proteins but uses direct binding of this element to rRNA ES9S on the ribosome. We figured the comparison to HCV, requiring only eIF3 and contacts to ES7S, is more fair, than to the multi-factor requiring EMCV IRES. That said, as explained in 3., a fresh EMCV plasmid Midiprep yielded robust IRES activity of EMCV in HEK cells in newly added data in Fig. 1B, C and Fig. 2B, E as explained above. Thus, EMCV is indeed displaying IRES activity in the circRNA assay and thus finds all required factors for ribosome recruitment expressed in HEK cells. With respect to cell type-specificity and co-factor expression across cell types, we know that the 5 cellular IRES-like elements from Byeon et al., 2021, newly included now we find that 2/5 are active in HEK cells (Fig. 1B). For all 5 IRES-like activity has been found to vary across different cell types (Byeon et al., 2021). It remains to be determined whether required co-factor ITAFs absent in HEK are needed for IRES-like activity regulation of the inactive constructs, which can be addressed in follow-up studies. We have now included a discussion of this possibility in the main text.

Referee #2:

While the paper's title indicates an interest in a general technique for examining IRES activity, most of the techniques in the paper are not particular novel, although their combination may be. In the opinion of this reviewer, the goal of the paper is to address the specific controversy of whether *Hoxa9* translation is driven by an IRES as illustrated by previous papers by the authors, or whether this is mainly an artifact as argued by two recent papers by other investigators.

We respectfully disagree with the reviewer that our primary goal of the paper is to refute a dispute in the field. While we have used the *Hoxa* 5' UTRs as a model system for analysis of 5' UTR IRES activity, we felt this was a very important 5' UTR for embryonic development and also an important example of translational control that warranted further analysis with a new toolbox of technologies. This one is perhaps, from the viewpoint of embryonic development, one of the most important putative IRESes to date. However, we have now even further expanded the scope of the paper. In particular, we now provide a whole new Figure 1 where we test a broad set of cellular IRESes from the previous literature using our circRNA reporter system. We tested an additional nine cellular IRESes (and one more viral IRES) in circRNAs and confirmed for 5/9 *bona fide* cellular IRES-like activity. The specificity of which we provide by newly added inverse sequence controls of the IRES inserts (the ideal control for insert length and GC content) that all diminish this activity. We also performed these inverse controls for *Hoxa* IRES-like elements with the same result. This reveals the importance of circRNAs in definitively confirming IRES activity and extending the toolbox of available technologies to characterize IRES activity. In our view the circRNA reporter in itself as characterized in our paper is of tremendous importance for the field.

I am not sure about the significance of this particular pipeline for general IRES examination since (i) usually the 5'UTR is not quite as controversial, (ii) the PAC-BIO assemblies do not appear to be a general solution to this issue and (iii) the smFISH data does not correspond to qRT-PCR and this study does not determine which should be relied on. Without resolving this, it is hard to support using smFISH as a technique in this field.

There are two major independent issues that are addressed concerning *Hoxa9*: (i) whether and to what proportion the 5'UTR of *Hoxa9* contains the putative IRES sequence; (ii) whether or not this sequence has IRES activity, or whether its major effect is due to its effects on transcription.

(i) Does the endogenous 5'UTR contain the IRES sequence

The paper uses three techniques to determine whether the putative IRES sequence is present in the 5'UTR of the *Hoxa9* mRNA: qRT-PCR, smFISH, and PAC-BIO long read sequencing.

The simplest to understand and most comparable is qRT-PCR. This paper sees a smaller number of

mRNAs that contain the IRES sequence than those that contain the CDS, approximately 20% for IRES compared to CDS, suggesting that 20% of the mRNAs contain the 5'UTR containing the putative IRES sequence. This is a larger percentage than seen in a similar study (Ivanov et al, 2022), but both studies agree that a) there are mRNAs containing this sequence and b) they are not a large fraction of the total mRNA.

The smFISH technique, used for the first time to examine 5'UTR usage of *Hoxa9* arrives at a distinct conclusion since the finding is that most transcripts contain both the IRES and CDS (probes overlap in approximately 80% of the transcripts). The conclusion is that "Our new data indicate that already at the transcription start site level, a tissue-specific decision is made to predominantly employ the long mRNA isoform containing the IRES-like element in *Hoxa9* mRNAs in somites and the neural tube that leads to translation of the homeobox-containing HOXA9 transcription factor. "

This discrepancy between the studies smFISH result (80% of transcripts contain IRES), and their qRT-PCR result (less than 20% of transcripts contain IRES) is explained in the discussion by the ability of smFISH to examine tissue specificity. I agree that the advantage of the smFISH technique to examine tissue specificity, but if this were the cause of the discrepancy, there should be tissues where the results are opposite than what was observed to explain the discrepancy with the qRT-PCR. These are not shown. In the absence of this evidence, the difference is likely to be technical (i.e. differential availability of distinct complexes; other explanations for colocalization, etc) and it is hard to argue against the statement that these technical issues may make this not a reliable technique for this type of analysis.

We aimed to resolve the tissue-specific expression of 5' UTR *Hoxa9* isoforms better by using more precise sections of the E11.5 embryo to better quantify tissue-specific expression of the expressed isoforms (Fig. 3). The RNA isolation and RT-qPCR approach from microdissected embryos used previously in all studies by our lab and others (Fig. 5) is lower in resolution to determine the mRNA isoform distribution across tissues than imaging of intact tissues in sectioned embryos (Fig. 3). For an extended mRNA isoform analysis across embryonic tissues, we added newly generated transverse sections (Fig. 3B), as well as a whole mount *in situ* experiment of *Hoxa9* mRNA as a reference (Fig. 3A) for the position of the section. The new smFISH experiments with isoform-specific RNAScope *Hoxa9* mRNA probes clearly indicate that there is a predominant 80% of long IRES-containing mRNA isoform A in the somites 20-28 and strikingly, in addition to what we previously showed, we can now observe 80% of the short isoform B or C expressed in the neural tube. These data allow for quantification of the mRNA isoform at their precise localization, which we could not achieve before without including sections of the neural tube, which is why we could not see this tissue-specific distribution before. These new data show that isoform-specific versions of the *Hoxa9* transcript exist in a tissue and cell type-specific manner in the embryo. As more of the neural tube tissue exists than the *Hoxa9*-expressing regions in a few somites (Fig. 3A) this now also explains the discrepancy between the RNAScope analysis and RT-qPCR results when microdissected regions of neural tube and somites are employed together (Fig. 5), as only a very minor population of cells would express the long isoform in this more crude tissue microdissection. All prior studies have employed this combination of microdissected neural tube and somites for their analysis of *Hoxa9* transcript isoforms. It is also not possible to accurately microdissect only the neural tube or somites at this stage of embryonic development when *Hoxa9* mRNA is expressed. Our findings therefore reveal the power of RNAScope to detect 5' UTR isoforms, which we strongly suspect may also be prevalent among other IRES-containing mRNAs as a means to control gene expression. We feel more confident with the biological meaning of very restricted long isoform expression specifically at posterior regions of the embryo (somites 20-28) that leads to translation of the homeobox-containing HOXA9 transcription factor. Independent work showed that HOXA9 is strongly expressed caudally to somite level 21–22, which corresponds to the tenth thoracic vertebrae (Sekimoto *et al*, 1998; Burke *et al*, 1995) and knockout mice for HOXA9 coding region show a homeotic transformation selectively at this position (Chen, Capecchi, *Dev. Bio.*, 1997). We thank the reviewer for suggesting further work in explaining the discrepancy between the RT-qPCR results and RNAScope imaging analysis. We have greatly benefited from this suggestion.

The final technique used to address this issue is the assembly of PAC-BIO long sequences. One assembly found contains the long 5'UTR, while the shorter 5'UTR is only found on an assembly with a later splice in the CDS. This differs from the previous paper that found an assembly fusing the *Hoxa10* transcript with *Hoxa9* containing the IRES part of the 5UTR, but no assemblies with the long 5'UTR connected directly to the CDS. In my opinion, this is not a particularly appropriate technique for this question, since as stated in the paper, the algorithm uses previous annotation in its decision making process, and even knowing the algorithm, the assembly process is difficult to assess without independent evidence. Thus, it is extremely difficult to assess which groups results are more 'correct'. I did not find this comparison useful.

We would like to emphasize that identification of accurate 5' UTR calling has been subject to debate among numerous 5' UTRs in addition to those of *Hoxa* and is therefore a widespread problem in the field (Akirtava et al 2022). Analysis of available technologies and which of them are situated to address this question is of broad importance to the entire translation community. We therefore determined mRNA isoform expression with PacBio long-read sequencing of microdissected embryonic neural tube and somite RNA not just for *Hoxa9* but now test four additional active cellular IRESes expressed in the embryo. We reanalyzed our data to now provide raw reads and present them, and also here find 5' UTR diversity in the embryo of IRES-containing mRNAs for *Hoxa9*, *Chrdl1*, *Dlx1*, *Cofilin* and *Fmr1*. Our long-read sequencing results can clearly distinguish that there are no *Hoxa9-Hoxa10* fusion transcripts present in the microdissected neural tube and somite RNA from embryos. The algorithms used to assign reads to specific transcript isoforms all have their intrinsic biases because they rely for transcript discovery and quantification on the assignment of reads to a specific isoform, which is challenging for isoforms with identical intron chains or splice junctions, but different transcription start or end sites. Therefore, it is hard to derive a clear answer about which 5' UTR isoforms are present in the RNA. This is a common bottleneck and thus far unaddressed problem in the long-read sequencing field. We therefore, in agreement with the reviewer, now strongly suggest caution in the interpretation of PacBio long read sequencing for the identification of transcript isoforms that was strongly relied on in previous studies to make broad conclusions (Akirtava et al., 2022). We obtained advice from the world's leading long-read sequencing experts Andrey Pribelski (University of Helsinki, Helsinki, Finland) and Angela Brooks (UC Santa Cruz, USA) on our data and which algorithms perform the best. Thus, despite the lack of reads having the complete long *Hoxa9* 5' UTR, the existence of the annotated long and short *Hoxa9* mRNA isoforms in E11.5 mouse embryo somite and neural tube tissues is not contradicted by our sequencing data. For *Dlx1*, *Chrdl1*, *Cofilin*, and *Fmr1* mRNA expression, we found that all their mRNA isoforms containing IRES-like sequences are also expressed in E11.5 mouse embryo somite and neural tissues. A large percentage of the corrected reads, especially prominent for *Dlx1* and *Chrdl1*, were shown to have truncated 5' ends. These truncations likely represent RT fall off or variable transcription start sites which are not fully represented in both Ensembl and RefSeq databases. As such, despite advances in long-read sequencing to identify novel transcripts, we need to account for its limitation especially in definitively capturing the complete 5' UTRs without the support of other orthogonal methods. That is why we integrated RNAScope (Fig. 3) and RT-qPCR isoform quantification (Fig. 5) with the long-read sequencing data to confidently conclude that IRES-containing *Hoxa9* 5' UTR is indeed expressed in the somites and neural tube. We now clearly indicate in the paper the drawbacks of PacBio and as now prominently shown with additional IRES elements in the possible toolbox to confirm IRES elements in 5' UTRs. This is crucial for the field as long read sequencing has been proposed in prior studies to accurately map 5' UTRs.

In conclusion, I am not sure much advance has been made on this question without more experiments that successfully address the discrepancy with the smFISH result. There is clearly some mRNA that contains the 5'UTR IRES sequence, but the plurality of evidence from all the studies suggest that it is a small proportion of the total mRNA.

As mentioned earlier, we performed a new smFISH experiment with RNAScope probes with accurate transverse sections of E11 embryos (Fig. 3B). The new RNAScope experiments on transverse sections with isoform-specific *Hoxa9* mRNA probes clearly indicate that there is a predominant 80% of long IRES-containing mRNA isoform A in the somites and 80% of the short isoform B or C expressed in the neural tube. These new data correlate well with the isoform-specific RT-qPCR from neural tube and somite tissue we already presented before (Fig. 5) and the finding that the PacBio data supports the expression of the long 5' UTR isoform of *Hoxa9* and not a *Hoxa9-a10* fusion transcript (Fig. 5).

(ii) Is there IRES activity in the long 5'UTR

Figure 3 examines whether the putative IRES region of the UTR has some transcriptional activity in a promoterless context. There is a large section on explanation of why an altered luciferase is less active, although this is irrelevant to the paper. I would suggest removing most, if not all of this discussion. I also do not understand the title "Hoxa IRES-like elements are independent of transcription" as the result of this experiment is that all of the IRES elements show some transcriptional activity in this assay. As pointed out by the reviewer, elements can have both activities. Moreover, transcriptional elements do not act in isolation, so whether or not a sequence has activity in the promoterless assay, does not strongly determine whether this sequence plays an important role in controlling transcription from this genetic locus.

We agree with the reviewer on the aspect of the improved luciferase reporter and completely moved the part of the altered luciferase to the supplement as it is not directly relevant to the conclusions of the experiments. We change the title of Fig. 4 to "Cryptic promoter activity from IRES-like element DNA in

promoterless reporters is not relevant for IRES-like mRNA element function.” to better represent the conclusions of the promoterless assay.

The point of including the promoterless assay at all, and optimizing a more sensitive luciferase assay for it to detect traces of cryptic transcription from IRES element stems from wanting to present a complete spread of methods that have collectively been used by the cellular IRES field for IRES characterization. The promoterless assay, however flawed, has been used by others in the past to show that IRES elements can also have promoter activity in an isolated assay, even if this activity does not contribute to and is independent from its IRES function as an mRNA element. IRES elements in mRNAs can have isolated promoter activity in genomic DNA. We have now clarified that the promoterless assay in isolation (as previously used) is not a good assay to study IRES elements. We now include an overview graphic in the end (Fig. 6) where we represent all methods in the toolbox tested and clearly cross out the promoterless assay from the list. We encourage the IRES field to not continue to use it as it does not allow any interpretations for the role of an IRES element in translation.

We agree with the reviewer that most of the IRES elements tested show some artificial transcriptional activity in isolation in the promoterless reporters and that transcriptional activity may be uncoupled from any role of the same element in translation. To strengthen this point, we also now include a new series of promoter-containing plasmid versions of the promoterless reporters (Fig. 4F-H). By placing back the SV40 promoter in front of different IRES-like sequences, tested for viral and *Hoxa* IRESes, we find that IRES-derived promoter activity we see in isolation in promoterless reporters is diminished in presence of a physiological promoter. This renders the cryptic activity meaningless as all IRES-containing 5' UTRs contain different endogenous upstream promoters. This strengthens our finding that promoterless reporters should not be included in the repertoire of tools to study IRES-like elements.

Figure 4 tests the IRES in a circular mRNA context that removes many of the caveats of other techniques set up to examine cap-independent translation. The circular mRNA appears to be an excellent test for the ability of the sequences to mediate cap-independent translation and these results do show that the complete 5'UTR as well as the P4 element alone does show cap-independent activity. Mutations in the P4 region can abrogate this activity. This is conclusively demonstrated by this assay. This agrees with the previous results of this group demonstrating that this sequence has the ability to promote cap-independent translation.

As one of the strongest pieces of evidence for cellular IRES-like activity, we decided to show the EGFP circRNA reporter system first and provide a whole new Figure 1 where we test a broad set of cellular IRESes from previous literature. We tested an additional nine cellular IRESes and one viral IRES control in circRNAs and confirm for 5/9 *bona fide* cellular IRES-like activity. The specificity of which we provide by newly added inverse sequence controls of the IRES inserts (the ideal control for insert length and GC content) that all diminish this activity. We also perform these inverse controls for *Hoxa* IRES-like elements with the same result in Fig. 2.

(iii) Other issues

The discussion has a large section arguing that even if this sequence acts as a transcriptional element, it cannot explain results in the 2015 paper because this paper shows that the mRNA levels do not change. This is not appropriate for this paper, as it does not reflect on any results in this paper and the actual arguments used in the other papers to question this result (whether or not it was conclusively demonstrated that mRNA levels were not changed in the transgenic animal) are not directly addressed and in this reviewer's opinion addressing issues raised in the conflicting papers would require additional experiments using these mice.

We agree that the previous discussion did not focus on the new data enough and we now drastically cut parts in the discussion, as well as in the introduction and results sections, that reference any previous papers. We now instead critically discuss the presented new results.

Referee #3:

The article by Foch et al aims to demonstrate that HOXA IRESs, that have been questioned by other authors as containing cryptic promoters, function in a circular RNA system. The final data of the paper are convincing regarding internal ribosome entry mediated by several HOXA elements, however it is a pity that the reader must wait Figure 4 to obtain this data. The reviewer appreciates that there is a lot of work, but the article is difficult to read and the figures are very complex. This prevents to focus on the main conclusion of the paper. More data should be provided with the circular RNA to provide a general interest to this paper, in addition to answer to the question about the existence of HOXA. In particular,

the absence of EMCV IRES activity in the circular RNA cannot be concluded by "the EMCV IRES is a weak IRES". The title of the paper is not attractive.

As one of the strongest pieces of evidence for cellular IRES-like activity, also based on this reviewer's very thoughtful and important suggestion, we decided to show the EGFP circRNA reporter system first and provide a whole new Figure 1 where we test a broad set of cellular IRESes from the previous literature. We tested an additional nine cellular IRESes and one viral IRES control in circRNAs and confirm for 5/9 *bona fide* cellular IRES-like activity. The specificity of which we provide by newly added inverse sequence controls of the IRES inserts (the ideal control for insert length and GC content) that all diminish this activity. We also performed these inverse controls for *Hoxa* IRES-like elements with the same result. This reveals the importance of circRNAs in definitively confirming IRES activity and extending the toolbox of available technologies to characterize IRES activity. In our view the circRNA reporter in itself as characterized in our paper is of tremendous importance for the field.

With regard to the activity of the EMCV IRES: We were also puzzled by the low activity of EMCV in the circRNA plasmid we initially detected. Thus, we generated a new plasmid Midiprep and re-sequenced the construct. We observed that the previous plasmid stock was not performing well also because the mRuby signal as internal control was generally lower from the old EMCV plasmid stock than for all other constructs tested. Data generated with the new plasmid prep are now included in Fig. 1B, 2B. They show activity of EMCV of slower kinetics than HCV for example. However, EMCV activity is sustained to similar levels of HCV after 120 hours (Fig. S3A). In addition, the new data for the inverse insert control showed for EMCV the specific reduction of its IRES activity at 72 hours (Fig. 1C, 2E). Thus, EMCV is indeed displaying IRES activity in the circRNA assay and thus finds all required factors for ribosome recruitment expressed in HEK cells.

We now provide an improved title: "A versatile toolbox to determine IRES activity in cells and embryonic tissues" that targets a broader reader audience.

Major concerns:

1/ During all the text : Why IRES-like? Many cellular IRESs have been identified and the reviewer does not understand why in the present paper the authors raise a doubt about all these papers by qualifying these IRESs as only "IRES-like". Such a doubt is not acceptable. IRES means internal ribosome entry site: thus if there is an internal entry the is an IRES.

We thank the reviewer for this comment. We now use "IRES-like" only for *Hoxa* IRES elements since that is how they were named in the first instance in prior research papers. Again, this is semantics as in those cases we wanted to distinguish them from IRES elements from viruses. In all other instances now in the revised paper we have changed the nomenclature to IRESes, as suggested by the reviewer.

2/ Page 4, Figure 1A: The schema is difficult to understand, too complex. If an mRNA is shown, why indicate TIE and homeobox?

We present both the annotations as used in the ENSEMBL (isoform A and B) and NCBI (isoform C) databases in the scheme as they differ in the isoforms they include. We clarify this point now in Fig. 3 and its legend. We removed the TIE label from the *Hoxa9* 5' UTR cartoon in Fig. 3A.

3/ Page 5, Figure 2A: Very difficult to understand again: what is the transcript represented with the legends (top right)? TSS and TES are mentioned in the text but not in the figure. There is often confusion between DNA and RNA in the drawings.

The schematic mRNA in the legend in Fig. 5 does not have a particular meaning but explains the display of the different mRNA regions (UTRs and CDS) in the PacBio data below for orientation. We added the labels of the non-abbreviated TSS and TES now in the legend to clarify this. This is different from Fig. 3A, as just explained, where we specifically added "*Hoxa9*" and the ENSEMBL annotation numbers added to the figure to scale. In all figures, it is always the mRNA topology and not DNA that is depicted.

4/ Page 7 and Figure 3B: Why such a decrease? It is surprising to have used such an inefficient reporter in conditions where sensitivity is necessary (and we see later that it cannot detect the activity of EMCV IRES).

To clear a misunderstanding with regard to the promoterless and circRNA reporter assay: The promoterless assay tests for the potential of a DNA sequence to promote transcription from an IRES DNA insert. This is done with a new Nluc-globin fusion reporter. We accept a decrease in Nluc activity with the fusion protein because we can much more reliably use the intron-containing globin sequence for distinguishing the spliced mature reporter mRNA from the plasmid DNA sequence by RT-qPCR. The EMCV insert does not have promoter activity which leads to no luciferase reporter activity (Fig. 4D, E). However, as explained before, the new repeated experiments show that in the circRNA assay with a fresh plasmid prep, EMCV is active (Fig. 1B, C; 2B, E). Overall, we make a stronger point that the

promoterless assay, as used before for claiming that IRES DNA sequences lead to cryptic promoter activity, is not a good assay to distinguish IRES-like activity at the mRNA level from promoter activity at the DNA level in absence of a physiological promoter. Both activities may exist in one sequence at the DNA and RNA level, but the promoterless reporter assay is not very meaningful as it tests the DNA element in isolation. To strengthen this point, we also now include a new series of promoter-containing plasmid versions of the promoterless reporters (Fig. 4F-H). By placing back the SV40 promoter in front of different IRES-like sequences, tested for viral and *Hoxa* IRESes, we find that IRES-derived promoter activity we see in isolation in promoterless reporters is diminished in presence of a physiological promoter. This renders the cryptic activity meaningless as all IRES-containing 5' UTRs contain different endogenous upstream promoters. This strengthens our finding that promoterless reporters should not be included in the repertoire of tools to study IRES-like elements.

5/ Page 7, Figure 3C: This structure prediction seems a very long explanation out of the main purpose of the paper. Maybe this should be in supplementary data?

We agree and moved the structure model to the supplement (Fig. S5), as well as most of the text describing the model to the supplemental figure legend. We restrict the description of the new Nluc fusion reporter to the minimum in the main text.

6/ Page 9, Figure 3F: Again difficult to understand. Regarding the artefactual activity, it is wellknown that there is residual promoter activity in the promoterless constructs (if not, nobody could have measured LucF/LucR activity; this is indeed non sense with a promoterless vector but is published in many papers). In addition, this paragraph does not provide any new information as regards the methodology to analyse IRES activity. At least it should lead to the conclusion that the promoter-less construct is not a useful technology to measure IRES activity, as there is an residual promoter activity that can bias the results.

We agree and we now make it a lot clearer in the text, that the promoterless assay is not useful to uncouple IRES-like activity at the mRNA level from promoter activity at the DNA level of one regulatory element. Both may exist at the same time at the DNA and RNA level. But the promoterless assay does not address both activities simultaneously and may not represent complex gene regulation in presence of an endogenous promoter in genomic DNA, which may be the cause of cryptic promoter activity we see in the promoterless reporter. We now include an overview graphic in the end (Fig. 6) where we represent all methods in the toolbox tested and clearly cross out the promoterless assay from the list. We encourage the IRES field to not continue to use it as it does not allow any interpretations for the role of an IRES element in translation.

7/ Page 10, Figure 4: EMCV is not a weak IRES. Although circular RNA is in principle a nice system to detect ribosome internal entry, the absence of EMCV IRES activity, known to work very well in HEK293, means that the test cannot detect all IRES activities.

We kindly refer to the reviewer's first point for an explanation about the EMCV activity.

—

** As a service to authors, EMBO Press provides authors with the possibility to transfer a manuscript that one journal cannot offer to publish to another EMBO publication or the open access journal Life Science Alliance launched in partnership between EMBO Press, Rockefeller University Press and Cold Spring Harbor Laboratory Press. The full manuscript and if applicable, reviewers' reports, are automatically sent to the receiving journal to allow for fast handling and a prompt decision on your manuscript. For more details of this service, and to transfer your manuscript please click on <https://emboj.msubmit.net/cgi-bin/main.plex?el=A1li6BCcV3A2EThO2X3A9ftdvdnREMPHRXcIL7uXSRfMfQY>. **

Dear Dr Leppek,

Thank you for submitting a revised version of your manuscript. Your study has now been seen by all original referees, who find that their previous concerns have been addressed and now recommend publication of the manuscript. There remain only a few mainly editorial points that have to be addressed before I can extend formal acceptance of the manuscript:

- Please provide the manuscript in .docx format and remove the figures from the ms file and upload as individual Figure files; figure legends should be listed below the References
- Please provide ORCID for both corresponding authors
- Please reduce the number of keywords on the abstract page to five (ideally choosing broad general terms).
- Please rename the Conflict of Interest section into "Disclosure and Competing Interests Statement", in accordance with our updated Guide to Authors (<https://www.embopress.org/competing-interests>)
- As we are switching from a free-text author contribution statement towards a more formal statement based on Contributor Role Taxonomy (CRediT) terms, please remove the present Author Contribution section and instead specify each author's contribution(s) directly in the Author Information page of our submission system during upload of the final manuscript. See <https://casrai.org/credit/> for more information.
- Please provide either a "Yes" or a "Not Applicable" answer to each one of the questions in your Author Checklist which will be sent to you separately by my colleague Hannah Sonntag. In the last column of this checklist, only the sections of the manuscript where the relevant information can be found should be listed (the information per se should be included in the main manuscript file).
- Please update all source file name, title, legend and manuscript callouts to Dataset EV1 instead of Supplemental Table S1 (or from Appendix: "Table S3: Quantification of transcript models discovered by IsoQuant from polyA enriched long-read RNA sequencing of E11.5 embryonic neural tube/somites."); legend should be included as a separate tab/sheet in the Excel file
- Please convert the APPENDIX 1 FILE into PDF format; - title page should contain "Appendix for + ms title" and ToC with the page numbers for the listed items; nomenclature should be Appendix Figure Sx and Appendix Table Sx throughout ms and Appendix PDF; the following needs to be removed from Appendix PDF: "Other supplementary material for this manuscript includes the following: Table S3: Quantification of transcript models discovered by IsoQuant from polyA enriched long-read RNA sequencing of E11.5 embryonic neural tube/somites." or included in ToC if the table is included in Appendix PDF
- Please remove the R&T TABLE from ms file and upload as an individual file using the template from our GTA
- Please provide source data. My colleague Hannah Sonntag will contact you with more information
- Please provide suggestions for a short 'blurb' text prefacing and summing up the conceptual aspect of the study in two sentences (max. 250 characters), followed by 3-5 one-sentence 'bullet points' with brief factual statements of key results of the paper; they will form the basis of an editor-written 'Synopsis' accompanying the online version of the article. Please also provide an altered synopsis image, making sure that the aspect ratio conforms to our website's format - it should be exactly 550 pixels wide and between 300-600 pixels high.
- The "SUPPLEMENTAL INFORMATION" should be removed from ms file
- Please adjust the order of the manuscript sections: Title page with complete author information, Abstract, Keywords, Introduction, Results, Discussion, Methods, Data Availability Section, Acknowledgements, Disclosure and Competing Interests Statement, References, Main figure legends, Tables, Expanded Figure Legends.
- The Figure 3B is reused in Figure Appendix S4. In the legend it states - Appendix Fig S4 related to Figure 3. Please be more specific and state the exact re-use of the cells.
- Please provide the accession ID for the "GSE250214" dataset in the data availability statement.
- Figure Legends (main + EV):
 1. Please note that the legend for figure 3E is missing in the manuscript. This needs to be rectified.
 2. Please indicate what */ **/ ***/ **** represents; if this represents p value(s), please indicate the statistical test used and where appropriate, and specify the exact p value in the legend(s) of figure(s) 1B, C; 2B, E, H; 4D, E, H.

3. Please note that the exact p values are not provided in the legends of figures 4G.
4. Please note that information related to n is missing in the legend of figure 3E.
5. Please note that n=2 in figures 1C, 2D, E; 4D.
6. Please note that the error bars are not defined in the legend of figure 3E.

With best regards,

Cornelius Schneider

Cornelius Schneider, PhD
Editor
The EMBO Journal
c.schneider@embojournal.org

We realize that it is difficult to revise to a specific deadline. In the interest of protecting the conceptual advance provided by the work, we recommend a revision within 3 months (17th Apr 2025). Please discuss the revision progress ahead of this time with the editor if you require more time to complete the revisions. Use the link below to submit your revision:

Referee #1:

The author have addressed my concerns and the additional experiments and further controls have improved the manuscript. In my opinion this work is an important addition to the field

Referee #2:

The paper uses a number of independent techniques to examine possible IRES sequences and in particular addresses some controversies concerning the Hoxa9 IRES. The authors conclude that Circular RNAs are a good technique for examination of IRES activity and that SMFISH can be used to determine the cell specific use of alternative transcripts containing distinct 5'UTRs. In contrast the authors find issues with using Long mRNA sequencing by PACBIO and promoterless plasmids for examining alternative transcripts and IRES activity respectively.

The authors have addressed all my previous objections to the paper and the inclusion of new data for SMFISH in particular strongly addresses an important issue in the field.

The one small non-essential suggestion is some comments on why a well established viral IRES, EMCV, showed no activity in the HEK293T cell circ RNA assay. While cellular IRES that are dependent on specific RBPs to associate with ribosomes and thus are likely to be cell specific, viral IRESes in general do not, and thus, the lack of activity of EMCV suggests the possibility of false negatives using this assay and this requires discussion.

Referee #3:

The authors describe different technologies to study cellular mRNA IRES function. They propose to analyze IRES activity using plasmids coding circRNAs. Interestingly, they demonstrate that the use of promoterless vectors is not relevant to analyze IRES activity as many DNA fragments exhibit a cryptic transcriptional activity when they are in a promoterless context. They show that single molecules of IRES-containing mRNAs can be detected by smFISH imaging, and that the abundance of IRES-containing mRNA isoforms can be measured by PacBio long-read sequencing, however with some limitations. Finally they analyze the translation status of IRES-containing mRNAs by RT qPCR from polysome gradients.

Global comment: The data are convincing and provide important technical information in the IRES field that sometimes remains (unfairly) controversial. The use of circRNAs to analyze IRES activity is indeed convincing, and it is great that somebody finally demonstrate the irrelevance of the promoterless vector test! The second part of the paper is focused on the detection of IRES-containing mRNAs. The technologies are convincing, but it should be more clearly indicated that this allows to distinguish IRES-containing mRNAs among different mRNA isoforms of a given gene when the IRES is already known. It does not allow to detect that a given mRNA contains an IRES if that IRES has not been identified yet! The third part, translation of IRES-containing mRNA, never mentions that and IRES-containing mRNA can also be translated by the cap-dependent mechanism. Globally this important distinction is never mentioned on the paper. Finally, the entire text has issues with punctuation and missing spaces (particularly after parentheses, hpt instead of hrs in the figure legend). Also, the text is very dense.

This information provided in this paper is important for the wider readership of The EMBO Journal, as too many authors have unfairly claimed the non-existence of cellular IRESs, often by using irrelevant promoterless vectors, so that the non-specialist scientist will keep in mind that IRES existence is questionable. This paper clears things up.

Major concerns:

- Page 7, 3rd paragraph. This paragraph lacks of clarity. While the data are convincing about the existence of transcriptional activity in the promoterless empty vector, demonstrating the uselessness of using promoterless vectors as a control for IRES activity, the use of monocistronic vectors prevent to distinguish IRES activity from cap-dependent translation. It would have been useful to use here bicistronic vectors that are the main test used to test IRES activity. Alternatively it should be more clearly explained that these monocistronic constructs are used in the aim of showing that every 5'UTR can artefactually generate a cryptic transcription in a promoter less context.

- In all the text, the ability of IRES-containing mRNAs to be translated by both cap-dependent and IRES-dependent mechanisms is never mentioned.

- Page 8, paragraph 2, line 1: "Long-read sequencing can be used to confirm the expression of an IRES element in a tissue." It is not clear what the authors mean by "expression of an IRES". Is it expression of the corresponding mRNA? By reading the rest of the paragraph it seems that this approach allows to detect different mRNA isoforms, but I do not see how this technology could allow to detect IRESs. It may be useful for specific cases such as Hox mRNAs whose IRESs are already known. Please clarify this question.

Minor concerns:

- The entire text has issues with punctuation and missing spaces (particularly after parentheses).
- Many errors in the text : hpt instead of hrs (hours)
- Figure 1B: why two different scales for the histograms?
- Can you indicate the significance of the p-values in figure legends?
- Fig. 3A: the picture of hybridization is too small to be able to see anything.

Editor: Cornelius Schneider, EMBO Journal (10th January 2025)

EMBOJ: EMBOJ-2024-117998R-Q Referee reports for cross-commenting

Dear Prof. Leppek,

thank you for contacting us regarding the status of your manuscript. We have now received reports from all three referees which you can find below. As you can see the referees are generally positive and support publication of the manuscript at The EMBO Journal. We will therefore invite minor revisions asking you to incorporate the textual requests by referee #2 and #3. We still have to go through our standard manuscript and figure check routine which will require a couple of working days to complete and I will come back with a formal decision letter as soon as we are ready.

With best regards,

Cornelius Schneider

Best regards,

Cornelius Schneider

Cornelius Schneider, PhD

Editor

The EMBO Journal

c.schneider@embojournal.org

Referee #1 (Report for Author)

The author have addressed my concerns and the additional experiments and further controls have improved the manuscript.

In my opinion this work is an important addition to the field

We thank the reviewer for their enthusiasm for our work as well as the importance of our findings. We also thank them for their thoughtful suggestions for improving the manuscript.

Referee #2 (Report for Author)

The paper uses a number of independent techniques to examine possible IRES sequences and in particular addresses some controversies concerning the Hoxa9 IRES. The authors conclude that Circular RNAs are a good technique for examination of IRES activity and that SMFISH can be used to determine the cell specific use of alternative transcripts containing distinct 5'UTRs. In contrast the authors find issues with using Long mRNA sequencing by PACBIO and promoterless plasmids for examining alternative transcripts and IRES activity respectively.

The authors have addressed all my previous objections to the paper and the inclusion of new data for SMFISH in particular strongly addresses an important issue in the field.

We thank the reviewer for their enthusiasm for our work as well as the importance of our findings. We also thank them for their thoughtful suggestions for improving the manuscript.

The one small non-essential suggestion is some comments on why a well established viral IRES, EMCV, showed no activity in the HEK293T cell circ RNA assay. While cellular IRES that are dependent on specific RBPs to associate with ribosomes and thus are likely to be cell specific, viral IRESes in general do not, and thus, the lack of activity of EMCV suggests the possibility of false negatives using this assay and this requires discussion.

We are sorry if the reviewer missed the additional data we provided during our previous round of revision. We observed that the previous plasmid stock was not performing well also because the mRuby signal as an internal control was generally lower from the old EMCV plasmid stock than for all other constructs tested. Data generated with the new plasmid prep are now included in Fig. 1B, 2B. They show activity for EMCV.

Referee #3 (Report for Author)

The authors describe different technologies to study cellular mRNA IRES function. They propose to analyze IRES activity using plasmids coding circRNAs. Interestingly, they demonstrate that the use of promoterless vectors is not relevant to analyze IRES activity as many DNA fragments exhibit a cryptic transcriptional activity when they are in a promoterless context. They show that single molecules of IRES-containing mRNAs can be detected by smFISH imaging, and that the abundance of IRES-containing mRNA isoforms can be measured by PacBio long-read sequencing, however with some limitations. Finally they analyze the translation status of IRES-containing mRNAs by RT qPCR from polysome gradients.

Global comment: The data are convincing and provide important technical information in the IRES field that sometimes remains (unfairly) controversial. The use of circRNAs to analyze IRES activity is indeed convincing, and it is great that somebody finally demonstrate the irrelevance of the promoterless vector test! The second part of the paper is focused on the detection of IRES-containing mRNAs. The technologies are convincing, but it should be more clearly indicated that this allows to distinguish IRES-containing mRNAs among different mRNA isoforms of a given gene when the IRES is already known. It does not allow to detect that a given mRNA contains an IRES if that IRES has not been identified yet! The third part, translation of IRES-containing mRNA, never mentions that and IRES-containing mRNA can also be translated by the cap-dependent mechanism. Globally this important distinction is never mentioned on the paper. Finally, the entire text has issues with punctuation and missing spaces (particularly after parentheses, hpt instead of hrs in the figure legend). Also, the text is very dense. This information provided in this paper is important for the wider readership of The EMBO Journal, as too many authors have unfairly claimed the non-existence of cellular IRESs, often by using irrelevant promoterless vectors, so that the non-specialist scientist will keep in mind that IRES existence is questionable. This paper clears things up.

We thank the reviewer for their enthusiasm for our work as well as the importance of our findings. We also thank them for their thoughtful suggestions for improving the manuscript. We now added sentences into the main text to clarify that 1. the long-read sequencing can only distinguish IRES-containing mRNA isoforms for known IRESes; 2. IRES-containing mRNAs, that do not have a built-in cap-repressive element such as the TIE in *Hoxa9* mRNA, can simultaneously to the IRES route be translated cap-dependently. We further proof-read the text and to improve clarity we abbreviate "hours post transfection" simply as "hrs" and not as "hpt", as defined previously, throughout the text now.

Major concerns:

- Page 7, 3rd paragraph. This paragraph lacks of clarity. While the data are convincing about the existence of transcriptional activity in the promoterless empty vector, demonstrating the uselessness of using promoterless vectors as a control for IRES activity, the use of monocistronic vectors prevent to distinguish IRES activity from cap-dependent translation. It would have been useful to use here bicistronic vectors that are the main test used to test IRES activity. Alternatively it should be more clearly explained that these monocistronic constructs are used in the aim of showing that every 5'UTR can artefactually generate a cryptic transcription in a promoter less context.

We are now more clearly stating in that paragraph that the promoterless reporter assay is not used to assess the IRES activity of a 5' UTR but the unrelated and uncoupled activity of cryptic transcription. It does not allow any claims of cap-independent function of the IRES in the resulting 5' UTRs with unclear start sites because these mRNA are capped. We have extensively used mRNA-normalized bicistronic luciferase assays in previous studies to test *Hoxa* and other IRES-like activity (Xue et al, 2015; Leppek *et al*, 2020; Leppek, Byeon. et al. 2021), thus we did not repeat these experiments here. We refer in the text now to these data.

- In all the text, the ability of IRES-containing mRNAs to be translated by both cap-dependent and IRES-dependent mechanisms is never mentioned.

We now include a sentence stating that in the introduction and in the polysome-qPCR section.

- Page 8, paragraph 2, line 1: "Long-read sequencing can be used to confirm the expression of an IRES element in a tissue." It is not clear what the authors mean by "expression of an IRES". Is it expression of the corresponding mRNA? By reading the rest of the paragraph it seems that this approach allows to detect different mRNA isoforms, but I do not see how this technology could allow to detect IRESs. It may be useful for specific cases such as Hox mRNAs whose IRESs are already known. Please clarify this question.

As we explain above, the long-read sequencing approach is aimed at confirming mRNA isoforms that contain a known IRES sequence in a tissue-specific manner, but it cannot aid to discover new IRES sequences. We now clarified this sentence to: “Long-read sequencing can be used to confirm the expression of different mRNA isoforms that may contain a known IRES element sequence in a specific tissue.” We also added the following sentence right after: “So far, it can only be used to detect and distinguish IRES-containing mRNA isoforms among different expressed isoforms for known IRES elements, but does not aid IRES element discovery nor add to its functional characterization, but rather support its presence or absence.”

Minor concerns:

- The entire text has issues with punctuation and missing spaces (particularly after parentheses). We further proof-read the text to improve clarity and adhere to the EMBO Journal guidelines in terms of formatting.

- Many errors in the text : hpt instead of hrs (hours)

As explained above we abbreviate “hours post transfection” simply as “hrs” and not as “hpt”, as defined previously, throughout the text now. We fixed this in the main text, figure legends and supplemental material.

- Figure 1B: why two different scales for the histograms?

For visual clarity, we find that the viral IRESes are overall much stronger in activity than the cellular IRESes, probably for physiological reason. Showing them side-by-side in the same scale would dwarf the cellular IRESes which is not a fair comparison. We do show viral and cellular IRESes together in one graph in the inverse data graph in Fig. 1C.

- Can you indicate the significance of the p-values in figure legends?

We have a “QUANTIFICATION AND STATISTICAL ANALYSIS” section at the end of the Methods section but now also indicate the meaning of the asterisks based on statistical analysis in Figure 1 representative for all other Figures as follows: “In all figures, data was presented as mean, SD or SEM as stated, and * $p \leq 0.05$ was considered significant (ns: $p > 0.05$; * $p \leq 0.05$; ** $p \leq 0.01$; *** $p \leq 0.001$; **** $p \leq 0.0001$).”

- Fig. 3A: the picture of hybridization is too small to be able to see anything.

We increased the size of the whole mount in situ hybridization in Fig. 3A and indicated more clearly that the probe region in orange in the 3' UTR was used for this staining.

Dear Prof. Leppek,

I am pleased to inform you that your manuscript has been accepted for publication in the EMBO Journal.

Yours sincerely,

Cornelius Schneider, PhD
Editor
The EMBO Journal
c.schneider@embojournal.org
